# A rare PRIMER cell state in plant immunity

Tatsuya Nobori[1,2,3,6], Alexander Monell[1,4], Travis A. Lee[1,2,3], Yuka Sakata[5], Shoma Shirahama[5], Jingtian Zhou[2,4,7], Joseph R. Nery[2], Akira Mine[5] & Joseph R. Ecker[1,2,3✉]

Plants lack specialized and mobile immune cells. Consequently, any cell type that encounters pathogens must mount immune responses and communicate with surrounding cells for successful defence. However, the diversity, spatial organization and function of cellular immune states in pathogen-infected plants are poorly understood[1]. Here we infect *Arabidopsis thaliana* leaves with bacterial pathogens that trigger or supress immune responses and integrate time-resolved single-cell transcriptomic, epigenomic and spatial transcriptomic data to identify cell states. We describe cell-state-specific gene-regulatory logic that involves transcription factors, putative *cis*-regulatory elements and target genes associated with disease and immunity. We show that a rare cell population emerges at the nexus of immune-active hotspots, which we designate as primary immune responder (PRIMER) cells. PRIMER cells have non-canonical immune signatures, exemplified by the expression and genome accessibility of a previously uncharacterized transcription factor, GT-3A, which contributes to plant immunity against bacterial pathogens. PRIMER cells are surrounded by another cell state (bystander) that activates genes for long-distance cell-to-cell immune signalling. Together, our findings suggest that interactions between these cell states propagate immune responses across the leaf. Our molecularly defined single-cell spatiotemporal atlas provides functional and regulatory insights into immune cell states in plants.

Interactions between hosts and microorganisms are heterogeneous for multiple reasons. Multicellular host tissues are composed of diverse cell types that have distinct capacities to respond to microorganisms, and microorganisms can occupy niches heterogeneously distributed in the host. Moreover, individual interactions between cells may occur asynchronously. Thus, diverse cell states can co-exist in a tissue. Such heterogeneity can mask fundamental principles of cellular interactions when hosts and microbes are analysed at the tissue scale.

Plant–pathogen interactions have been studied to understand the molecular mechanisms that underlie host immunity and pathogen virulence[2]. Single-cell RNA sequencing (scRNA-seq) of leaf protoplasts has revealed heterogeneous responses of plants infected by virulent bacterial and fungal pathogens[3,4]. However, our understanding of cell-state diversity is largely limited to a specific cell type (the mesophyll[3]) infected by immunosuppressive virulent pathogens. Moreover, the spatiotemporal dynamics of plant immune responses are unclear owing to the low-throughput nature of transgenic reporter assays[1,5]. Furthermore, we have a limited understanding of the gene-regulatory mechanisms that underlie cell-state diversity, such as specific binding of transcription factors (TFs) to cell-state-specific *cis*-regulatory elements (CREs). Single-nucleus assay for transposase-accessible chromatin followed by sequencing (snATAC–seq) is often used to identify potential CREs in individual cell types and states[6], but it has not been applied to study plant immune responses. These gaps in knowledge represent substantial roadblocks to understanding how the plant immune system operates as a collective entity of cell populations with distinct functions[1].

In this study, we aimed to fill these gaps through single-nucleus multiomic (snMultiome) and spatial transcriptomic analyses of a host plant infected by virulent or avirulent bacterial pathogens in a time-course experiment. Specifically, we use single nucleus RNA-seq (snRNA-seq), snATAC–seq and multiplexed error robust fluorescence in situ hybridization (MERFISH)[7] and the following three bacterial pathogens: *Pseudomonas syringae* pv. *tomato* DC3000 (hereafter DC3000), DC3000 AvrRpt2 and DC3000 AvrRpm1 (hereafter AvrRpt2 and AvrRpm1, respectively). DC3000 is a virulent pathogen that can suppress plant immunity through effectors and toxins[8], whereas AvrRpt2 and AvrRpm1 are avirulent pathogens that carry effectors indirectly recognized by plant nucleotide-binding domain and leucine-rich repeat receptors to initiate effector-triggered immunity (ETI)[9–11]. Our single-cell multidimensional atlas reveals previously uncharacterized cell states, including a rare cell state that emerges at the centre of ETI tissue regions (designated as PRIMER cells). Another cell state surrounding PRIMER cells (designated as bystander cells) is described, and we provide insights into the functions and gene-regulatory mechanisms of these cell states.

## Time-resolved snMultiome analysis

We generated a time-resolved snMultiome atlas of *A. thaliana* leaves infected by DC3000, AvrRpt2 or AvrRpm1 (Fig. 1a). Pathogens were inoculated into leaves at a low dose (optical density at 600 nm (OD$_{600}$) = 0.001), at which only a subset of plant cells are anticipated to

[1]Plant Biology Laboratory, The Salk Institute for Biological Studies, La Jolla, CA, USA. [2]Genomic Analysis Laboratory, The Salk Institute for Biological Studies, La Jolla, CA, USA. [3]Howard Hughes Medical Institute, The Salk Institute for Biological Studies, La Jolla, CA, USA. [4]Bioinformatics and Systems Biology Program, University of California, San Diego, La Jolla, CA, USA. [5]Laboratory of Plant Pathology, Graduate School of Agriculture, Kyoto University, Kyoto, Japan. [6]Present address: The Sainsbury Laboratory, University of East Anglia, Norwich, UK. [7]Present address: Arc Institute, Palo Alto, CA, USA. ✉e-mail: ecker@salk.edu

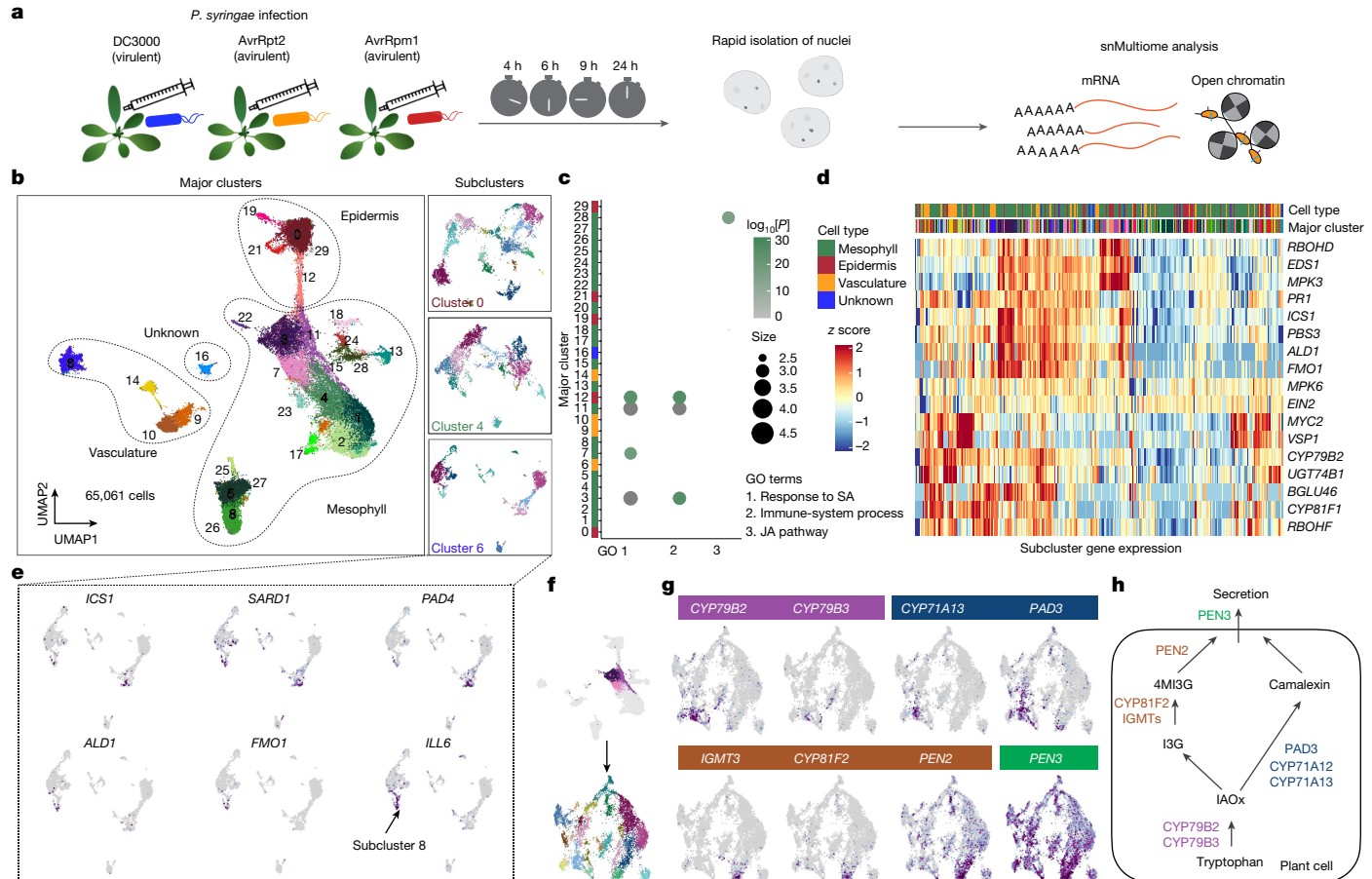

**Fig. 1 | Identification of diverse cell states in *A. thaliana* leaves infected by bacterial pathogens. a**, Schematic of the time-course snMultiome analysis. **b**, Two-dimensional embedding of nuclei from all samples by uniform manifold approximation and projection (UMAP) based on the transcriptomic data. Nuclei are coloured according to Leiden clusters. Left, major clusters. Right, examples of subclustering of major clusters. Cell types were annotated on the basis of marker gene expression. **c**, GO enrichment analysis for marker genes of each major cluster (Extended Data Fig. 2a). GO terms related to defence responses are shown. Adjusted *P* values from a one-sided hypergeometric test followed by Benjamini–Hochberg correction are shown. See Extended Data Fig. 2d for a more comprehensive analysis. **d**, Heat map showing normalized pseudobulk expression of subclusters. Well-characterized defence-related genes are shown. See Extended Data Fig. 3c for a comprehensive analysis. The top bars indicate the cell type and major cluster from which each subcluster is derived from. The colour scheme for the major clusters matches with **b**. **e**, Subclustering of major cluster 6 (PCCs). Defence-related genes showing subcluster-specific expression are shown. **f**, Schematic of subclustering of clusters 3, 7 and 11 (immune-active mesophyll cells). **g**, Expression of genes involved in different steps of the biosynthesis and secretion of tryptophan-derived secondary metabolites shown in **h**. **h**, Simplified schematic of the biosynthesis and secretion of tryptophan-derived secondary metabolites.

encounter pathogen cells. Infected leaves were sampled at four different time points (4, 6, 9 and 24 h). As a control, we prepared plants that were mock-infected with water and sampled after 9 h. Two independent replicates were made for the AvrRpt2-infection 9-h condition and the mock condition. We developed a protocol to rapidly isolate nuclei so that transcriptome and epigenome changes during sample preparation were minimized (Methods). A total of 65,061 cells from 15 samples passed quality control for both RNA-seq and ATAC–seq data analyses (Extended Data Fig. 1a–d). Our snATAC–seq data identified more accessible chromatin regions (ACRs) than previously reported bulk ATAC–seq data from immune-activated leaves[12]. Moreover, the datasets showed large overlaps, which suggests that snATAC–seq can capture both known and unknown ACRs (Extended Data Fig. 1e). ATAC–seq peaks associated with a housekeeping gene (*ACTIN2*) and an immune-related gene (*ICS1*) were consistently detected across replicates, which provided support for the high reproducibility of the snATAC–seq data (Extended Data Fig. 1f). Two independent replicates showed consistent transcriptional reprogramming caused by AvrRpt2 infection (Extended Data Fig. 1g,h), which further confirmed the reproducibility of our data.

Dimensionality reduction and clustering were performed using the snRNA-seq data. Some clusters were enriched in specific infection conditions and at specific time points (Extended Data Fig. 1i), which suggests that the clustering analysis captured distinct cell states induced by pathogen infection. We identified genes specifically expressed in individual clusters (top markers are shown in Extended Data Fig. 2a), which further clarified the identity of each cluster (major cell-type annotations are shown in Fig. 1b). Cell types were also predicted on the basis of ATAC–seq data. For instance, a cluster-specific ACR (peak at chromosome 2 position 11172821–11173529) of clusters 0, 12, 19, 21 and 29 was associated with *FDH*, a marker gene for the epidermis (Extended Data Fig. 2b,c; full names of genes highlighted in this study are provided in the Methods). Overall, the snMultiome data classified cell types and states of *A. thaliana* leaves infected with a pathogen.

Gene ontology (GO) enrichment analysis of marker genes of each cluster identified mesophyll (clusters 3, 7 and 11) and epidermis (cluster 12) cell populations enriched with defence-related genes, including those involved in the defence hormone salicylic acid (SA) pathway[13] (Fig. 1c and Extended Data Fig. 2d). These clusters were well represented in ETI conditions (AvrRpt2 and AvrRpm1 infection) (Extended Data Fig. 1i), which suggests that these cells were responding to the immune-activating pathogens. This finding was also supported by the strong expression of *ICS1*, a key gene for pathogen-induced SA

biosynthesis[14] (Extended Data Fig. 2e). We observed increased accessibility to chromatin regions upstream of the *ICS1* locus after infection by the ETI strains (Extended Data Fig. 1f), a result consistent with a previous bulk ATAC–seq study[12]. Together, our data captured heterogeneous and coordinated changes in defence gene expression and chromatin accessibility during pathogen infection.

## Fine dissection of immune-cell states

Although the major clusters captured immune-active cells in the mesophyll and the epidermis (Fig. 1c), clusters for other cell types, such as the vasculature, contained both immune-active and non-active cells. This result is probably due to strong developmental signatures (for example, vasculature marker genes are expressed at high levels regardless of immune activation). To capture cell-type-specific immune responses, we performed a second round of clustering for each major cluster, which resulted in 429 subclusters with diverse transcriptome patterns (Fig. 1b (right) and Extended Data Fig. 3a). Analyses of selected immune-related genes showed both cell-type-specific gene expression and diverse expression in cell types (that is, cell-state diversity) (Fig. 1d). The subclustering of our snRNA-seq data revealed complex immune responses in leaf tissue that were not captured by bulk RNA-seq (Extended Data Fig. 3h). Moreover, in some cases, genes seemed to be highly co-expressed at the bulk transcriptome level but specifically expressed at the subcluster level (Extended Data Fig. 3i,j), a finding that further highlights the value of single-cell analyses.

A more detailed analysis of specific subclusters revealed strong expression of markers for phloem companion cells (PCCs) (Extended Data Fig. 3d) in major cluster 6. This cluster could be separated into 12 subclusters, some of which were enriched with immune-related genes (Fig. 1e). Important genes for systemic acquired resistance (SAR), such as *ALD1* and *FMO1*, were specifically expressed in PCC subcluster 8 (Fig. 1e). This result indicates that a subset of PCCs contribute to sending long-distance signals to systemic leaves. Notably, we did not observe strong expression of *ALD1* or *FMO1* in other vasculature clusters (Extended Data Fig. 3e). *ILL6* was among the marker genes enriched in PCC subcluster 8 (Fig. 1e and Extended Data Fig. 3f), and this gene is involved in SAR[15]. Together, these findings suggest that this cell population may have a role in SAR, and *ILL6* may be a PCC-specific SAR regulator. We identified additional genes specifically enriched in PCC subcluster 8 but not in other major clusters (Extended Data Fig. 3g). As SAR requires a mobile signal that travels from locally infected leaves to systemic leaves through the vasculature, it is possible that the identified PCC population and marker genes have a specific role in SAR.

As another example of cell-population-specific immune responses, we analysed tryptophan-derived defence-related secondary metabolite pathways in immune-active mesophyll cells (clusters 3, 7 and 11) (Fig. 1f). Distinct expression of genes involved in different steps or pathways of the biosynthesis or secretion of the defence metabolites camalexin and indole glucosinolate was identified (Fig. 1g,h). This result indicates that there is compartmentalization in the activation of defence pathways that can potentially compete for resources (tryptophan). Together, these examples highlight the diversity of plant immune-cell states, thereby confirming the importance of understanding immune signalling pathways and networks at the cell-state level.

## Linking the transcriptome and epigenome

Our snMultiome data enabled us to directly compare mRNA expression and ACRs to identify potential CREs in different cell types, infection conditions and time points (Extended Data Fig. 4a). We identified a total of 29,002 significantly correlated (or linked) ACR–gene pairs within 500 kb of each gene across cells in each infection condition. Most links were within a short distance of gene loci (<400 bp; potential promoter regions), whereas others were more distal (potential enhancer regions)

(Extended Data Figs. 4b and 5a). We summarized peak-to-gene linkage data for each gene by using the maximum Pearson's correlation coefficient values (peak-to-gene linkage score) (Extended Data Fig. 4c). Genes that showed links in all the conditions (cluster 8; Extended Data Fig. 4c) were enriched with cell-type marker genes such as *FDH*, *BCA2* and *MAM1* (Extended Data Fig. 4c,d). Genes that showed links specifically in ETI-activated conditions (cluster 4; Extended Data Fig. 4c) were enriched with immunity-related genes (Extended Data Figs. 4d and 5b). *CBP60G*, a transcriptional regulator of immunity, had multiple ACRs for which accessibility significantly correlated with its mRNA expression (Extended Data Fig. 4e). Genes that showed links specifically in DC3000 infection (cluster 2; Extended Data Fig. 4c) were enriched with jasmonic acid (JA)-related genes (Extended Data Fig. 5b). This finding is consistent with the ability of DC3000 to activate the JA pathway in plants using the toxin coronatine and effectors to suppress plant immunity[16]. These results indicate that coordinated and cell-population-specific reprogramming of chromatin accessibility and gene expression is a key feature of leaf development, immunity and exploitation by virulent pathogens.

Although we identified many ACRs that were closely associated with defence genes, 48% (304 out of 627) of defence genes (marker genes of immune-active clusters 3, 4, 7, 11 and 12) did not show significant links with ACRs. Such non-linked defence genes often had constitutively opened chromatin (Extended Data Fig. 4f) despite cluster-specific changes in gene expression (Extended Data Fig. 4f, violin plot). This finding suggests that there is an additional layer of gene regulation, for example, the expression of upstream TFs. Different motifs were enriched in ACRs 2 kb upstream of linked and non-linked defence genes (Extended Data Fig. 4g,h), which suggests that some defence TFs function together with chromatin reprogramming whereas others do not.

## Identifying TF–ACR–gene modules

To identify cell-type-specific and state-specific gene-regulatory mechanisms, we performed motif enrichment analysis for linked ACRs specific to individual clusters. For instance, comparing an immune-active mesophyll cluster (cluster 3) and a non-immune-active mesophyll cluster (cluster 1) identified the enrichment of motifs for many TFs known to be involved in immunity, including WRKYs and CAMTAs (Fig. 2a). This result highlights the utility of this strategy. We extended this analysis to marker ACRs for all the clusters and identified cluster-specific motif enrichment (Fig. 2b). These findings suggest that different cell types and states use both shared and distinct gene regulation through TF–DNA binding. Moreover, this approach can accelerate the identification of TFs with cell-type-specific or state-specific function.

To analyse motif accessibility at the single-cell level—not the cluster level—we calculated the motif enrichment score for each cell using ChromVAR[17] (Fig. 2c), which revealed heterogeneous accessibility to 465 motifs between and within clusters (Extended Data Fig. 6a). We then asked whether the TF motif enrichment score correlates with TF expression, and the results also showed heterogeneous expression (Extended Data Fig. 6b). We identified TFs that showed top correlations between motif enrichment scores and their mRNA expression in different cell types (Fig. 2d and Extended Data Fig. 6c). For instance, *WRKY46* mRNA expression and accessibility to WRKY46-binding sites overlapped mainly in immune-active mesophyll and epidermal cells across time points (Fig. 2e). This finding suggests that the WRKY46 regulon has a key role during immune responses in these cell types. By searching for genes that are linked with ACRs containing WRKY46-binding sites (Fig. 2f), we identified potential target genes of WRKY46, including many known defence-related genes (Fig. 2g). This result is consistent with the known function of WRKY46 in the SA pathway and defence against *P. syringae*[18]. Finally, we performed target gene prediction for all the TFs shown in Fig. 2d and found that many TFs target genes related to plant defence (Fig. 2h). TFs that belong to the same family tended to show overlapping target genes (Fig. 2h). In addition to WRKYs, motifs

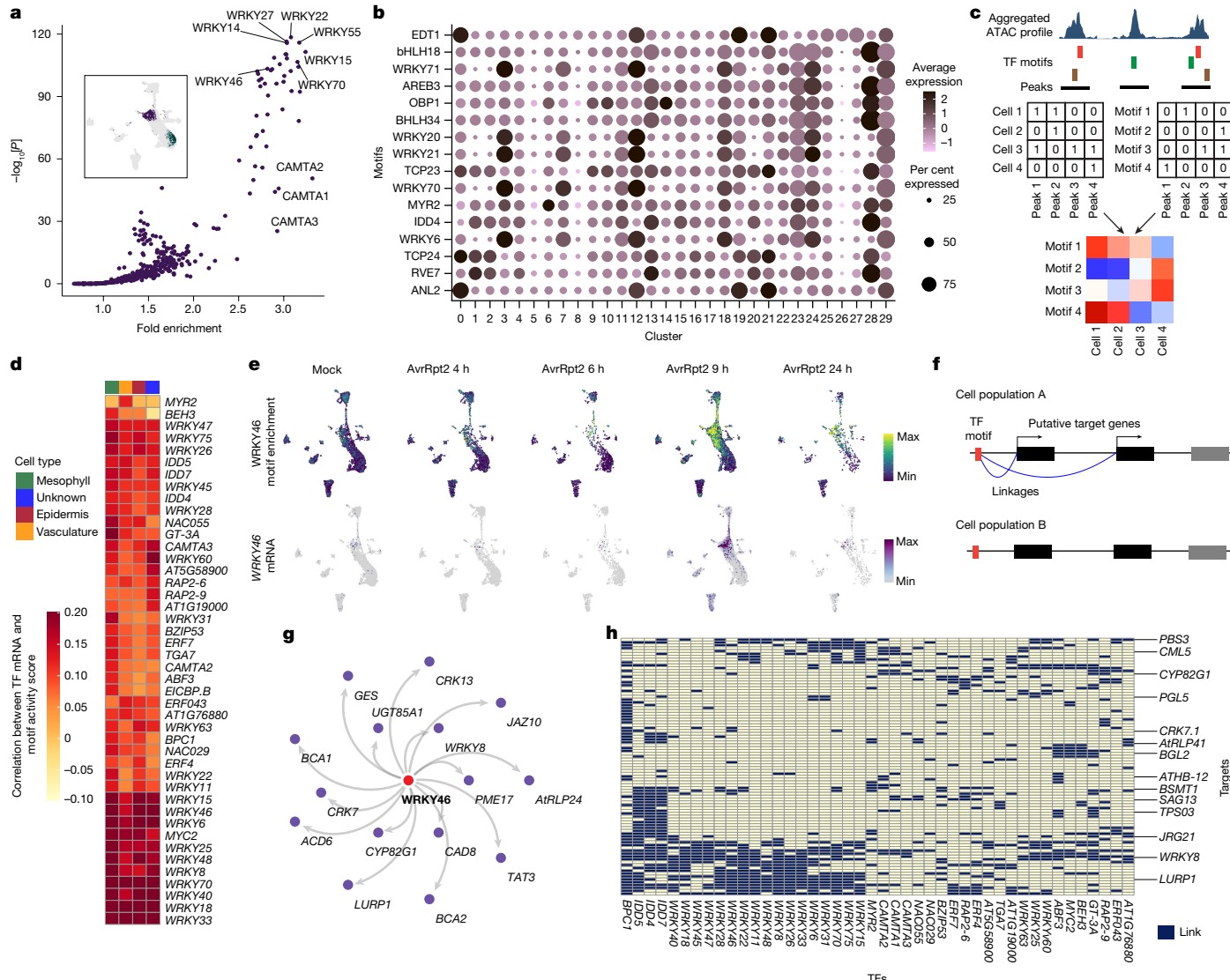

**Fig. 2 | Identification of TF–ACR–gene modules. a**, Scatter plot showing motif enrichment in cluster 3 (immune-active mesophyll) compared with cluster 1 (non-immune-active mesophyll). *P* values were calculated using a one-sided hypergeometric test. The inset shows cluster 3 (purple) and cluster 1 (green). **b**, Top marker motifs enriched in each cluster. **c**, Schematic of the motif enrichment score analysis. ChromVAR[17] was used to generate the cell-by-motif matrix by combining the cell-by-peak (ACR) matrix and the peak-by-motif matrix. **d**, Heat map showing Pearson's correlation coefficients between motif enrichment scores and mRNA expression of the corresponding TFs in each cell type (shown in the top bar). TFs with high correlation are shown. See Extended Data Fig. 6c for visualization of the full data. **e**, Motif enrichment scores (top) and mRNA expression (bottom) of *WRKY46* in the entire snMultiome dataset. **f**, Schematic of target gene prediction of TF motifs. Genes that were significantly linked to any ACR containing a motif of interest were defined as putative targets. **g**, Predicted target genes of WRKY46. **h**, Putative target genes (columns) for TFs for which expression highly correlated with motif activity score (rows). Dark boxes indicate predicted regulatory links.

for TFs such as IDDs, BPC1, TGA7 and ERF7 were predicted to be involved in the regulation of defence-related genes (Extended Data Fig. 6d). In summary, our datasets identified numerous TF–ACR–gene modules that potentially function during pathogen infection.

## Time-resolved spatial transcriptomics

To validate the cell populations identified in the snMultiome analysis and to characterize gene-regulatory modules in the context of tissue, we performed spatial transcriptomics using MERFISH[7] on tissue sections of infected leaves (Fig. 3a and Extended Data Fig. 7a). We curated 500 target genes (Supplementary Table 1), including markers of leaf cell types, genes involved in processes such as immunity, hormone pathways and epigenetic regulation, and a variety of TFs. In addition to MERFISH, we performed standard single-molecule fluorescence in situ hybridization

(smFISH; single-round imaging) on the same tissues targeting *ICS1*. We also analysed target bacterial genes to locate bacterial cells in the tissue section (Methods). We profiled leaves infected by AvrRpt2 at four time points, to match the snMultiome experiments, and mock-infected leaves (Fig. 3a). The spatial localization of the transcripts for 500 genes was decoded after the combinatorial smFISH imaging experiments, and we detected millions of transcripts per sample (Fig. 3b and Extended Data Fig. 7b). MERFISH analysis identified induction of the defence gene *ALD1*, which indicated its target specificity (Extended Data Fig. 7c).

A key step for single-cell analysis of MERFISH data is cell segmentation (Fig. 3a). The standard segmentation approach using nuclear (DAPI) and cytoplasmic (poly(A)) staining did not provide high-quality segmentation results (Fig. 3b). We therefore implemented a separate segmentation approach based on the distribution of transcripts, and this strategy successfully segmented cells (Fig. 3c, Methods and

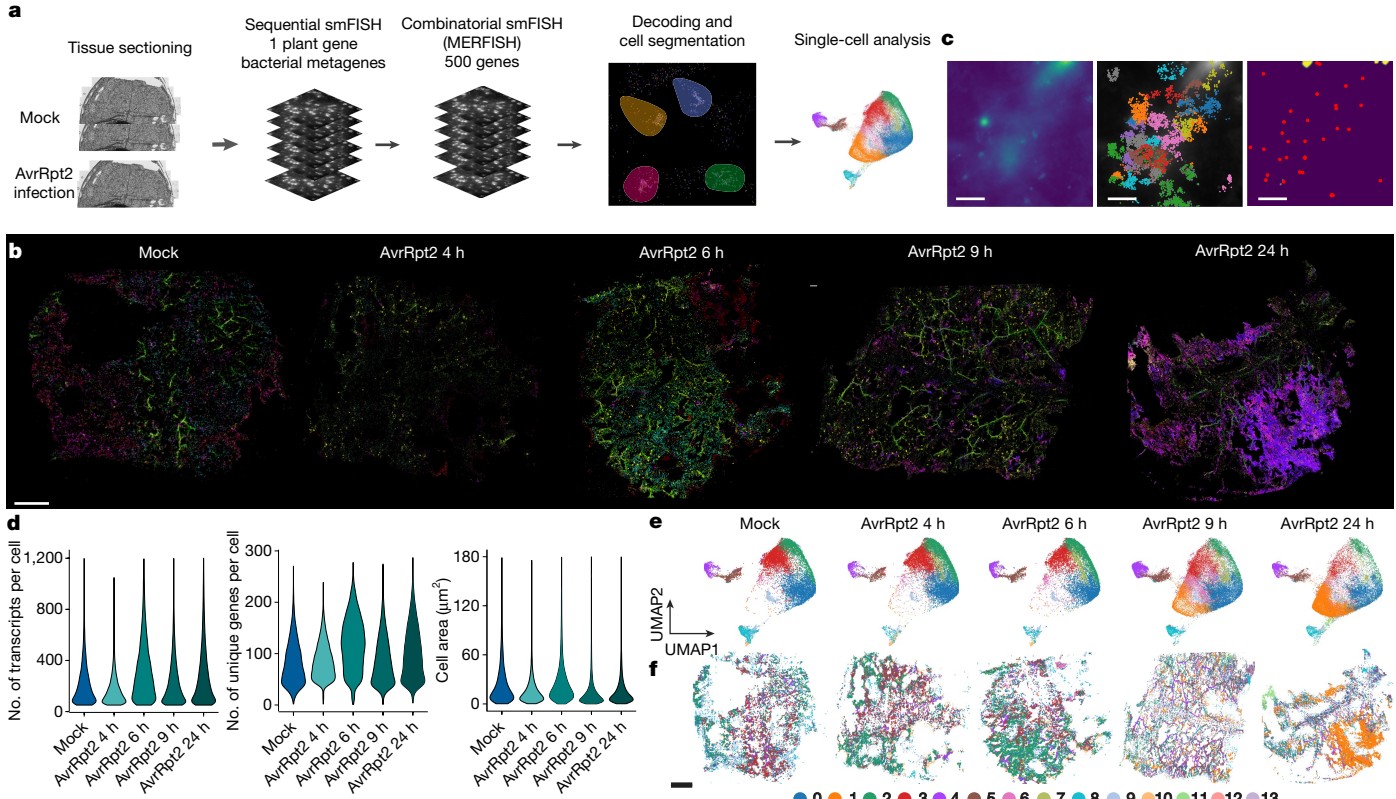

**Fig. 3 | Time-resolved spatial transcriptomics in pathogen-infected leaves.** **a**, Schematic of the MERFISH experiments. $n = 1$ for each condition. See Methods for detailed procedures. **b**, Two-dimensional plots of transcripts for 16 selected genes (Supplementary Table 1) detected using MERFISH in each sample. Plots with transcripts for all 500 genes are provided in Extended Data Fig. 7b. **c**, Left, a representative field of view (FOV) that shows obscure signalling of DAPI-stained nuclei. Middle, transcript-based segmentation in the same FOV (Methods). Transcripts were coloured on the basis of assigned cells. Right, centroids of cells detected using the transcript-based segmentation method (red dots) and the result of failed DAPI-staining-based segmentation (yellow region). Similar patterns were observed across FOVs and samples. A systematic quantitative analysis is provided in Extended Data Fig. 7f. **d**, Violin plots showing the number of transcripts per cell (left) and unique genes per cell (middle) detected in each MERFISH sample and the area size per cell (right). **e**, UMAP embeddings of cells in each sample based on the expression of 500 genes detected using MERFISH. All MERFISH samples are integrated, and cells are coloured on the basis of de novo Leiden clusters. **f**, Spatial mapping of Leiden clusters in each sample using the same colour scheme as in **e**. Scale bar, 40 μm (**c**) or 1 mm (**b**,**f**).

Extended Data Fig. 7d). After segmentation, transcripts were assigned to cells, which produced a cell-by-gene matrix with each cell having its spatial coordinates. Overall, we detected a median of 161 transcripts and 79 genes per cell from a total of 121,998 cells from 5 leaf sections (1 mock infection and 4 time points after AvrRpt2 infection) (Fig. 3d). We performed integration and de novo clustering of all MERFISH samples, which identified 14 clusters (Fig. 3e and Extended Data Fig. 7g). These clusters were spatially mapped in individual samples, which revealed spatially organized cell populations (Fig. 3e). For instance, a de novo MERFISH cluster captured vasculature cells in tissue sections (Extended Data Fig. 7h). The single-cell and quantitative gene-expression properties of MERFISH provide the opportunity to integrate spatially resolved MERFISH data and molecular-information-rich snMultiome data.

## Integrating snMultiome and MERFISH

We integrated MERFISH and snMultiome data (five conditions matching the MERFISH samples; Fig. 4a) using the shared 500 genes, and cluster labels defined by snMultiome were transferred to the MERFISH cells (Fig. 4a,b and Methods). On the basis of this data integration, we spatially mapped clusters defined in the snMultiome data. Major cell types defined by snMultiome were successfully mapped on the expected regions in a MERFISH tissue sample (Fig. 4c), which indicated successful data integration. This integration of MERFISH and snMultiome data enabled us to explore the spatial distribution of cell populations defined in the snMultiome analysis.

Using this integrated dataset, we spatially imputed the entire transcriptome and chromatin accessibility information (Extended Data Fig. 8a–d). Imputed *ICS1* (not included in the MERFISH panel) expression accurately predicted the real spatial expression of *ICS1* (based on smFISH) (Extended Data Fig. 8a,b), a result that confirmed the accuracy of data imputation. We also spatially imputed ATAC activity scores (Extended Data Fig. 5f) of *ICS1* and *ALD1*, and these showed consistent patterns with mRNA expression (Extended Data Fig. 8c). The motif activity of HSFB2b was predicted to be high in the immune-active regions of a leaf at 24 h post inoculation (h.p.i.) (Extended Data Fig. 8d), which was consistent with the mRNA expression pattern of *HSFB2b* validated by MERFISH (Extended Data Fig. 8d). Overall, these results indicate that spatial data imputation of the transcriptome and the epigenome was accurate, which enabled the analysis of gene-regulatory mechanisms at single-cell and spatial resolution. To facilitate exploration of our data, we imputed all 25,299 transcripts detected in the snMultiome analysis and 465 motif enrichment scores on the 5 tissues used in MERFISH experiments and made the data available on a data browser (https://plantpathogenatlas.salk.edu).

## Modelling immune-response dynamics

To understand the temporal dynamics of immune responses, we applied pseudotime analysis to our snMultiome data. Pseudotime analysis aligns cells as a trajectory based on their gene expression, which is commonly used for inferring developmental trajectories[19]. We proposed

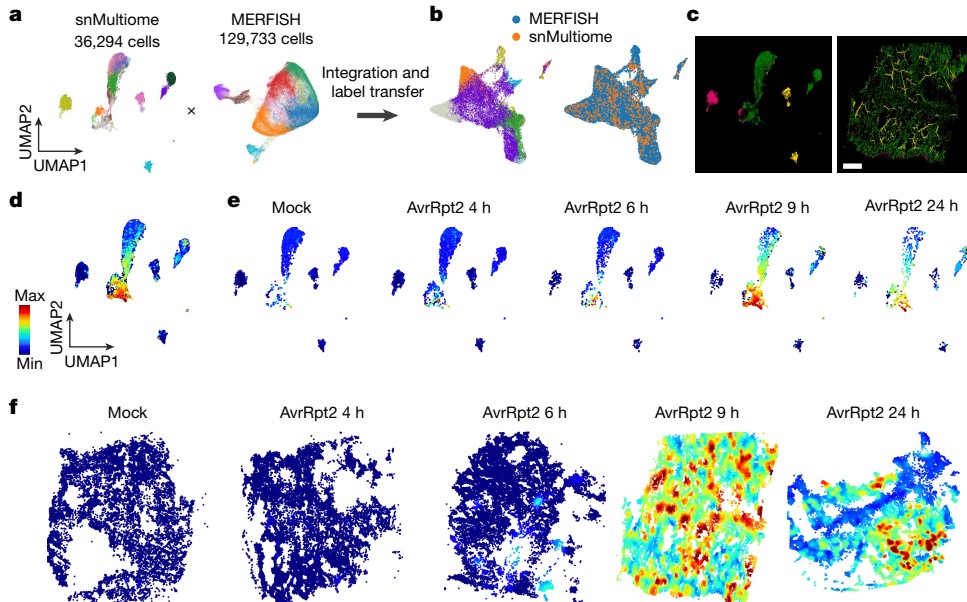

**Fig. 4 | Integration of snMultiome and spatial transcriptome data.**
**a**, Integration of snMultiome data (mock infected and AvrRpt2 infected; five samples in total) and the MERFISH data shown in a UMAP. **b**, Integrated UMAP. Left, nuclei and cells are coloured on the basis of cluster labels, which were transferred from snRNA-seq to MERFISH (Methods). Right, the same integrated UMAP coloured by assay types. **c**, Left, UMAP of snMultiome data coloured on the basis of major cell types: epidermis (magenta), mesophyll (green) and vasculature (yellow). Right, spatial mapping of snMultiome cells coloured on the basis of major cell types. AvrRpt2 9 h.p.i. sample was used. Scale bar, 1 mm. **d**, Pseudotime values calculated for mesophyll cells in the snRNA-seq data. **e**, UMAP plots showing pseudotime values in cells from each time point. **f**, Spatial mapping of pseudotime values based on data integration and label transfer.

that by applying pseudotime analysis to a single developmental cell type with various immune states, we could better model the temporal dynamics of heterogeneous infection and immune responses in an infected leaf. Calculation of pseudotime scores for mesophyll cells (Fig. 4d and Methods) showed that the distribution of predicted pseudotime scores in each sample was consistent with what is expected from the real sampling time point (Fig. 4e). That is, cells with low pseudotime scores were enriched at early time points, whereas cells with high pseudotime scores emerged at later time points. This result indicated that the temporal dynamics of immune responses were successfully modelled. Cells with a wide range of pseudotime scores coexisted at 9 and 24 h.p.i., which suggests that cellular immune responses are asynchronous in infected leaves.

To understand the spatial distribution of heterogeneous immune states, we spatially mapped the pseudotime scores using label transfer from snMultiome to MERFISH data (Fig. 4f). The spatial mapping of the pseudotime scores revealed immune-active areas that were distributed in pathogen-infected leaf tissues in a restricted manner (Fig. 4f). These immune-active areas expanded over time and seemed to merge at 24 h.p.i. (Fig. 4f). Notably, immune states also dynamically change over time in each immune-active area, with older immune-active cells (higher pseudotime scores) being surrounded by younger immune-active cells (lower pseudotime scores) (Fig. 4f). These results indicate that the oldest immune-active cells are probably plant cells that had direct contact with pathogen cells at early time points and that immune responses spread to surrounding cells through cell–cell communication over time.

## Spatial mapping of bacteria

We sought to investigate whether spatially restricted immune-gene expression can be explained by the distribution of bacterial cells. smFISH targeting bacterial metagenes (19 highly expressed genes) detected bacterial colonies at 24 h.p.i., which we overlaid with the spatial map of the pseudotime scores (Extended Data Fig. 8g). As a control,

we performed another MERFISH experiment using an *A. thaliana* leaf infected by the immunosuppressive pathogen DC3000 at 24 h.p.i. (Extended Data Fig. 8e–g). The distribution of the immune-activating strain AvrRpt2 overlapped with tissue regions with high pseudotime scores (immune heightened) in contrast to the immunosuppressive DC3000 strain (Extended Data Fig. 8e–g). We confirmed this observation by quantitatively analysing the neighbouring plant cells of individual bacterial colonies (Extended Data Fig. 8h,i). Taken together, these results indicate that immune-active regions defined by the pseudotime analysis interact with the ETI-triggering pathogen; we also captured potential immunosuppression by the virulent DC3000 pathogen.

## PRIMER and bystander cells

We systematically identified genes for which expression significantly changed across the pseudotime trajectory in the MERFISH data using cell-type-specific inference of differential expression (C-SIDE)[20] (Fig. 5a and Methods). *BON3*, *ALD1* and *FMO1* were among the genes identified for which expression was higher towards the centre of immune-active regions (Fig. 5a). *BON3* was induced in highly restricted areas in the tissue after infection by AvrRpt2 (at 9 h.p.i.) (Fig. 5b), which was distinct from *ALD1* and *FMO1* (Fig. 5b–d and Extended Data Fig. 9a). Our subclustering analysis of snMultiome data confirmed the pattern observed with the MERFISH results (Fig. 5e), which indicates that we identified two distinct immune cell states. Expression of *BON3* was enriched in cells with the highest pseudotime scores (that is, the oldest immune-active cells) (Fig. 4e,f and Extended Data Fig. 9b), which suggests that these cells are early responders to pathogen invasion. Therefore, we designate this cell state as PRIMER cells, from which immune responses might spread. Cells surrounding PRIMER cells are designated as bystander cells.

PRIMER and bystander cells showed distinct transcriptional and epigenetic signatures. PRIMER cells were enriched in CAMTA motifs and a GT-3A motif, whereas bystander cells were enriched in WRKY

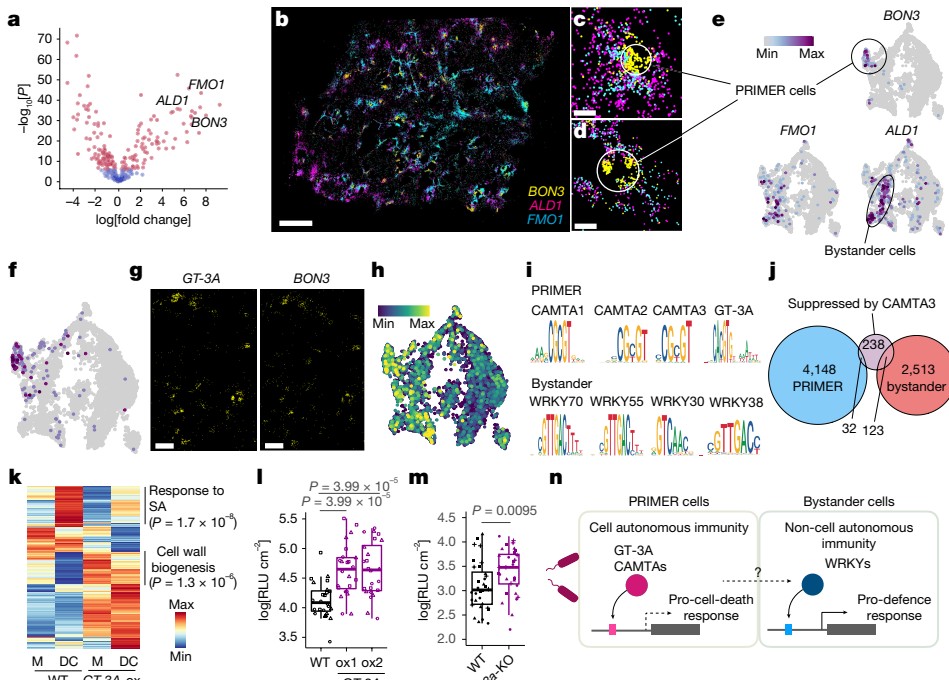

**Fig. 5 | Identification and functional characterization of immune-cell populations. a**, MERFISH-analysed genes showing significant changes in expression across the pseudotime trajectory (Fig. 4f) in an AvrRpt2 9 h.p.i. sample. Positive fold change values indicate higher expression in cells with high pseudotime values. Top differentially expressed genes (DEGs; $P < 1 \times 10^{-8}$) are labelled in red. **b–d**, Spatial expression of *ALD1*, *FMO1* and *BON3* at 9 h.p.i. (**b**) and cropped images of two different locations in **b** (**c** and **d**). **e**, Expression of *ALD1*, *FMO1* and *BON3* in immune-active subclusters from snRNA-seq data. **f–h**, mRNA expression (**f**), MERFISH images (**g**) and motif activity (**h**) of GT-3A in immune-active mesophyll cells in the snMultiome data. *BON3* expression is also shown in **g**. **i**, Enriched motifs linked to genes enriched in PRIMER or bystander cells. **j**, Venn diagram showing overlapping of genes highly expressed in PRIMER or bystander cells with genes previously shown to be suppressed by CAMTA3 (ref. 39). **k**, Bulk RNA-seq of wild-type (WT) plants and *GT-3A*-ox plants

treated with water (M) or DC3000 (DC) at 24 h.p.i. DEGs (adjusted $P < 0.05$, $|\log_2$-adjusted fold change $| > 1$) between WT and *GT-3A*-ox after infection are shown. GO terms enriched in specific gene sets are shown with adjusted $P$ values. Three replicates per condition. **l,m**, Growth of DC3000 (**l**) or AvrRpt2 (**m**) in Col-0 (WT) plants compared with *GT-3A*-ox plants (**l**) and *gt3a*-KO plants (**m**) at indicated times. $n = 24$ (**l**) and $n = 32$ (**m**) leaves from 3 and 4 independent experiments, respectively. RLU, relative light unit. **k–m**, Pathogens were syringe-infiltrated at a suspension dose of $OD_{600} = 0.001$. Adjusted $P$ values were calculated using two-tailed Student's $t$-test with Benjamini–Hochberg correction. **l,m**, Results are shown as box plots, in which boxes represent the 25th–75th percentiles, the centre line indicates the median, and whiskers extend to minimum and maximum values within 1.5 times the interquartile range. **n**, Proposed model for the potential role and regulation of immune-cell states. Scale bar, 100 μm (**c,d**), 200 μm (**g**) or 1 mm (**b**).

motifs (Fig. 5i and Extended Data Fig. 9i). Genes previously shown to be repressed by CAMTA3 were significantly overrepresented in bystander cells compared with PRIMER cells (Fig. 5j; false discovery rate (FDR) = $5.0 \times 10^{-46}$, with hypergeometric test correction using the Benjamini–Hochberg method), a result that supports the transcriptional-repressive role of CAMTA3 in PRIMER cells.

To understand the function of PRIMER cells and their gene-regulatory mechanisms, we investigated an snMultiome subcluster in which *BON3* was enriched (subcluster 4). Although many PRIMER cell marker genes were previously uncharacterized, we found several known genes, including *WRKY8* and *LSD1*, as PRIMER cell markers (Extended Data Fig. 9c). Notably, *BON3*, *WRKY8* and *LSD1* are all negative regulators of immunity[21–23], although the precise mechanisms of how they suppress immune responses remain elusive.

Further marker gene analysis of the PRIMER cell cluster identified *GT-3A* (which encodes a trihelix DNA-binding TF)[24], the function of which in leaf immunity remains uncharacterized (Fig. 5f). *GT-3A* showed a spatial expression pattern similar to that of *BON3* (MERFISH; Fig. 5g), and the GT-3A motif was highly accessible in PRIMER cells (snATAC–seq; Fig. 5h), which implies that it has a gene-regulatory function in this cell population. To understand the role of *GT-3A*, we generated transgenic *A. thaliana* plants that ectopically overexpress this gene (*GT-3A*-ox). Bulk RNA-seq analyses revealed impairment in the induction of genes involved in the SA pathway of *GT-3A*-ox plants infected by DC3000 (Fig. 5k). Furthermore, two independent lines of

*GT-3A*-ox were more susceptible to DC3000 infection (Fig. 5l), which suggests that this previously unidentified cell-state-specific TF can negatively regulate immunity. We also tested plants in which *GT-3A* was knocked out (*gt3a*-KO), and this mutant was more susceptible to AvrRpt2 infection (Fig. 5m). This result indicates that GT-3A function in PRIMER cells is required for optimal defence against avirulent pathogens. snRNA-seq analysis of *gt3a*-KO plants revealed genes that are potentially regulated by GT-3A, either directly or indirectly. Among such genes, *PUB36* expression in PRIMER cells was significantly impaired in *gt3a*-KO plants at both 9 and 24 h.p.i. Notably, *PUB36* has a GT-3A-binding motif in the upstream region (Extended Data Fig. 9d). SA-related genes, including *ALD1*, were expressed at lower levels in bystander cells in *gt3a*-KO plants than in wild-type plants (Extended Data Fig. 9e,f). This finding suggests that induction of *GT-3A* in PRIMER cells is important for the proper induction of defence genes in surrounding cells. In summary, through the integration of time-resolved snMultiome and spatial transcriptome data, we identified previously unknown immune-cell states and a cell-population-specific TF that regulates plant immunity.

## Discussion

Plant immunity comprises a multicellular network in which individual cells interpret input signals with their distinct molecular networks and communicate with other cells. Our molecularly defined spatiotemporal

atlas of pathogen-infected leaves revealed various cell states in transcriptome and epigenome detail. This resource also provides a means to investigate individual cell states that have been obscured in conventional bulk or dissected tissue analyses and by live imaging of a limited number of reporter lines. For instance, we identified a PCC subpopulation with a distinct state characterized by the induction of SAR genes (Fig. 1e) and mesophyll subpopulations that activate different branches of tryptophan-derived defence metabolite pathways (Fig. 1g).

We also described a rare cell state located at the nexus of immune-active hotspots, which we termed the PRIMER cell state (Fig. 5). In addition to mapping previously characterized genes (*BON3*, *WRKY8*, *LSD1* and *CAMTA3*) with common immunosuppressive functions to PRIMER cells, our integrative snMultiome and MERFISH analyses identified another PRIMER cell marker gene, *GT-3A*, which encodes a TF, and demonstrated that it contributes to plant immunity against pathogen infection (Fig. 5). Comparisons between PRIMER cells and their surrounding cells (bystander cells) revealed distinct transcriptional and epigenetic landscapes (Fig. 5). It is possible that there are additional specialized cell states in PRIMER and bystander cells. A deeper understanding of immune-cell states requires the development and application of new methodologies, such as imaging techniques that visualize both pathogen effector proteins and numerous plant genes simultaneously in three dimensions at single-cell resolution[25,26] (see the section 'Limitation of this study' in the Supplementary Information).

Previous studies have shown that GT-3A negatively regulates plant defence against nematode infection in the root[27], which suggests that GT-3A may have a role in defence against different types of pathogens that infect other tissues. In support of this hypothesis, *GT-3A*-ox plants were more susceptible to the fungal pathogen *Colletotrichum higginsianum* (Extended Data Fig. 9j).

Recent discoveries have shed light on the role of nucleotide-binding domain and leucine-rich repeat receptors as calcium channels or NADases[28–35]. However, the exact mechanisms that underlie ETI activation and how this process effectively suppresses pathogen growth are not fully understood. It has been proposed that localized acquired resistance (LAR)[36] may be important for ETI[37]. LAR is a strong defence response in cells surrounding those that have been exposed to pathogen effectors. We revealed the spread of ETI responses from PRIMER cells (Fig. 5e) and captured potential LAR responses with detailed molecular information. Notably, *ALD1* and *FMO1*, encoding canonical SAR components, were expressed in bystander cells but not in PRIMER cells (Fig. 5b–e), which indicates that the SAR pathway has a specific role in LAR responses. This idea is plausible given the long-distance signalling capability of SAR. We propose that PRIMER cells may undergo hypersensitive cell death, which subsequently sends signals to neighbouring cells that activate immune responses, including the SAR pathway. As activation of the SA pathway can suppress cell death[38], GT-3A may contribute to cell death in PRIMER cells by suppressing SA signalling (Fig. 5n). Further analysis of our data could uncover the roles of various immune cell states and how these cells communicate with surrounding cells to confer successful defence.

In addition to identifying previously unknown immune-cell populations, our snMultiome data predicted numerous putative TF–ACR–gene modules. The successful integration of snMultiome and MEFISH data enabled us to spatially map gene expression and chromatin states (Extended Data Fig. 8a–d). These results can be used to discover previously uncharacterized immunity-related genes, and defence-related CREs can also be identified through de novo motif analysis.

Finally, we built a database (https://plantpathogenatlas.salk.edu) to facilitate the exploration of previously uncharacterized cell populations associated with disease and resistance with spatial and temporal information and potential regulatory mechanisms. Our database can be used for hypothesis generation and testing and will catalyse new discoveries of molecular mechanisms that underlie plant–microorganism interactions at high resolution.

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

## Methods

### Reagents and kits

The following reagents and kits were used: ammonium persulfate (Sigma, 09913); BSA solution (Sigma, A1595); Chromium Next GEM Single Cell Multiome ATAC + Gene Expression kits (10x Genomics, PN-1000283); Corning Falcon cell strainers (Corning, 08-771-2); dithiothreitol (Thermo, R0861); EDTA, pH 8.0 RNase-free (Invitrogen, AM9260G); KCl (2 M) RNase-free (Invitrogen, AM9640G); MACS SmartStrainers (Milteny Biotec, 130-098-458); MERSCOPE 500 Gene Imaging kits (Vizgen, 10400006); MERSCOPE 500 Gene Panel (Vizgen, 10400003); MERSCOPE Sample Prep kits (Vizgen, 10400012); *N,N,N′,N′*-tetramethylethylenediamine (Sigma, T7024-25ML); NaCl (5 M) RNase-free (Invitrogen, AM9759); NP40 (IGEPAL CA-630) (Sigma, I8896); paraformaldehyde (Sigma, F8775); PBS (10×) pH 7.4 RNase-free (Thermo Fisher, AM9625); protease inhibitor cocktail (Sigma, P9599); Protector RNase inhibitor (Sigma, 3335402001); Scigen Tissue-Plus OCT compound (Fisher, 23-730-571); spermidine (Sigma, S2626); spermine (Sigma, 85590); Triton-X (Sigma, 93443); and UltraPure 1 M Tris-HCl buffer pH 7.5 (Invitrogen, 15567027).

### Gene symbols, names and ordered locus names

The following genes were highlighted in this study: *ALD1* (*AGD2-LIKE DEFENCE RESPONSE PROTEIN 1, At2g13810*); *BCA2* (*BETA CARBONIC ANHYDRASE 2; At5g14740*); *BON3* (*BONZAI 3; At1g08860*); *CBP60g* (*CALMODULIN BINDING PROTEIN 60-LIKE G; At5g26920*); *FDH* (*FIDDLEHEAD; At2g26250*); *FMO1* (*FLAVIN-DEPENDENT MONOOXYGENASE 1; At1g19250*); *GT-3A* (*TRIHELIX DNA-BINDING FACTOR (GT FACTOR) BINDING 3A; At5g01380*); *ICS1* (*ISOCHORISMATE SYNTHASE 1; At1g74710*); *LSD1* (*LESION SIMULATING DISEASE 1; At4g20380*); *HSFB2b* (*HEAT SHOCK TRANSCRIPTION FACTOR B2b; At4g11660*); *ILL6* (*IAA-LEUCINE RESISTANT (ILR)-LIKE GENE 6; At1g44350*); *MAM1* (*METHYLTHIOALKYLMALATE SYNTHASE 1; At5g23010*); *WRKY8* (*At5g46350*); and *WRKY46* (*At2g46400*).

### Plant growth and bacterial infection for single-cell and spatial analyses

*A. thaliana* Col-0 was grown in a chamber at 22 °C with a 12-h light period and 60–70% relative humidity for 30–31 days. Bacterial strains were cultured in King's B liquid medium with antibiotics (rifampicin and tetracycline) at 28 °C. Three bacterial strains—*P. syringae* pv. *tomato* DC3000 with an empty vector (pLAFR3), avrRpt2 (pLAFR3) and avrRpm1 (pLAFR3)—have been previously described[9–11]. Bacteria were collected by centrifugation and resuspended in sterile water to an $OD_{600}$ of 0.001 (approximately $5 \times 10^5$ c.f.u. ml$^{-1}$). In total, 20 *A. thaliana* leaves (4 fully expanded leaves per plant) were syringe-inoculated with bacterial suspensions using a needleless syringe. Syringe infiltration was performed on one or two corners of each leaf, and bacterial suspensions were spread throughout the entire leaf. We chose four different time points (4, 6, 9 and 24 h), representing early stages of infection and when dynamic transcriptional reprogramming was observed in a previous study that used bulk RNA-seq[40]. For each strain, four time points were sampled at the same time of day by infiltrating bacteria at different times to minimize the influence of circadian rhythms. The 20 infected leaves were collected using forceps and immediately processed for extraction of nuclei. For the mock condition, water-infiltrated leaves were collected after 9 h.

### Generation of transgenic plants and pathogen-infection assays

pWAT206 was a gift from A. Takeda. The following plasmids were constructed using HiFi DNA assembly kits (New England Biolabs) and were verified by Sanger sequencing. Three PCR fragments were amplified from pBICRMsG[41] using primer pairs (mEGFP-Nter_1_F plus mEGFP_1_R; mEGFP_2_F plus mEGFP_2_R; and mEGFP_3_F plus mEGFPNter_3_R) and were then assembled into StuI/AscI-digested pBICAscII[42]. The resulting plasmid was used as a template for two PCR reactions using primer pairs mEGFP_Nter_1_F plus mEGFP_4R and mEGFP_4F plus mEGFP_Nter_3R. These PCR products were assembled into StuI/AscI-digested pBICAscII, which produced pBIC_mEGFP_Nter. A PCR fragment was amplified from pBIC_mEGFP_Nter using mEGFP_Cter_F1 plus mEGFP_Cter_R, and then used as a template for PCR using mEGFP_Cter_F2 plus mEGFP_Cter_R. The resulting PCR fragment was assembled into StuI/AscI-digested pBIC_mEGFP_Nter to obtain pBIC_mEGFP_Cter. The coding sequence of *GT-3A* (*At5g01380*) was amplified by PCR using the primer pair GT3a_mEGFP_F plus GT3a_mEGFP_R and was assembled into StuI-digested pBIC_mEGFP_Cter to obtain pBIC_GT3a_mEGFP. The sequence encoding carboxy-terminally mEGFP-fused GT-3A was amplified from pBIC_GT3a_mEGFP by PCR using a primer pair (35Sp_HiFi_F and 35Ster_HiFi_R) and was then assembled into XhoI/XbaI-digested pWAT206 to obtain pWAT206_GT3a_mEGFP. *A. thaliana* Col-0 plants were transformed using *Agrobacterium tumefaciens* GV3101 (pMP90, pSoup) with pWAT206_GT3a_mEGFP as previously described[43]. Two independent Basta-resistant transgenic lines were used for bacterial growth assays with a bioluminescent *P. syringae* pv. *tomato* DC300043 strain and for lesion development analysis using *C. higginsianum* MAFF 305635. Plants were grown in a climatic chamber with a temperature of 22 °C, 60% relative humidity and light intensity of 6,000 lux (about 100 μmol m$^{-2}$ s$^{-1}$) for 10 h. Bacterial suspensions at $OD_{600} = 0.001$ in sterile water were syringe-infiltrated into leaves of 4–5-week-old plants. Bacterial growth was measured as bioluminescence using a GloMax Navigator Microplate luminometer (Promega) as previously described[44]. For lesion development analysis, leaves of 4–5-week-old plants were drop-inoculated with a 5 μl conidial suspension of *C. higginsianum* ($1 \times 10^5$ conidia per ml). The inoculated plants were kept under high humidity in a climatic chamber with a temperature of 22 °C and light intensity of 6,000 lux for 10 h. Lesion size was measured at 6 days after inoculation using ImageJ.

### Generation of *gt3a*-KO plants

The *gt3a*-KO plants were created by genome editing with CRISPR–*Cpf1*. The plasmid pWAT235-Cas9-HF-cc was provided by A. Takeda. The *Cpf1* sequence was amplified from pY004 (Addgene, 69976) by PCR using a primer pair (SpeI-NLS-Cpf1-F and BamHI-NLS-Cpf1-R; Supplementary Table 2), followed by digestion with SpeI and BamHI. The digested DNA fragment was ligated to pWAT235-Cas9-HF-cc, which had been digested with SpeI and BamHI, resulting in pWAT235-Cpf1-cc. Two PCR fragments were amplified from pWAT235-Cpf1-cc using primer pairs Gb-ccdB-F and Gb-ccdB-R, and Gb-U626-F and Gb-U626-R. These PCR fragments and BsaI-digested pWAT235-Cas9-HF-cc were assembled into a single plasmid using a HiFi DNA assembly kit, which resulted in pWAT235-Cpf1-U626-cc-polyT. Two oligonucleotides, GT3a_gRNA2_F and GT3a_gRNA2_R, were hybridized and ligated to BsaI-digested pWAT235-Cpf1-U626-cc-polyT, which resulted in pWAT235-Cpf1-U626-GT3a-gRNA2-polyT. *A. thaliana* Col-0 plants were transformed using *A. tumefaciens* GV3101 (pMP90, pSoup) with pWAT235-Cpf1-U626-cc-polyT as previously described[43]. A transgenic line with a premature stop codon due to a 5-bp deletion in the *GT-3A* gene was selected and used throughout this study.

### Bulk RNA-seq of plants

Leaves of wild-type plants and one of the transgenic lines expressing GT-3A–mEGFP were syringe-infiltrated with water or DC3000 at an $OD_{600}$ of 0.001 and were collected 24 h after infiltration. Total RNA was extracted using TRI Reagent (Sigma). One microgram of total RNA was used for library preparation using the BrAD-seq method to create strand-specific 3′ digital gene-expression libraries[45]. The libraries were sequenced on a DNBSEQ-G400 platform at BGI, which produced 100-bp end reads. Because the quality of reverse reads was poor due to the poly(A) sequence, only forward reads were used for analysis. Trimming of the first 8 bases and adaptors and quality filtering were

performed using fastp (v.0.19.7)[46] with the parameters -x -f 8 -q 30 -b 50. The trimmed and quality-filtered reads were mapped to the *Arabidopsis* genome (TAIR10) using STAR (v.2.6.1b)[47] with default parameters and transformed to a count per gene per library using featureCounts (v.1.6.0)[48]. Statistical analysis of the RNA-seq data was performed in the R environment (v.4.1.3). Because the BrAD-seq method involves poly(A) enrichment, mitochondrial and chloroplast genes were excluded. Genes with mean read counts of fewer than ten per library were excluded from the analysis. The resulting count data were subjected to TMM normalization using the function calcNormFactors in the package edgeR, followed by log transformation by the function voomWithQualityWeights in the package limma. To each gene, a linear model was fit using the function lmFit in the limma package. For variance shrinkage in the calculation of *P* values, the eBayes function in the limma package was used. The resulting *P* values were then corrected for multiple hypothesis testing by calculating Storey's *q* values using the function qvalue in the package qvalue. To extract genes with significant changes in expression, cut-off values of *q* < 0.05 and |log$_2$-transformed fold change| > 1 were applied.

### Extraction of nuclei and single-nucleus sequencing

Fresh nucleus purification buffer (NPB; 15 mM Tris pH 7.5, 2 mM EDTA, 80 mM KCl, 20 mM NaCl, 0.5 mM spermidine, 0.2 mM spermine, 1:100 BSA and 1:100 protease inhibitor cocktail) was prepared before the experiment and chilled on ice. All the subsequent procedures were performed on ice or at 4 °C. Twenty leaves were chopped in 500–1,000 µl cold NPB with 1:500 Protector RNase IN with a razor blade on ice for 5 min to release nuclei and then incubated in 20 ml NPB. The crude extract of nuclei was sequentially filtered through 70-µm and 30-µm cell strainers (70 µm, Corning Falcon cell strainers, Corning, 08-771-2; 30 µm, MACS SmartStrainers, Milteni Biotec, 130-098-458). Triton-X and NP40 were added to the extract to a final concentration of 0.1% each, and the extract was incubated at 4 °C for 5 min with rotation. The suspension was centrifuged at 50*g* for 3 min in a swing-rotor centrifuge to pellet non-nucleus debris and the supernatant was recovered. The nuclei were pelleted by centrifugation at 500*g* for 5 min in a swing-rotor centrifuge. When the pellet was green, the pellet was resuspended in 20 ml NPBd (NPB with 0.1% Triton-X and 0.1% NP40) with 1:1,000 Protector RNase IN by pipetting, followed by centrifugation at 500*g* for 5 min. When the pellet was translucent, the NPBd wash was skipped. The pellet was then washed by resuspending it in 20 ml NPB with 1:1,000 Protector RNase IN and centrifuging at 500*g* for 5 min in a swing-rotor centrifuge. The pellet was resuspended in 950 µl of 1× Nuclei Buffer (10x Genomics, PN-2000207) with 1:40 Protector RNase IN and 1 mM dithiothreitol. The suspension of nuclei was centrifuged at 50*g* for 3 min in a swing-rotor centrifuge to pellet non-nucleus debris and the supernatant was recovered. This step was repeated one more time. The resulting nuclei were manually counted using a haemocytometer. Nuclei were pelleted by centrifugation at 500*g* for 5 min in a swing-rotor centrifuge and the supernatant was removed, leaving approximately 10 µl of the buffer. Nuclei were counted again, and up to 16,000 nuclei were used for subsequent steps. However, in most samples, we did not load the maximum number of nuclei that the 10x Genomics kit accepts to avoid the risk of clogging the instrument, which can result in variable numbers of recovered nuclei among samples (Extended Data Fig. 1b). scRNA-seq and ATAC–seq libraries were constructed according to the manufacturer's instructions (10x Genomics, CG000338). scRNA-seq libraries were sequenced using an Illumina NovaSeq 6000 in dual-index mode with ten cycles for i7 and i5 indices. snATAC–seq libraries were also sequenced using an Illumina NovaSeq 6000 in dual-index mode with 8 and 24 cycles for the i7 index and the i5 index, respectively.

### Single-cell multiomic analysis

**Raw data processing.** Sequence data were processed to obtain single-cell feature counts by running cellranger (v.6.0.1) and cellranger-arc

(v.2.0.0) for snRNA-seq data and snATAC–seq data, respectively. For snRNA-seq, the –include-introns option was used to align reads to the *A. thaliana* nuclear transcriptome built using the TAIR10 genome and the Araport 11 transcriptome. The chloroplast genome was removed from the reference genome for the analysis of both snRNA-seq and snATAC–seq data. The *A. thaliana* TAIR10 genome was downloaded from https://plants.ensembl.org/, and the chloroplast genome was manually removed. The *A. thaliana* TAIR10 gene annotation file was manually modified by removing chloroplast genes and replacing semicolons in the 'gene_name' column with hyphens to prevent errors during cellranger-arc processing. We note that a mean of 23.4% and 26.0% of reads were mapped on the chloroplast genome in snRNA-seq data and snATAC–seq data, respectively. Removing the chloroplast genome from the reference did not affect the overall RNA and ATAC count distribution. Count data were analysed using the R packages Seurat (v.5)[49] and Signac[50].

**Quality control and cell filtering.** Before integrating the datasets, doublets were predicted using DoubletFinder[51] and filtered out. Quality control matrices for snATAC–seq were generated using a modified version of the loadBEDandGenomeData function in the R package Socrates[52]. ACRs were identified using MACS2 (ref. 53) with the following parameters: -g (genomesize)=0.8e8, shift=−50, extsize=100, and --qvalue=0.05, --nomodel, --keep-dup all. The fraction of reads mapping to within 2 kb upstream or 1 kb downstream of the transcription start site (TSS) was calculated. Nuclei were filtered using the following criteria: 200 < RNA UMI count < 7,000; RNA gene count > 180; 200 < ATAC UMI count < 20,000; and fraction of RNA reads mapped to mitochondrial genome < 10%. Seurat objects of individual samples were merged using the Merge_Seurat_List function of the scCustomize package.

**snRNA-seq clustering.** snRNA-seq clustering was performed using the R package Seurat. The cell-by-gene RNA count matrix was normalized using SCTransform. Dimension reduction was performed using principal component analysis with RunPCA. Technical variance among samples was reduced using Harmony[54] using principal components (PCs) 1–20. Graph-based clustering was performed on the Harmony-corrected PCs 1–20 by first computing a shared nearest-neighbour graph using the PC low-dimensional space (with *k* = 20 neighbours). Louvain clustering (resolution = 1.0) was then applied, and the clusters were projected into an additionally reduced space using UMAP (n.neighbours=20 and min.dist=0.01).

**snATAC–seq peak calling.** Peaks were called independently on each cluster defined by snRNA-seq data and then combined using the CallPeaks function of Signac, which uses MACS2 with the following parameters: effective.genome.size=1.35e8, extsize=15, shift=−75. Peak counts were quantified using the FeatureMatrix function. Compared with the default peak calling pipeline of cellranger-arc (24,394 peaks), this cluster-specific peak calling approach was able to capture more peaks (35,560 peaks), which is consistent with a previous report[50].

**snATAC–seq clustering.** Dimensionality reduction was performed using latent semantic indexing (LSI)[55]. First, the top 95% most common features were selected using the FindTopFeatures function. Then, the term-frequency inverse-document-frequency (TF-IDF) was computed using RunTFIDF with scale.factor=100,000. The resulting TF-IDF matrix was decomposed with singular value decomposition with RunSVD, which uses the irlba R package. Technical variance among samples was reduced using Harmony with LSI components 2–10. Graph-based clustering was performed on the Harmony-corrected LSI components 2–20 by first computing a shared nearest-neighbour graph using the LSI low-dimensional space (with *k* = 20 neighbours). Louvain clustering (resolution = 0.8) was then applied, and the clusters were

projected into an additionally reduced space with UMAP (n.neighbours= 30L and min.dist=0.01).

**RNA–ATAC joint clustering.** The two modalities were integrated by weighted nearest-neighbour analysis using FindMultiModal-Neighbors of Seurat with Harmony-corrected PCs 1–20 for RNA and Harmony-corrected LSI components 2–20 for ATAC. Then, SLM (resolution = 0.5) was applied and projected with UMAP (n.neighbours=30L and min.dist=0.1).

**ATAC gene activity score.** ATAC gene activity score was calculated using the GeneActivity function of Signac with extend.upstream=400.

**Peak-to-gene linkage analysis.** LinkPeaks of Signac was used to call significant peak-to-gene linkage for each infection condition (mock, DC3000, AvrRpt2 and AvrRpm1). Background-corrected Pearson's correlation coefficients between the gene expression of each gene and the accessibility of each peak within 500 kb of the gene TSS were calculated. A $P$ value was calculated for each peak–gene pair using a one-sided $z$-test, and peak–gene pairs with $P < 0.05$ and a Pearson's correlation coefficient of >0.05 were retained as significant links.

**Motif enrichment analysis.** Motifs present in the JASPAR2020 database[56] for *Arabidopsis* (species code 3702) were used. Cluster-specific peaks were first identified using FindMarkers of Seurat with default parameters. A hypergeometric test was used to test for the over-representation of motifs in the set of differentially accessible peaks using FindMotifs of Signac. Motif plots were generated using MotifPlot. Motif enrichment scores (motif deviation scores) of individual TF motifs in individual cells were calculated using chromVAR[17]. For the integration of motif enrichment scores and mRNA expression, motif names in JASPAR2020 were matched with gene names. Motifs that could not be uniquely associated with genes were removed from analysis.

**TF target prediction.** To predict genes that are regulated by a TF, ACRs containing each TF motif were extracted. Then, genes for which expression correlated with these ACRs were identified. We considered genes within 5 kb from an ACR with a linkage score of >0.1 as significant candidates. GO enrichment analysis was performed for these candidates for each TF using the enrichGO function of clusterProfiler with org.At.tair. db annotation. For the analysis in Fig. 2h, a more stringent threshold (linkage score > 0.2) was applied. GO enrichment plots were created using ggplot2 (ref. 57).

**Subcluster analysis.** Nuclei with the same subcluster label were aggregated, and log$_2$-transformed transcripts per million (TPM) values were calculated. Subclusters with more than 18,000 undetected genes were removed from the analysis. For the subclusters that passed the filtering step, genes were clustered using $k$-mean clustering with $k = 12$ (determined using the elbow method) (Extended Data Fig. 3b). Then, GO enrichment analysis was performed for the genes in each cluster (Extended Data Fig. 3b). Three clusters (1, 5 and 8) showed enrichment of an immunity-related function (responses to SA); genes in these clusters were defined as 'putative immune genes' and used for downstream analysis.

**Pseudotime analysis.** To calculate pseudotime, the cell-by-gene matrix for cells annotated as mesophyll was obtained from the snMultiome data. We used the function scanpy.tl.dpt with n_dcs=2. Pseudotime trajectories were constructed with each mesophyll cell in mock samples being used as a starting cell. Then, these individual trajectories were averaged in each cell to create a single, unified pseudotime trajectory. Heat map gene trends for a given gene were calculated by fitting a linear GAM using the pygam function LinearGAM with s (0, lam=400) to fit a GAM to RNA expression levels across sorted pseudotime values.

## Comparisons between PRIMER and bystander cells

Among the subclusters of immune-active mesophyll cells in the snMultiome data, PRIMER and bystander cell clusters were defined on the basis of expression of the marker genes *BON3* and *ALD1*, respectively. By comparing these cell states, DEGs were identified using the FindMarkers function of Seurat. Motif enrichment analysis was performed on ACRs within 2 kb of the cell-state-enriched DEGs.

To assess the contribution of CAMTA3 in different cell states, genes regulated by CAMTA3 were first identified using a published bulk RNA-seq dataset[39], comparing wild-type and camta3-D (a dominant negative mutant of *CAMTA3*). Genes suppressed by CAMTA3 after ETI (triggered by AvrRpm1 or AvrRps4 infection) were overlapped with the cell-state DEGs defined above. Hypergeometric tests were performed to assess the significance of the overlaps.

## snRNA-seq analysis of *gt3a*-KO plants

snRNA-seq was performed on *gt3a*-KO plants infected by AvrRpt2 or treated with water (mock) at 9 and 24 h.p.i. Two independent replicates were prepared for each infection condition. The data were filtered using the same criteria as for the snMultiome analysis without considering ATAC–seq information. The *gt3a*-KO snRNA-seq data were integrated with the Col-0 snMultiome data (mock and AvrRpt2 conditions), and de novo clustering was performed in the same way as described above. On the basis of immune-gene expression, immune-active mesophyll clusters were identified and further subclustered. From these subclusters, PRIMER and bystander cells were defined on the basis of expression of the marker genes *BON3* and *ALD1*, respectively. For each cell state, DEG analysis was performed comparing Col-0 and *gt3a*-KO plants to assess the effect of the *GT-3A* mutation.

## Comparisons between single-cell and bulk omics datasets

Our snATAC–seq data were compared with published bulk ATAC–seq data of mature *A. thaliana* leaves activating pattern-triggered immunity (PTI), ETI or both PTI and ETI, as well as non-immune-active leaves[12]. All ACRs identified in the bulk ATAC–seq datasets were combined and compared with the ACRs identified in our snATAC–seq datasets.

## MERFISH

**MERFISH panel design.** We curated 500 target genes that included the following genes: (1) previously defined markers of *A. thaliana* leaf cell types[58]; (2) genes involved in various processes such as immunity, hormone pathways and epigenetic regulation; and (3) a variety of TFs previously analysed using DAP-seq (a TF–DNA interaction assay)[59]. Genes for which more than 25 specific probes could not be designed based on probe design software from Vizgen were excluded from the target gene panel. Highly expressed genes could cause the overcrowding of smFISH signals and hinder MERFISH quantification. To avoid including highly expressed genes in the panel, we assessed target gene expression by using a publicly available bulk RNA-seq dataset of *A. thaliana* infected by AvrRpt2 in the same setup as the current study at eight different time points (1, 3, 4, 6, 9, 12, 16 and 24 h)[40]. For each gene, the highest expression value among the eight time points was used. Genes that showed TPM values of >710 were not included in the panel. The total TPM of the 500 genes was approximately 22,000. *ICS1* was targeted with a single round of smFISH as this gene is highly expressed. The smFISH result was provided as an image without quantitative information. Bacterial cells were visualized by targeting 19 highly expressed genes (based on previously published in planta bulk RNA-seq data[60]) as a single target. Supplementary Table 1 has a list of the genes targeted by MERFISH. All the probes were designed and constructed by Vizgen.

**Tissue sectioning, fixation and mounting.** Plants were grown according to the methods described above. Leaves matching the aforementioned treatments and time points were excised and immediately

incubated and acclimated in OCT (Fisher) for 5 min. Following incubation, the leaves were immediately frozen as previously described[61]. Tissue blocks were acclimated to −18 °C in a pre-cooled cryostat chamber (Leica) for 1 h. Tissue blocks were trimmed until the tissue was reached, after which 10-µm sections were visually inspected until the region of interest was exposed. Sample mounting and preparation were performed according to the MERSCOPE user guide, but with slight modifications. In brief, a 10-µm section was melted and mounted onto a room-temperature MERSCOPE slide (Vizgen, 20400001), placed into a 60-mm Petri dish and re-frozen by incubation in the cryostat chamber for 5 min. Subsequent steps were performed with the mounted samples in the Petri dish. The samples were then baked at 37 °C for 5 min and were incubated in fixation buffer (1× PBS and 4% formaldehyde) for 15 min at room temperature. Samples were then washed with 1× PBS containing 1:500 RNase inhibitor (Protector RNase inhibitor, Millipore Sigma) for 5 min at room temperature in triplicate. Following the final PBS wash, samples were dehydrated by incubation in 70% ethanol at 4 °C overnight.

**MERFISH experiment.** Tissue sections were processed following Vizgen's protocol. After removing 70% ethanol, the sample was incubated in the sample prep wash buffer (PN20300001) for 1 min and then incubated in the formamide wash buffer (PN20300002) at 37 °C for 30 min. After removing the formamide wash buffer, the sample was incubated in the MERSCOPE Gene Panel mix at 37 °C for 42 h. After probe hybridization, the sample was washed twice with the formamide wash buffer at 47 °C for 30 min and once with the sample prep wash buffer at room temperature for 2 min. After the washing step, the sample was embedded in hydrogel by incubation in the gel embedding solution (gel embedding premix (PN20300004), 10% (w/v) ammonium persulfate solution and *N,N,N',N'*-tetramethylethylenediamine) at room temperature for 1.5 h. Then, the sample was cleared by first incubating it in digestion mix (digestion premix (PN 20300005) and 1:40 protector RNase inhibitor) at room temperature for 2 h, followed by incubation in the clearing solution (clearing premix (PN 20300003) and proteinase K) at 47 °C for 24 h and then at 37 °C for 24 h. The cleared sample was washed twice with the sample prep wash buffer and stained with DAPI and PolyT staining reagent at room temperature for 15 min. Samples were then washed with the formamide wash buffer at room temperature for 10 min and rinsed with the sample prep wash buffer. The sample was imaged using a MERSCOPE instrument, and detected transcripts were decoded on the MERSCOPE instrument using a codebook generated by Vizgen. Transcripts were visualized using Vizgen's Visualizer.

**MERFISH segmentation and processing.** Cell-boundary segmentation was performed for each MERSCOPE data output. DAPI-targeting and poly(A)-targeting probes demonstrated variable success in staining nuclei and cytoplasm, respectively, depending on the samples and tissue regions examined (Fig. 3b and Extended Data Fig. 7e show failed and successful segmentation, respectively). By contrast, dense transcript areas marked nucleus locations more robustly (Extended Data Fig. 7e). Therefore, a transcript-based segmentation method was used. For each sample, a two-dimensional Numpy array of zeroes was generated, which modelled the total pixel area imaged. The coordinates of identified RNA transcripts were changed from 0 to 1 in this array. Next, the array was blurred using the cv2.GaussianBlur function in OpenCV with ksize=(5,5). The resulting array was chunked into 2,000 × 2,000 pixel regions. These regions were loaded into Cellpose[62], a deep-learning-based segmentation tool, and a custom segmentation model was trained by manually segmenting nucleus objects across ten 2,000 × 2,000 pixel regions in the Cellpose GUI.

The custom model was then used to predict the segmentation boundaries in all the remaining regions, with the parameters diameter=22.92, flow_threshold=0.7 and cell probability threshold=−2. Next, the total number of cells in an experiment was calculated by summing the number of unique cells across all regions. This number was then used to initialize the --num-cells-init command in another segmentation tool, Baysor[63], which considers the joint likelihood of transcriptional composition and cell morphology to predict cell boundaries. Baysor was run using a downloaded Docker image and parameters -s 250, --n clusters 1, -i 1, --force-2d, min-molecules-per-gene=1, min-molecules-per-cell=50, scale=250, scale-std="25%", estimate-scale-from-centers=true, min-molecules-per-segment=15, new-component-weight=0.2, new-component-fraction=0.3.

To test the quality of our transcript-based segmentation method, we used an FOV with successful DAPI staining (which was rare in our samples) and performed DAPI-based watershed segmentation and transcript-based segmentation (Extended Data Fig. 7d). Results from these two segmentation strategies agreed with each other in general, with the transcript-based approach capturing transcripts in the cytoplasm in addition to those in the nucleus (Extended Data Fig. 7d–f). This result indicated that our segmentation approach can reliably capture cells.

After Baysor segmentation, a cell-by-gene matrix was created from the transcript cell assignments. Cells with fewer than 50 assigned transcripts were removed. Scanpy was used for post-processing of our MERFISH experiments. After loading the respective cell-by-gene matrix into an Anndata object for each experiment, we stored the spatial coordinates of each cell obtained from Baysor. The individual transcript counts in each cell were normalized by the total number of transcript counts per cell. The Anndata cell-by-gene matrix was then log-scaled.

**MERFISH–snMultiome data integration and label transfer.** To integrate the MERFISH experiments with each other, we used FindIntegrationAnchors followed by the IntegrateData function in Seurat (v.5). To integrate the MERFISH experiment data with our snMultiome data, we integrated the corresponding time points in each modality separately. For each time point in the infection data (mock, 4, 6, 9 and 24 h), we used gimVI from scvi-tools to project both the snMultiome and MERFISH cells into the same latent space using their RNA expression counts. gimVI was trained over 200 epochs with a size 10 latent space for each time point. To transfer continuous observations (pseudotime, RNA and ATAC gene activity counts, and motif enrichment) from the snMultiome data to the MERFISH data, we assigned the numerical average of the nearest 30 snMultiome cells in the gimVI latent space for each MERFISH cell. Similarly, to transfer categorical observations, including cell types and cluster labels, we assigned each MERFISH cell the most common label in the set of its 30 nearest snMultiome cells. To plot the joint embedding of both modalities (MERFISH and snMultiome) for a single time point, we ran UMAP on the gimVI latent space with n_neighbors=30 and min_dist=0.1. Note that in the spatial mapping of cell types predicted in snMultiome (Fig. 4c), although the section was in the middle of the leaf, some epidermal cells were included because the section was not completely flat.

**smFISH quantification and bacterial colony identification.** Quantification of transcripts labelled by smFISH was performed using the Python package Big-FISH. Seven *z* planes of MERSCOPE smFISH images were projected into two dimensions by using numpy.max along the *z* axis and chunked into 2,000 × 2,000 regions. Spots were then called using the function bigfish.detection.detect_spots with threshold=50, spot_radius=(10, 10) and voxel_size=(3, 3). To identify bacterial colonies, bigfish.detection.detect_spots was called to label 'bacterial meta gene' locations with threshold=200, log_kernel_size=(1.456, 1.456) and minimum_distance=(1.456, 1.456). To account for autofluorescence from the plant tissue, the same spot-caller was used to call spots on DAPI and *ICS1* channels. The spots from all three channels were aggregated, and DBSCAN from scikit-learn.cluster was used on all spots with eps=35 and min_samples=5. We kept all DBSCAN clusters for which the DAPI and *ICS1* spots constituted less than 30% of the total cluster spots.

These remaining clusters represented potential bacterial colonies. We manually evaluated each cluster to merge those marking the same colony and removed the clusters marking obvious autofluorescence, for example, signals from stomata. Windows of 300 × 300 pixels were generated around each final cluster, and detect_spots was used with threshold=95, spot_radius=(17, 17) and voxel_size=(3, 3) for accurate quantification of individual bacteria per cluster.

**Bacterial neighbourhood analysis.** To determine the level of immunity of the neighbourhood around each bacteria colony, we identified the 1–1,000 nearest cells in proximity to the colony. Then, the smoothed imputed pseudotime values of these neighbouring cells were averaged to obtain a single mean pseudotime value for each neighbour size. Error curves were calculated using the standard error divided by the mean at each neighbourhood size. These values serve as indicators of the overall immunity level of the area around each colony.

**Spatial differential expression analysis.** To identify genes that are spatially associated with immune-rich areas (related to Fig. 5a), we first smoothed our imputed spatial pseudotime over the spatially nearest 100 cells to find areas of immune-active hotspots. We then used RCTD[64] to deconvolve doublets in our spatial data using the cell types in our snMultiome data as a reference. We created a reference object with min_umi = 15 and ran RCTD on our MERFISH data with the parameters gene_cutoff = 0.0001, gene_cutoff_reg = 0.0001, fc_cutoff = −3, fc_cutoff_reg = −3 and doublet_mode = 'doublet'. We then ran C-SIDE[20] of spacexr with run.CSIDE.single with cell_type_threshold = 0 to find the top spatially differentially expressed genes per cell type along the smoothed spatial pseudotime (explanatory variable) and plotted the negative log-transformed $P$ value against the log-transformed fold change.

## Reporting summary

Further information on research design is available in the Nature Portfolio Reporting Summary linked to this article.

## Data availability

All information supporting the conclusions are provided with the paper. The single-cell and bulk sequencing data generated in this study have been deposited in the National Center for Biotechnology Information Gene Expression Omnibus database (accession numbers GSE226826 and GSE248054). Reference genomes, annotation and fully processed data for snMultiome analyses are available from the Salk website (http://neomorph.salk.edu/download/Nobori_etal_merfish). The MERFISH data are available from the Salk website (http://plantpathogenatlas.salk.edu). Genes targeted using MERFISH are listed in Supplementary Table 1. Information about primers used in this study is provided in Supplementary Table 2. Source data are provided with this paper.

## Code availability

The codes to analyse snMultiome and MERFISH data are available at GitHub (https://github.com/tnobori/snMultiome and https://github.com/amonell/Spatial_Plant_Pathogen_Atlas).

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

**Acknowledgements** We thank K. Tsuda and J. Walker for comments on the manuscript. T.N. was supported by a Human Frontiers Science Program (HFSP) Long-term Fellowship (LT000661/2020-L). J.R.E. is an Investigator of the Howard Hughes Medical Institute.

**Author contributions** T.N. conceived and designed the study and experiments with guidance from J.R.E. T.N. performed snMultiome and MERFISH experiments and analysed data. T.A.L. provided tissue sections for the MERFISH experiments. A. Monell performed MERFISH data analyses and data integration with assistance from J.Z. Y.S., S.S. and A. Mine produced GT-3A mutants and performed pathogen growth assays. J.R.N. assisted with sequencing and data management. T.N. wrote the initial draft of the manuscript. T.N. and J.R.E. edited the manuscript.

**Competing interests** J.R.E. serves on the scientific advisory board of Zymo Research and of Cibus. The other authors declare no competing interests.

**Additional information**
**Correspondence and requests for materials** should be addressed to Joseph R. Ecker.

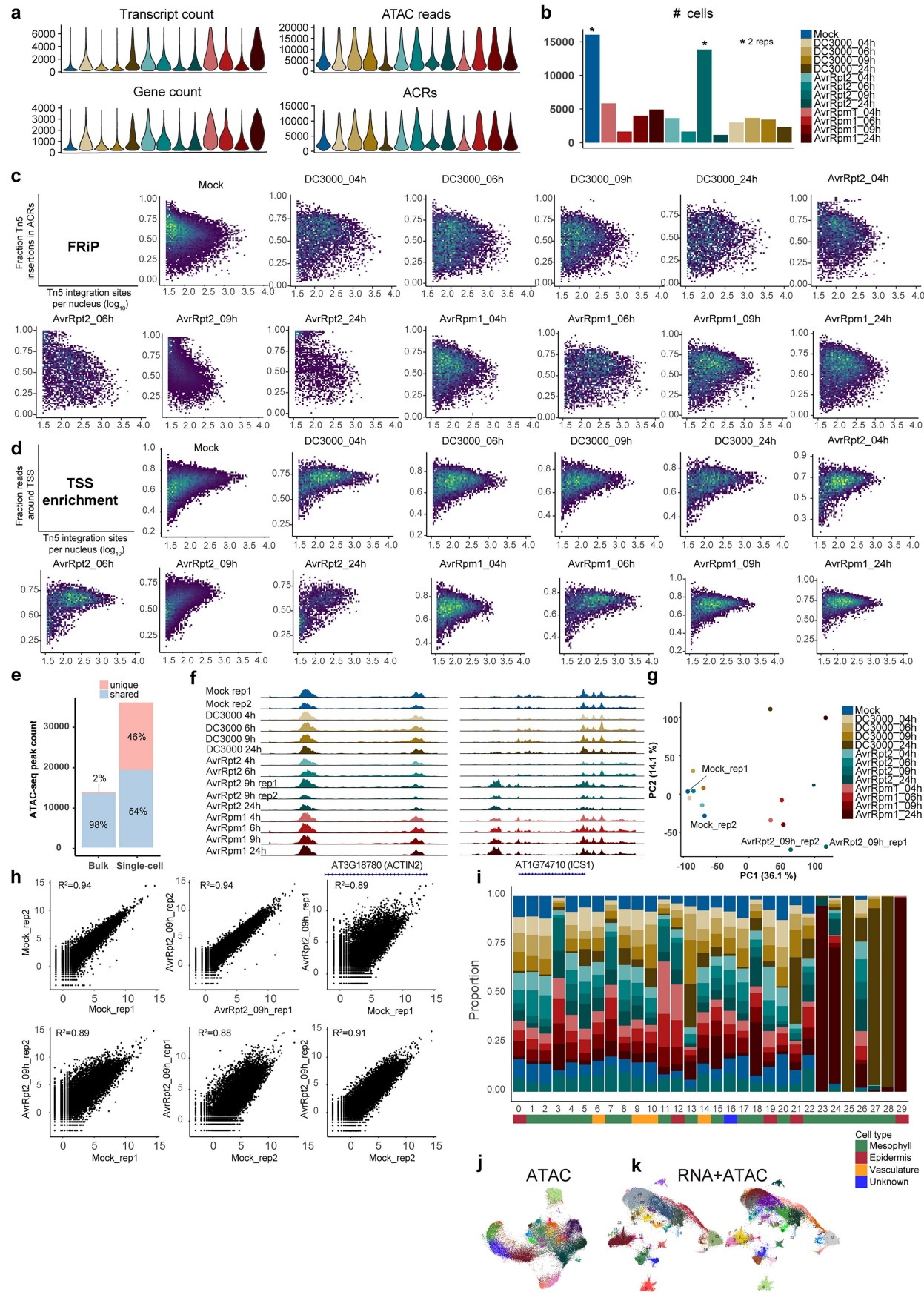

**Extended Data Fig. 1** | See next page for caption.

**Extended Data Fig. 1 | Quality control of snMultiome data. a**, Violin plots showing transcripts per nucleus, genes per nucleus, ATAC reads per nucleus, and accessible chromatin regions (ACRs) per nucleus in each sample. **b**, The number of cells per sample. Two independent replicates were analyzed and combined for Mock and AvrRpt2 9 hpi conditions. **c**, Density scatterplots showing fraction reads in peaks (FRiP) score in each sample before cell filtering. x-axis: $\log_{10}$ transformed read depths. y-axis: fraction of Tn5 integration sites in ACRs. **d**, Density scatterplots of $\log_{10}$ transformed read depths (x-axis) by the fraction of Tn5 integration sites mapping to within 2 kb upstream and 1 kb downstream of transcription start sites (TSSs) (y-axis). Data in each sample before cell filtering is shown. **c-d**, Two replicates of Mock and AvrRpt2 9 h conditions were combined. **e**, Bar plot showing the number of ATAC-seq peaks identified in previous bulk ATAC-seq data and the present snATAC-seq data. Shared and assay unique peaks are shown in blue and red, respectively.

**f**, Sample-aggregated chromatin accessibility around *ACTIN2* (left) and *ICS1* (right). **g**, Principle component analysis of pseudobulk transcriptome of each sample. Independent replicates of Mock and AvrRpt2 9 h were labeled. **h**, Scatter plots comparing pseudobulk transcriptomes of Mock and AvrRpt2 9 h samples. Pearson's correlation coefficient values were shown. **i**, Stacked bar plots showing the representation of gene expression-based Leiden clusters in each sample. **j**, Two-dimensional embedding of chromatin accessibility similarity among nuclei from all samples with uniform manifold approximation and projection (UMAP). Nuclei are colored by Leiden clusters. **k**, UMAP embeddings based on a joint neighbor graph that represents both gene expression and chromatin accessibility measurements. Nuclei are colored by de novo Leiden clusters based on the joint analysis (left) and Leiden clusters defined by gene expression measurement alone (right; Fig. 1b).

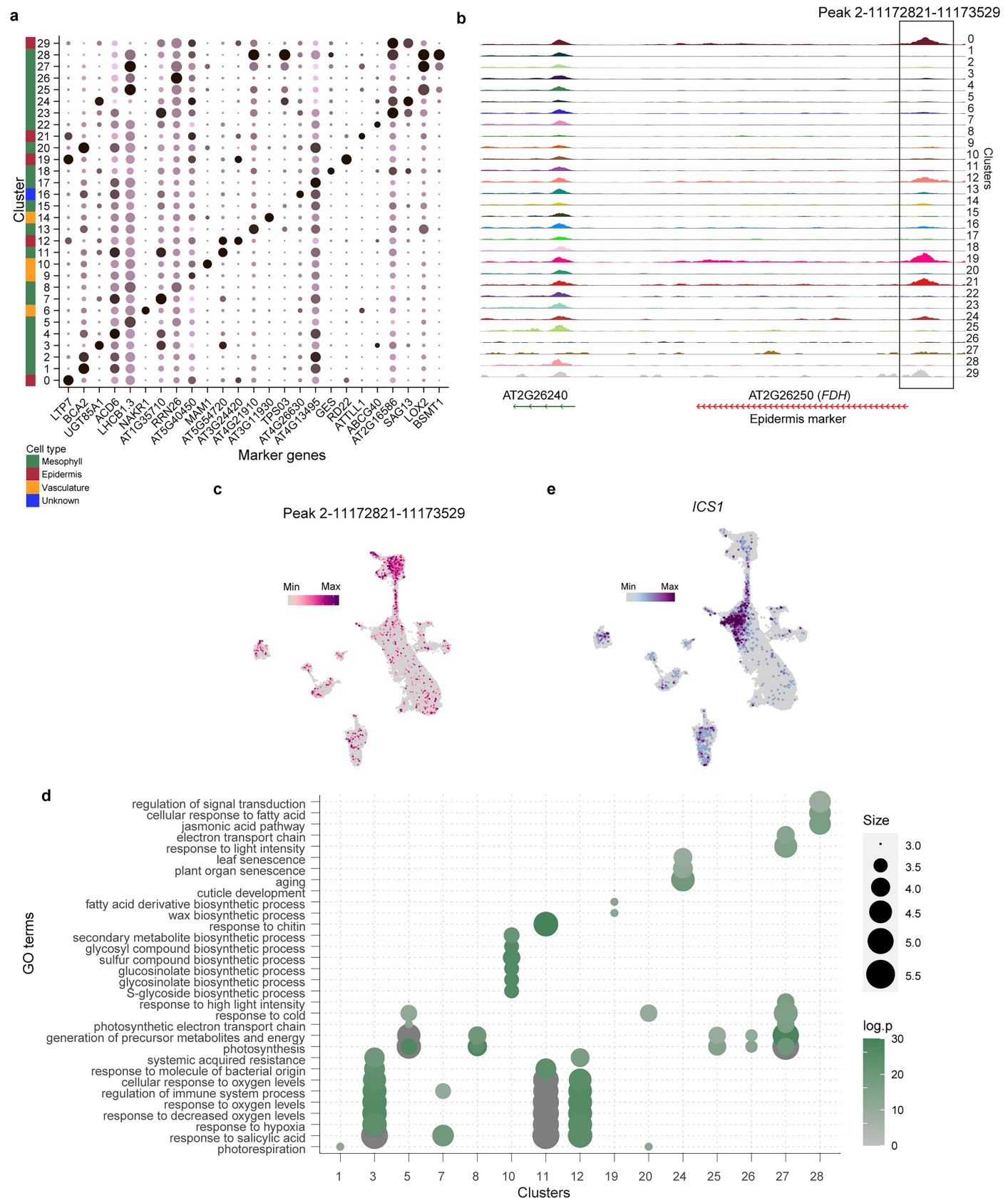

**Extended Data Fig. 2 | Single-cell analysis of gene expression and chromatin accessibility. a**, Dot plot showing the top marker genes of individual clusters. **b**, Cluster-aggregated chromatin accessibility surrounding *FDH*, a known marker gene for leaf epidermis. **c**, UMAP plot showing the ATAC-seq count on a peak near *FDH* (chromosome 2, position 11172821-11173529) in each nucleus. **d**, GO enrichment analysis for marker genes of each cluster. **e**, Expression of *ICS1* mRNA in each nucleus in each sample. Adjusted p-values from a one-sided hypergeometric test followed by Benjamini-Hochberg correction are shown.

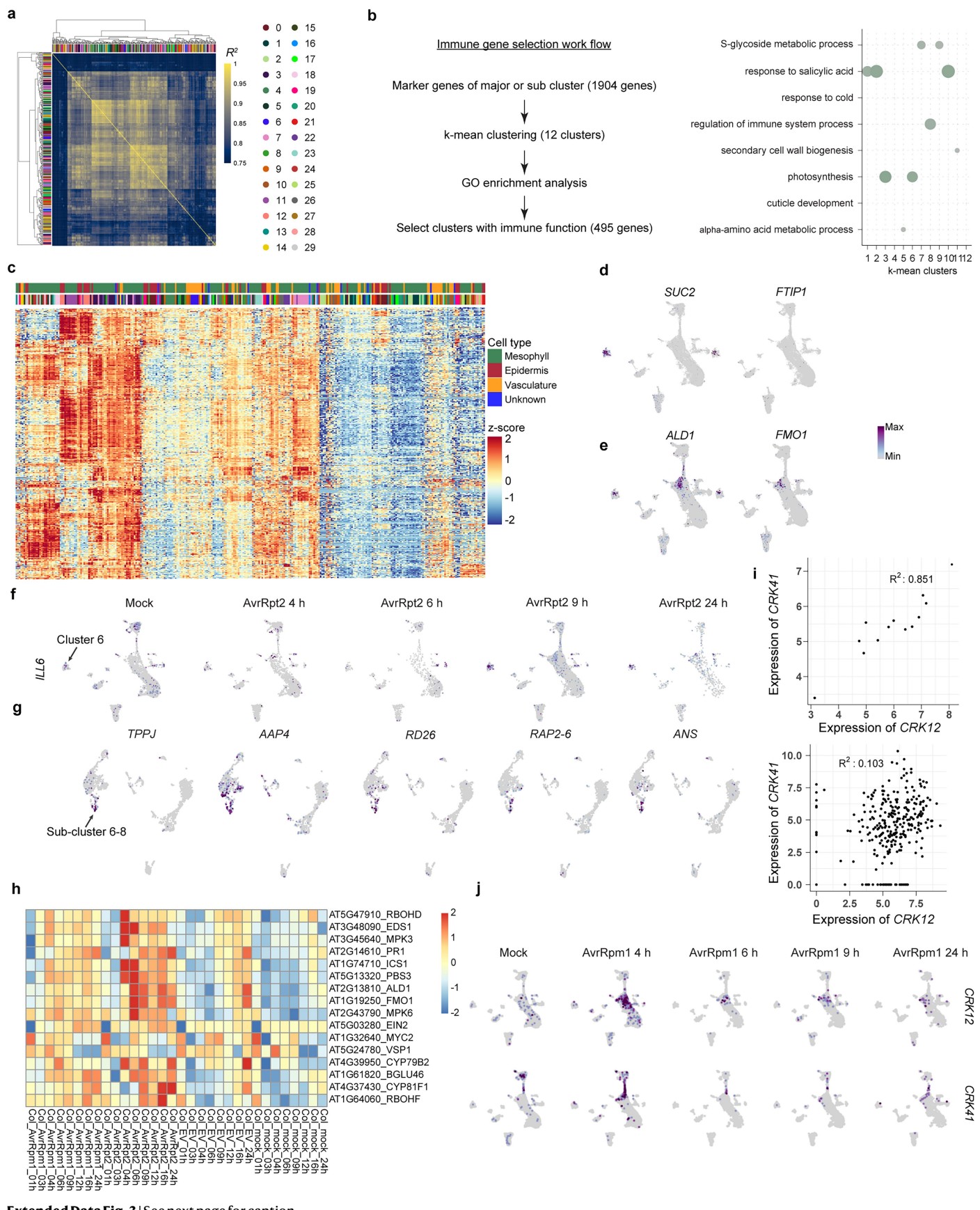

**Extended Data Fig. 3** | See next page for caption.

**Extended Data Fig. 3 | Comprehensive characterization of distinct cell populations. a**, Heatmap showing pair-wise correlation of pseudobulk transcriptomes between sub-clusters of individual clusters. The top and side bars show major cluster labels. **b**, (Left) Schematic workflow for the selection of highly variable immune-related genes. First, genes that are significantly enriched in at least one of major or sub clusters were selected. Then, these genes were clustered based on pseudobulk expression of sub-clusters using k-mean clustering (k value was determined by the elbow method), followed by GO enrichment analysis of each k-mean cluster. Finally, gene clusters with enriched immunity-related GO terms were selected. (Right) Top GO terms enriched in 12 k-mean clusters. Clusters 1, 2, and 10 were selected as immune genes. **c**, Heatmap showing normalized expression of immune-related genes selected in (b) across all the sub clusters. The top bars indicate cell type and major cluster that each sub-cluster derived from. **d**, Expression of the phloem companion cell markers *SUC2* and *FTIP1*. **e**, Expression of *ALD1* and *FMO1*. **f**, Expression of *ILL6* upon infection of AvrRpt2 in time course, showing specific induction in cluster 6. **g**, Expression of genes specifically expressed in sub-cluster 6-8. **h**, Heatmap showing expression of immune-related genes shown in Fig. 1d in a previous time-course bulk RNA-seq study, where *A. thaliana* leaves were infected by DC3000, AvrRpt2, or AvrRpm1 or treated with mock control (water). These data do not capture the presence of various cell populations identified in Fig. 1d. **i**, Scatter plots showing the correlation between *CRK12* and *CRK41* in the time course bulk RNA-seq (top) and single-nucleus RNA-seq (snRNA-seq; bottom). For snRNA-seq analysis, pseudobulk gene expression data of the subclusters (shown in **c**) were used. **j**, Expression of *CRK12* and *CRK41* upon infection with AvrRpm1 as shown in the UMAP. These apparently correlated genes in bulk RNA-seq showed a highly different expression pattern at the single-cell level.

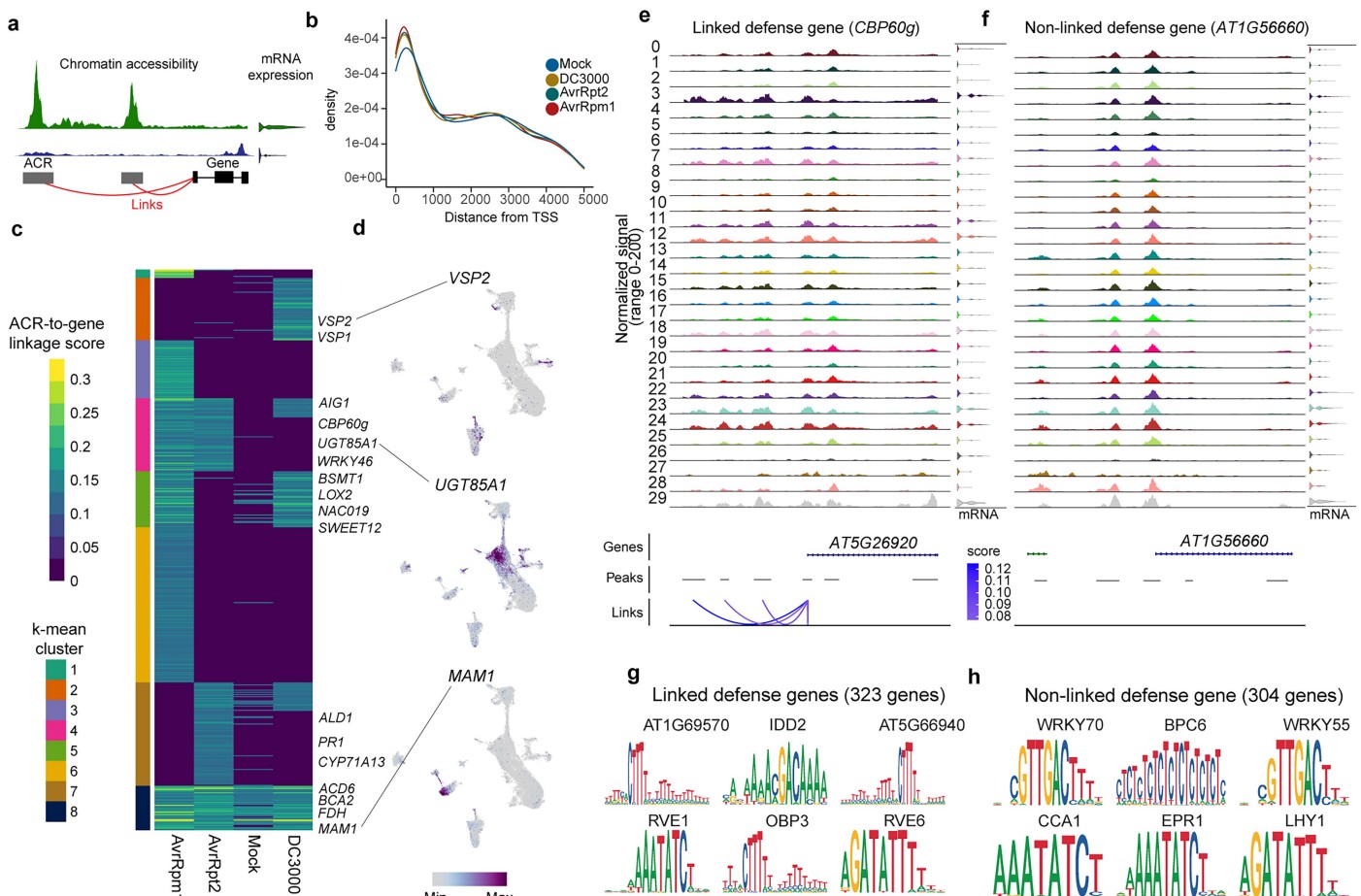

**Extended Data Fig. 4 | Linking gene expression and chromatin accessibility at the single-cell level. a**, Schematic diagram of linked accessible chromatin regions (ACRs) and a gene. An ACR and a gene are "linked" when there is a significant correlation between chromatin accessibility and mRNA expression across individual cells. **b**, Density plot showing the frequency of linkages at different distances from the transcription start sites (TSSs). **c**, Heatmap showing the linkage score (Pearson correlation coefficient between ACR count and mRNA expression) for genes that showed at least one significant link in at least one of the infection conditions or Mock. When a gene had multiple links, the maximum linkage score was shown. The sidebar shows the k-mean cluster

annotation (the k value was determined by the elbow method). **d**, Expression of mRNA encoding *VSP2*, *UGT85A1*, and *MAM1*. **e,f**, Cluster-aggregated chromatin accessibility surrounding *CBP60g* (e) and *AT1G56660* (f). Violin plots on the side show aggregated mRNA expression of each gene. Both genes were highly expressed in a cell-specific manner, but only *CPB60g* showed correlated (linked) chromatin accessibility patterns. **g,h**, Top motifs enriched in the promoter regions (2 kb upstream from the TSS) of defense genes (markers of immune-active mesophyll and epidermis clusters 3, 4, 7, 11, and 12) that are linked (g) and not linked (h).

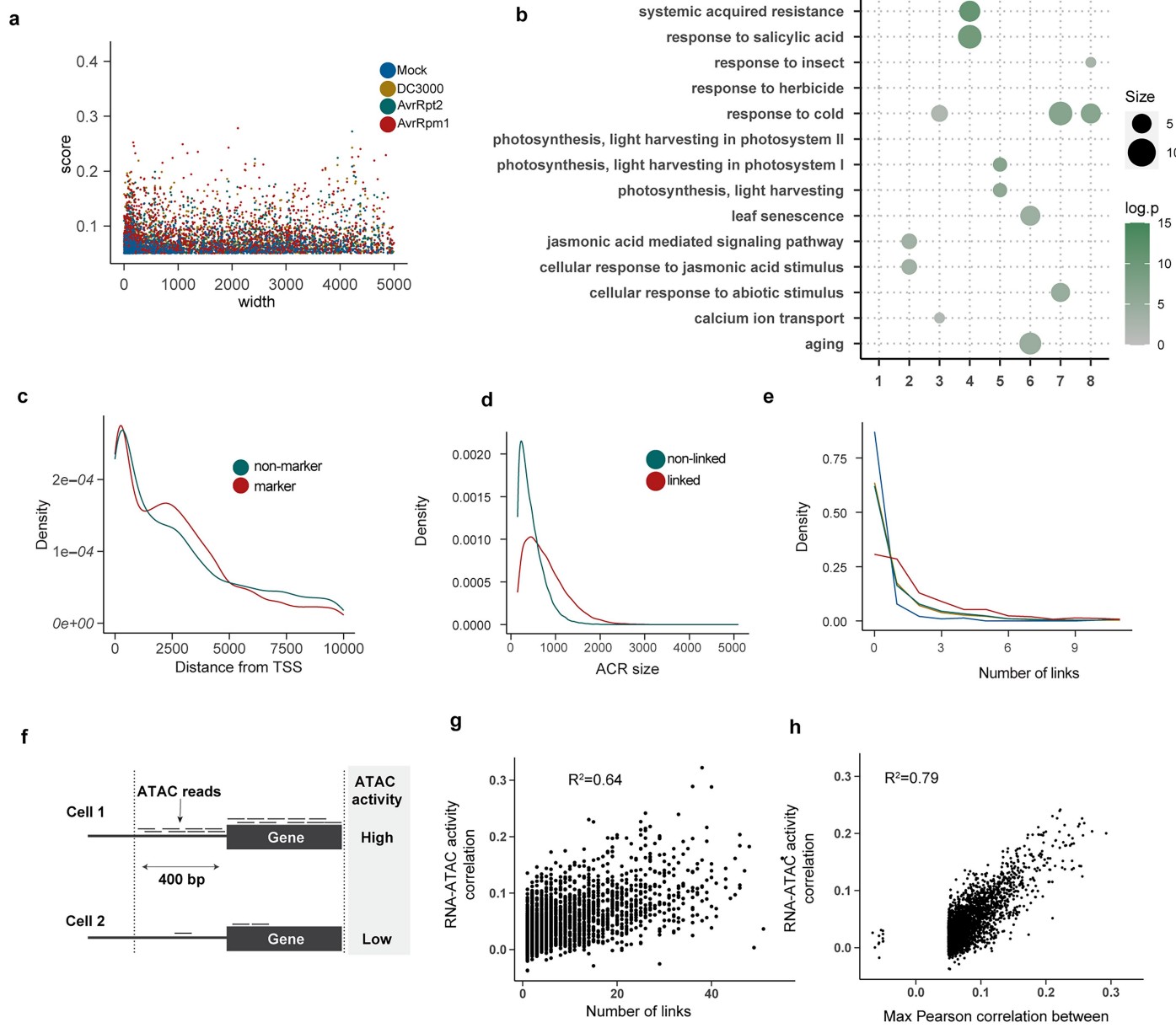

**Extended Data Fig. 5 | Links between transcriptome and chromatin accessibility. a**, Scatter plot showing the distribution of the linkage score (Pearson correlation coefficient between ACR count and mRNA expression) at different distances from TSSs. **b**, GO enrichment analysis of genes in each k-mean cluster shown in (Extended Data Fig. 4c). Adjusted p-values from a one-sided hypergeometric test followed by Benjamini-Hochberg correction are shown. **c**, Density plot showing the frequency of linkages at different distances from TSS for cluster marker genes and non-marker genes. **d**, Density plot showing the frequency of the size of ACRs for linked or non-linked genes.

**e**, Density plot showing the frequency of genes with different numbers of linked ACRs. **f**, Schematic diagram of the ATAC activity analysis. For each gene, ATAC reads mapped on the gene body or the 400 bp upstream region were aggregated to calculate the score. **g**, Scatter plot showing the relationship between the number of linkages (x-axis) and RNA-ATAC activity correlation score (y-axis). **h**, Scatter plot showing the relationship between the maximum correlation coefficient between each gene and linked ACRs (x-axis) and RNA-ATAC activity correlation score (y-axis).

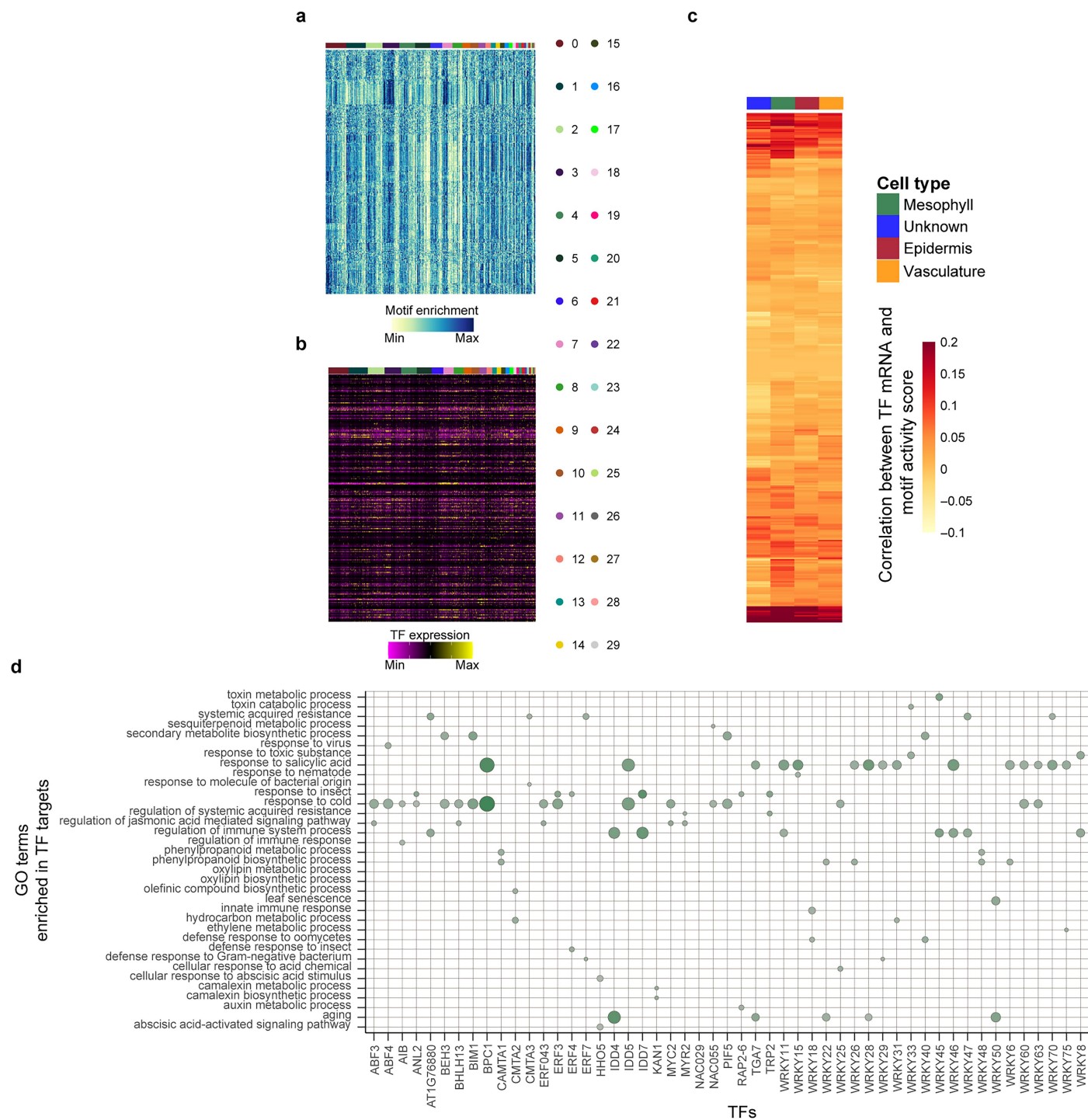

**Extended Data Fig. 6 | Identification of TF-gene modules. a,b** Heatmaps showing (a) enrichment scores of 465 motifs and (b) expression of corresponding transcription factors (TF) in each nucleus. The top bars show cluster annotation defined in Fig. 1b. **c**, Heatmap showing Pearson correlation coefficient between motif enrichment scores and mRNA expression of the corresponding TFs in each cell type (shown in the top bar). All TFs are shown. **d**, GO enrichment analysis of genes predicted to be targeted by TFs shown in Fig. 2h.

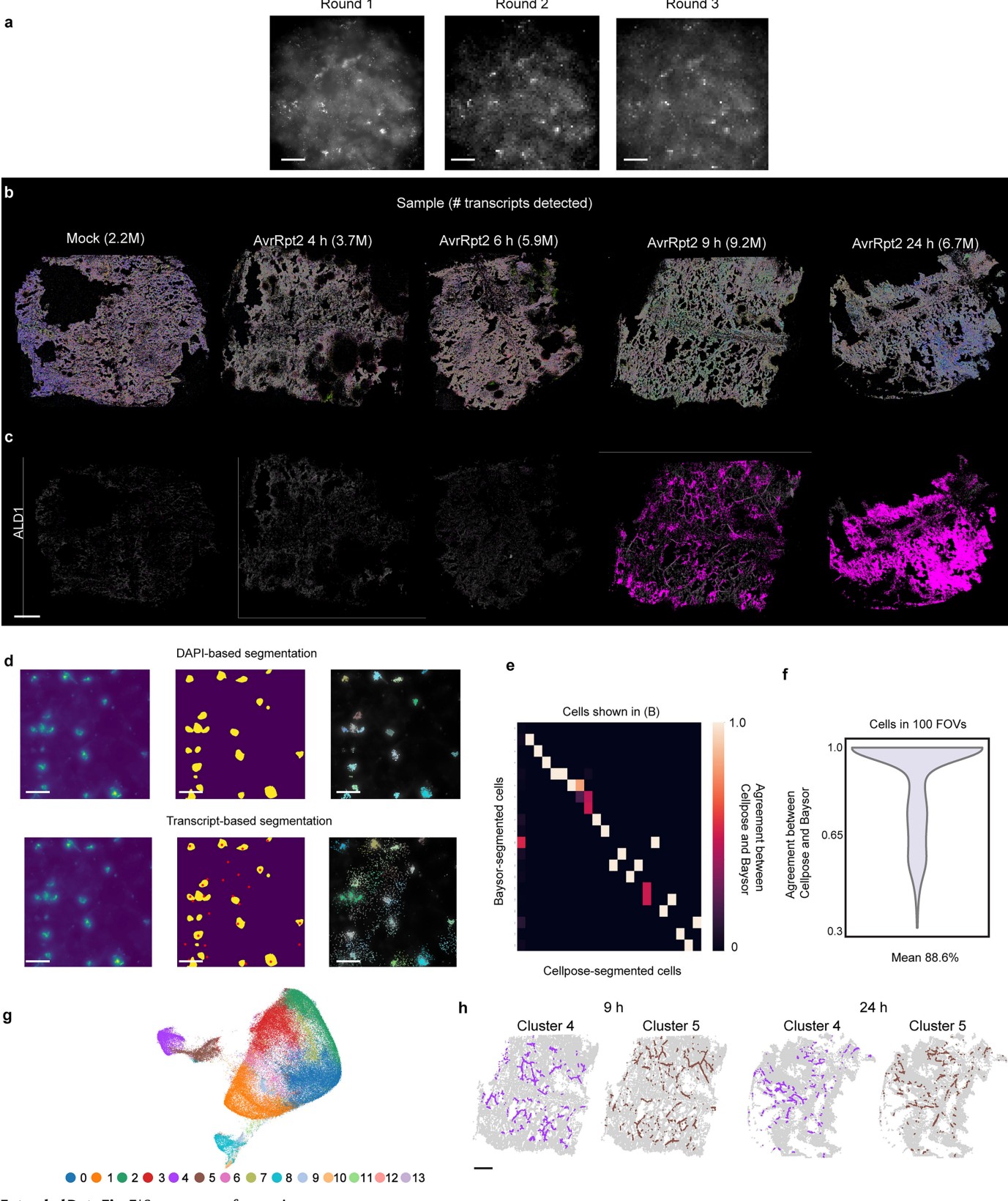

**Extended Data Fig. 7** | See next page for caption.

**Extended Data Fig. 7 | MERFISH data analysis. a**, Example raw images of MERFISH that capture the same region of tissue across three imaging rounds. White spots are signals derived from single mRNA molecules. We observed similar results in all independent samples. **b**, Two-dimensional plots of all the transcripts detected by MERFISH in each sample. The number of transcripts is shown. **c**, Spatial expression pattern of *ALD1* detected by MERFISH in each sample. **d**, (Left) A field of view (FOV) that shows obscure DAPI nuclei staining signal. (Middle) Transcript-based segmentation in the same FOV (see Method). Transcripts were colored by assigned cells. (Right) Centroids of cells detected by the transcript-based segmentation (red dots) and the result of failed DAPI-staining-based segmentation (yellow region). Similar patterns were observed across FOVs and samples. A systematic quantitative analysis is provided in (f). (a,d) Scale bars = 40 μm. **e, f**, The fraction of transcripts in Cellpose-segmented cells fell within Baysor-segmented cells. (e) Cells detected in the field of view (FOV) shown in (d) were used. (f) Cells detected in 100 FOVs were used. **g**, Integrated UMAP of all MERFISH data. **h**, Spatial mapping of de novo MERFISH clusters annotated as vasculature. b,c,h Scale bars = 1 mm.

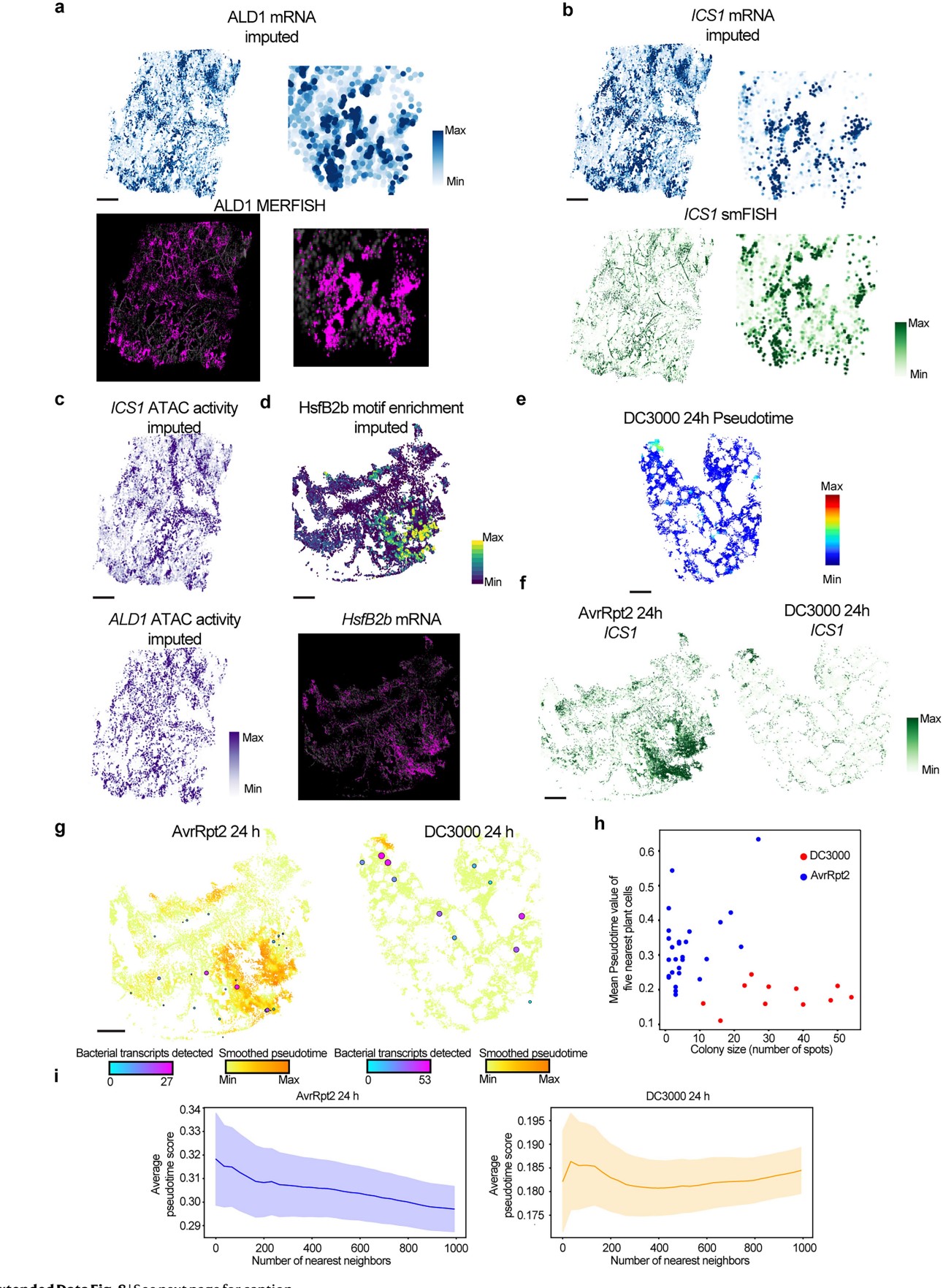

**Extended Data Fig. 8** | See next page for caption.

**Extended Data Fig. 8 | Spatial mapping of whole transcriptome and epigenome. a**, Imputed mRNA expression of *ALD1* (Top) predicted its mRNA expression pattern measured with MERFISH (Bottom). Magnified images are shown on the right. **b**, Imputed mRNA expression of *ICS1* (Top) predicted its mRNA expression pattern measured with smFISH (Bottom). Magnified images are shown on the right. **c**, Imputed ATAC activity (Extended Data Fig. 5f) of *ICS1* (top) and *ALD1* (bottom), which showed consistent patterns with mRNA expression (a,b). **d**, Imputed motif enrichment scores of HsfB2b (Top) was consistent with mRNA expression of *HsfB2b* confirmed by MERFISH (Bottom). **e**, Spatial mapping of pseudotime values based on data integration and label transfer between snRNA-seq and MERFISH data of DC3000-infected plants.

**f**, smFISH of *ICS1* in leaves infected by AvrRpt2 or DC3000 at 24 hpi. **g**, Spatial mapping of bacterial transcripts detected with smFISH in plants infected by AvrRpt2 (left) and DC3000 (right) at 24 h post-infection (hpi). Pseudotime values are also visualized in the background. Dot size reflects the number of bacterial transcripts detected. a-g, Scale bars = 1 mm. **h**, Scatter plot showing the number of bacterial transcripts (x-axis) and averaged pseudotime values of five nearest neighbor plant cells (y-axis) for each bacterial colony in plants infected by AvrRpt2 (blue) and DC3000 (red) at 24 hpi. **i**, Averaged pseudotime scores of cells surrounding each bacterial colony. The x-axis shows the number of nearest plant cells analyzed for each bacterial colony. Shaded error bands indicate standard error of the mean.

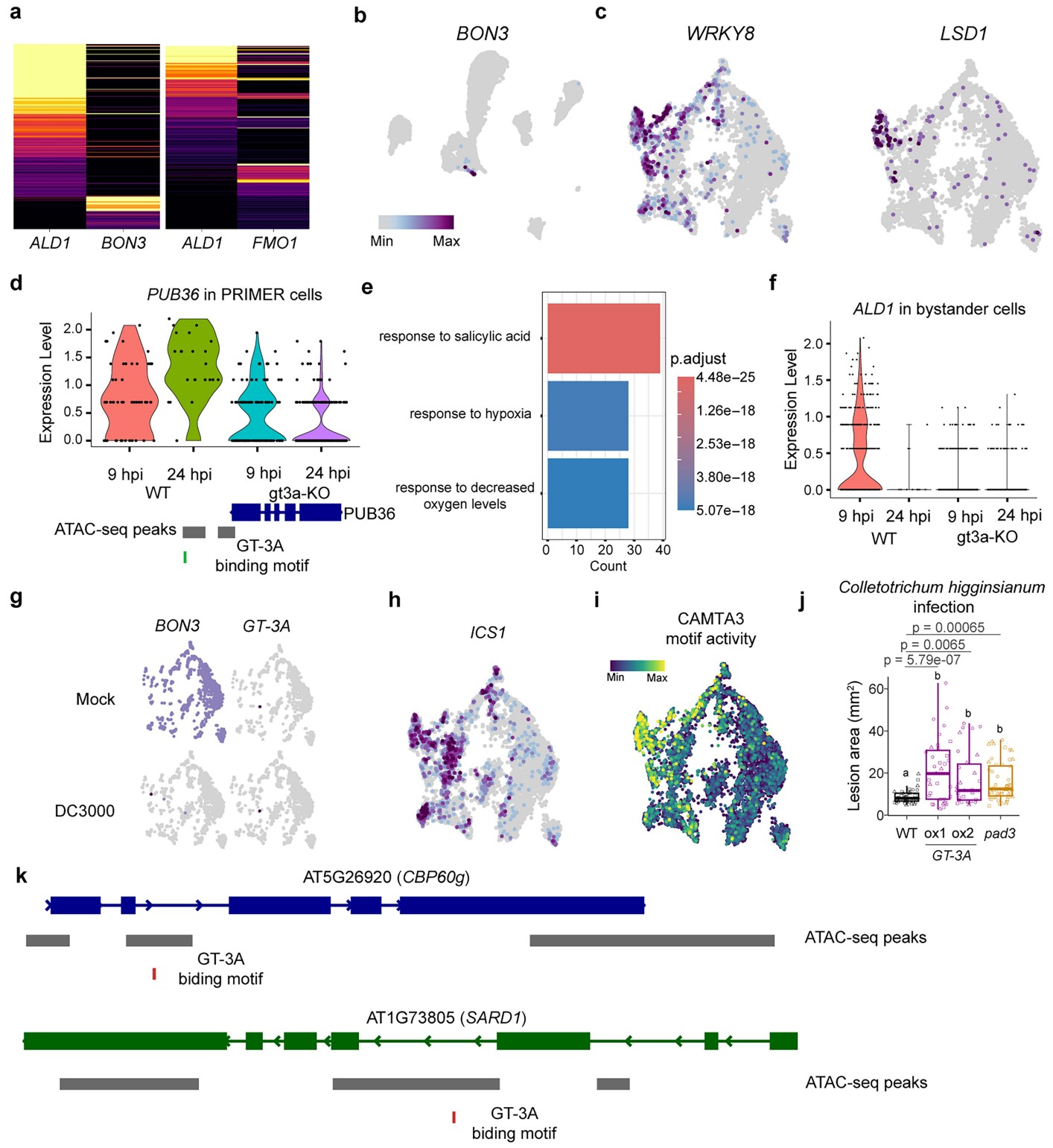

**Extended Data Fig. 9** | See next page for caption.

**Extended Data Fig. 9 | Characterization of PRIMER and bystander cells.**
**a**, Heatmaps showing expression of *ALD1*, *FMO1*, and *BON3* in each cell at 9 hpi.
**b**, Expression of *BON3* shown in the UMAP in Figs. 4a and 5d. BON3 expression is
enriched in cells with the highest Pseudotime scores. **c**, Expression of *WRKY8*
and *LSD1* in the sub-clusters of clusters 3, 7, and 11 in the snRNA-seq data (Fig. 1f).
**d**, Violin plot showing expression *PUB36* in the PRIMER cell cluster of WT and
*GT-3A* knockout (gt3a-KO) plants. The gene model of *PUB36*, ATAC-seq peaks,
and a GT-3A binding motif are shown below. **e**, GO enrichment analysis of genes
downregulated in a bystander cell cluster of gt3a-KO compared to WT. Adjusted
p-values from a one-sided hypergeometric test followed by Benjamini-Hochberg
correction are shown. **f**, Violin plot showing expression of *ALD1* in a bystander
cluster. **d-f**, Two independent replicates of each sample were analyzed with
single-nucleus RNA-seq. **g**, Expression of *BON3* and *GT-3A* in cells in mock (top)
or DC3000-infection (bottom) condition. **h**, Expression of *ICS1* in cells infected
by AvrRpt2. **c,g,h**, All time points were combined. **i**, Motif activity of GT-3A
in immune-active mesophyll cells. **j**, Lesion area created by *Colletotrichum
higginsianum* infection in Col-0 and two independent *GT-3A* overexpression
lines and *pad3* mutant at 6 days post inoculation. Different letters indicate
statistical significance (adjusted P < 0.01). Adjusted p-values were calculated
with two-tailed Student's t-test followed by Benjamini–Hochberg method.
n = 45, 35, 30, and 51 leaves for WT, *GT-3A-ox1*, *GT-3A-ox2*, and *pad3*, respectively,
from three independent replicates. Results are shown as box plots with boxes
displaying the 25th–75th percentiles, the centerline indicating the median,
whiskers extending to the minimum, and maximum values no further than
1.5 interquartile range. **k**, The gene models of *CBP60g* and *SARD1* with ATAC-seq
peaks and GT-3A binding motifs.

# Reporting Summary

## Statistics

For all statistical analyses, confirm that the following items are present in the figure legend, table legend, main text, or Methods section.

| n/a | Confirmed | |
|---|---|---|
| ☐ | ☒ | The exact sample size (*n*) for each experimental group/condition, given as a discrete number and unit of measurement |
| ☐ | ☒ | A statement on whether measurements were taken from distinct samples or whether the same sample was measured repeatedly |
| ☐ | ☒ | The statistical test(s) used AND whether they are one- or two-sided *Only common tests should be described solely by name; describe more complex techniques in the Methods section.* |
| ☒ | ☐ | A description of all covariates tested |
| ☐ | ☒ | A description of any assumptions or corrections, such as tests of normality and adjustment for multiple comparisons |
| ☐ | ☒ | A full description of the statistical parameters including central tendency (e.g. means) or other basic estimates (e.g. regression coefficient) AND variation (e.g. standard deviation) or associated estimates of uncertainty (e.g. confidence intervals) |
| ☐ | ☒ | For null hypothesis testing, the test statistic (e.g. *F*, *t*, *r*) with confidence intervals, effect sizes, degrees of freedom and *P* value noted *Give P values as exact values whenever suitable.* |
| ☒ | ☐ | For Bayesian analysis, information on the choice of priors and Markov chain Monte Carlo settings |
| ☒ | ☐ | For hierarchical and complex designs, identification of the appropriate level for tests and full reporting of outcomes |
| ☐ | ☒ | Estimates of effect sizes (e.g. Cohen's *d*, Pearson's *r*), indicating how they were calculated |

*Our web collection on statistics for biologists contains articles on many of the points above.*

## Software and code

Policy information about availability of computer code

| Data collection | Illumina NovaSeq 6000, Vizgen MERSCOPE, BGI DNBSEQ-G400, GloMax Navigator Microplate Luminometer (Promega) |
|---|---|
| Data analysis | Seurat (v5.0.3), ggplot2 (v3.3.5), Signac (v1.12.9007),  harmony (v0.1.0), org.Athaliana.eg.db (v0.1),  chromVAR (1.14.0),  STAR (v2.6.1b), fastp (v0.19.7), cellranger (v6.0.1), cellranger-arc (v2.0.0), cellpose (v2.1.1), scanpy (v1.9.1), scikit-learn (v1.1.2), spacexr (v2.2.1), Baysor (v0.5.2), scvi-tools (v1.0.2), clusterProfiler (v4.12.6), big-fish (v0.6.2), numpy (v1.24.4), opencv-python(v4.10.0.84), pygam (v0.9.1), edgeR (v4.2.1), limma (v3.60.6), ImageJ (v2.14.0), irlba (v2.3.5.1), qvalue (v2.36.0), Socrates (v0.0.9), scCustomize (v2.1.2).<br><br>Code availability:<br>The code to analyze snMultiome and MERFISH data is available at https://github.com/tnobori/snMultiome and https://github.com/amonell/Spatial_Plant_Pathogen_Atlas. |

For manuscripts utilizing custom algorithms or software that are central to the research but not yet described in published literature, software must be made available to editors and reviewers. We strongly encourage code deposition in a community repository (e.g. GitHub). See the Nature Portfolio guidelines for submitting code & software for further information.

## Data

Policy information about availability of data

All manuscripts must include a data availability statement. This statement should provide the following information, where applicable:

- Accession codes, unique identifiers, or web links for publicly available datasets
- A description of any restrictions on data availability
- For clinical datasets or third party data, please ensure that the statement adheres to our policy

All information supporting the conclusions are provided with the paper. The single-cell and bulk sequencing data generated in this study are deposited in the National Center for Biotechnology Information Gene Expression Omnibus database (accession no. GSE226826 and GSE248054). Reference genome, annotation, fully processed data for snMultiome analyses are available at neomorph.salk.edu/download/Nobori_etal_merfish. The MERFISH data are available at plantpathogenatlas.salk.edu. Genes targeted with MERFIHS are listed in Supplementary Table 1. Information of primers used in this study is provided in Supplementary Table 2. Source data are provided with this paper. Reference genome, annotation, fully processed data for snMultiome analyses are available at neomorph.salk.edu/download/Nobori_etal_merfish.

## Human research participants

Policy information about studies involving human research participants and Sex and Gender in Research.

| | |
|---|---|
| Reporting on sex and gender | N/A |
| Population characteristics | N/A |
| Recruitment | N/A |
| Ethics oversight | N/A |

Note that full information on the approval of the study protocol must also be provided in the manuscript.

# Field-specific reporting

Please select the one below that is the best fit for your research. If you are not sure, read the appropriate sections before making your selection.

☒ Life sciences  ☐ Behavioural & social sciences  ☐ Ecological, evolutionary & environmental sciences

For a reference copy of the document with all sections, see nature.com/documents/nr-reporting-summary-flat.pdf

# Life sciences study design

All studies must disclose on these points even when the disclosure is negative.

| | |
|---|---|
| Sample size | No statistical method was used to determine sample size. Instead, sample sizes were determined based on a combination of factors, including standard practices in plant biology research and the objectives of the experiments. For the pathogen growth assay, we selected sample sizes based on their effectiveness in previous similar studies in the field (such as PMID: 35545668, PMID: 37704725, and PMID: 35508659). For the snMultiome and MERFISH time course experiments, we primarily analyzed one replicate for each condition. Second replicates of snMultiome were performed for key conditions, which is a similar setup with a recent time course single-cell RNA-seq study (PMID: 36996230). In addition, we used two different pathogens known to induce similar responses and used matching conditions for independent snMultiome experiments, further ensuring the reproducibility of our data. Although the MERFISH experiments were performed with single replicates, the sample conditions match those of the snMultiome experiments, allowing cross-validation of conclusions from these orthogonal analyses. Therefore, we believe that the sample sizes used for snMultiome and MERFISH experiments are sufficient to support the conclusions of this study. |
| Data exclusions | No data was excluded for the analyses in this study. |
| Replication | For pathogen growth assays and bulk RNA-seq, three to four independent replicates were analyzed for each condition. Two independent replicates were analyzed for snMultiome for Mock and AvrRpt2 9 hr samples. No replicate was used for MERFISH time course analysis.<br><br>All attempts at replication reported were successful. |
| Randomization | Plants were grown in the same tray to ensure consistent environmental conditions. Individual plants were then randomly assigned to different treatment groups. This random allocation was employed to minimize selection bias and distribute any uncontrolled variables evenly across all treatment groups. |
| Blinding | In pathogen infection assays, certain treatments produce visible disease phenotypes, making it impossible to conceal the treatments applied. To minimize potential biases, we implemented consistent experimental protocols and standardized analysis pipelines for all samples. |

# Reporting for specific materials, systems and methods

We require information from authors about some types of materials, experimental systems and methods used in many studies. Here, indicate whether each material, system or method listed is relevant to your study. If you are not sure if a list item applies to your research, read the appropriate section before selecting a response.

## Materials & experimental systems

| n/a | Involved in the study |
|-----|----------------------|
| ☒ | ☐ Antibodies |
| ☒ | ☐ Eukaryotic cell lines |
| ☒ | ☐ Palaeontology and archaeology |
| ☒ | ☐ Animals and other organisms |
| ☒ | ☐ Clinical data |
| ☒ | ☐ Dual use research of concern |

## Methods

| n/a | Involved in the study |
|-----|----------------------|
| ☒ | ☐ ChIP-seq |
| ☒ | ☐ Flow cytometry |
| ☒ | ☐ MRI-based neuroimaging |

