## [Peer Review File · Nature]

A rare PRIMER cell state in plant immunity

Corresponding Author: Professor Joseph Ecker

Version 0:

Reviewer comments:

Referee #1

(Remarks to the Author)

This manuscript reports a large-scale analyses of Arabidopsis leaves infected with the bacterial pathogen *Pseudomonas syringae* using a variety of cutting-edge approaches (snMultiome, snRNA-seq + snATAC-seq, and spatial transcriptomics). The paper is well-written and computational analyses are well-done. This study will be an important resource for the plant-microbe community. However, the functional analyses and mechanistic insight is the weak point of the paper. In the past year, there have been several single-cell and spatial studies on the Arabidopsis-bacteria/fungal interaction (*Pseudomonas syringae* and *Colletotrichum*: three for single-cell and a recent combination of spatial metatranscriptomics for both plant and pathogens in *Nature Biotechnology*). This manuscript goes beyond those mentioned but is missing the level of biological insight likely required for this journal.

The concept of PRIMER cells is interesting, but functional analyses are lacking aside from overexpression of GT-3a. Many immune regulators have SA pathways impaired when overexpressed, what makes Gt-3a unique in this? How does the knockout line behave? What is the mechanism behind GT-3a negative regulation and is this a common mechanism for other pathogens?

The concept of spatial restriction of immune activated cells directly surrounding cells undergoing ETI and their cell-cell communication is not an entirely new concept. For example: *Nature Plants* 2023, 9:1184-1190; *Molecular Plant* 2022, 6:1059-1075; *Plant Cell Physiology* 2018, 59:8-16). How do PRIMER cells play a role in this phenomenon and is GT-3a a critical regulator in this process (or one of many synergistic regulators)?

How do negative regulators in PRIMER cells spread immune signals to the surrounding cells? As noted in *Nature Plants* 2023, 9:1184-1190, *Pseudomonas* growth is not restricted in the actual area undergoing ETI, but bacterial spread outside of this zone is restricted. The hypothesis that dying cells spread the signal has been noted in multiple papers, but without genetic evidence.

Does GT-3A overexpression show uncontrolled or altered spatial patterns in plant immune activation compared to wild type? Similar to B0N3/WRKY8/LSD1, do knockouts of these genes show impaired distribution of plant immune response?

Two avirulent *Pseudomonas* (AvrRpt2 and AvrRpm1) strains were used in the manuscript. The timing and amplitude of immune responses triggered by these two effectors are different. Were you able to capture alterations in immune response intensity/distribution between the two?

Referee #2

(Remarks to the Author)

Dr. Nobori and colleagues have put together an impressive dataset focused on understanding transcriptional dynamics associated with plant immunity and virulent disease caused by a well-studied model plant/pathogen system (*Pseudomonas syringae* vs. *Arabidopsis thaliana*). Overall, I would recommend this paper for publication. I found the paper to be of high interest to the not only the plant/pathogen field, but also more broadly to those studying multi-kingdom interactions, that can serve as a model for future studies. Combined single-cell multiome data is rare in the plant functional genomics literature, and the combination of this with spatially-resolved transcriptomics via MERFISH is highly novel. The main insight derived by the authors (the identification of a late-stage "PRIMER" cell state) was of particular interest. I was also impressed by the novel methodology in plant image segmentation, imputing transcription factor-target relationships using multiome data and existing TF binding motifs, and in implementing multiple experimental factors into a single-cell study. However, I do have

some concerns that I believe should be addressed that will (I hope) improve the manuscript. These relate to analysis code, data presentation, experimental design, and other details that need more clarification.

Data/Code:

1. After looking through the code repositories (on github), I found most of it sufficiently documented (with the R scripts better documented than the python code), and well-functionalized (a rarity, in my opinion).
2. The major flaws in both repositories, however, are that they include hard links to data repositories that are not accessible (even via GEO, which hosts the finished R objects, and would require some pre-processing to partially recreate). Thus, I was not able to run the scripts to verify that they work (though I have no doubt that they do).
3. Data pre-processing scripts were also not provided (i.e. to go from sequence data to raw counts matrices), which could resolve another question I had about data processing (discussed below).
4. While both repositories do well to include libraries/packages that are needed for their analysis, these could be better provided in containerized form and hosted i.e. on dockerhub, or another container hosting service (or at the very least, provided as a Docker file/folder that can be later built). This is helpful for reproducing figures/analyses that can be very particular to the analysis environment.
5. GEO repository GSE248054 was not accessible without a reviewer token, which I did not find

Data presentation

6. Throughout the manuscript, clusters are often referred to by number, and presented in cluster numerical order. This makes interpretation of the data with respect to cell types difficult. I would suggest the authors re-number (or re-label) each cluster with a cell type label to help evaluate each panel. This would be especially helpful for plots in Ext. Data 1h, Ext. Data 2a and d.
7. For Ext. Data Fig 2d, why are some clusters missing? Did they not have any enrichment with the GO terms?

Experimental Design

8. The authors attempted to mitigate influences of the circadian clock by staggering the time of infection, then isolating infected leaf tissue at the same time of day. This might not totally circumvent clock influence, as plants can have different levels of immune responses depending on the time of day they were infected (see Wang, et al., 10.1038/nature09766, Bhardwaj, et al., 10.1371/journal.pone.0026968, review: Cheng, et al., 10.1016/j.chom.2019.07.009). A more thorough discussion of the impacts of the clock on immunity might be warranted, though I'm not sure deconvolving the clock from the snMultiome data is possible without better controls.
9. The authors also only sampled mock-treated plants at a single time point (9hpi), making analysis at other time points complicated, as the mock infection may induce changes related to tissue damage, which overlap with immunity. I think this is mostly important for accurate pseudotime interpolation, since it's difficult to know exactly how a control-treated plant would look like at each of the time points sampled (can't assume it will be like the sampled control). A discussion on experimental considerations (and justifications thereof) would add to this paper's strength.

Other details

10. In data pre-processing, the authors mention they used cellranger and cellranger-arc to analyze single-nuclei RNA and ATAC-seq separately. I don't have experience with multiome data personally, but I thought that cellranger-arc handled both modalities. Was there something specific about the version of cellranger used for RNA that was better than the method for cellranger-arc? Some clarity here might help future multiome data generators.
11. Was there any assessment of chloroplast contamination? The authors mentioned the genome was removed from the analysis pipelines, and I'm not sure whether it would impact analysis, but it could be useful to include some metrics for others attempting to evaluate their data.
12. What about the default peak calling in cellranger-arc led to the re-annotation of peaks using cluster-specific data? Again, this would be very useful for the community.
13. When imputing a pseudotime trajectory, the authors mentioned they selected a start cell based on location in UMAP space, however as UMAP includes some stochastic steps, this definition is subject to some change. The authors do include the cell name as a hard-coded line in their jupyter notebook, but it might be even safer to have this called out in the manuscript as well.
14. For the bacterial neighborhood analysis, what was the rationale behind selecting a 5-cell neighborhood? In their UMAP overlay of pseudotime and DC3000 colony detection, some colonies are detected in close proximity to the 24h pseudotime "hotspot" (Ext. Data Fig 7f), which could undermine the claim that DC3000 location is not associated with immune activity. I also wonder whether this under-representation of immune hotspots DC3000-infected leaf is reproducible across multiple plants, or whether it was artifact of the time selected (see Fig 5f, where 9h infected plants had lots of immune regions, while the 24 time point seems to have fewer). In either case (for the AvrRpt2 and DC3000 infected tissues), colony size does not appear to correlate well with pseudotime (Ext. Fig. 7g). Was ICS1 smFISH done on this tissue as well? This might provide further evidence that DC3000 is suppressing immune activity.
15. The authors rely heavily on clustering (and sub-clustering) their single-cell multiome dataset for defining cell states, cell types, etc. They state that the "major clusters" might be too broadly defined and that they may miss additional heterogeneity. I wonder whether the authors considered clustering their data at a higher resolution, or using a tool like scSHC (Grabski, et al., 10.1038/s41592-023-01933-9) which applies a statistical test to determine whether cells should be included in a cluster or not, and also provides a method to de-novo cluster cells.
16. L91-92 authors state that increased number of ATAC-seq ACRs in snMultiome dataset is likely due to the ability to capture cell type-specific loci. Is there a way to quantify this? That is, look for peaks that are unique to the ATAC-seq dataset, then test whether they are cell type-specific? This could be a neat result if true.
17. L124-125 authors state that some clusters contain a mix of immune states, likely due to developmental signatures, but do not elaborate – what do they mean by developmental signature?
18. L261-263 and Ext. Data Fig. 7 – authors state that the imputed expression of ALD1 correlating with the MERFISH-

derived expression of the same gene is evidence that the integration is accurate, however ALD1 is part of the dataset that was used for integration. This seems circular to me. I think the better evidence is from ICS1, which was not in the MERFISH dataset. Related, more details might be useful on the smFISH in conjunction with MERFISH; this was not clear to me.

Referee #3

(Remarks to the Author)

The manuscript by Nobori et al profiled time-resolved single-cell RNA-seq and ATAC-seq as well as the spatial transcriptomics in Arabidopsis leaves infected by bacterial pathogen.

This work is an extension of two recent single-cell transcriptomics studies of Arabidopsis leaf cells infected by *Pseudomonas syringae* (Zhu et al 2023 and Tang et al 2023). Not surprisingly, the authors found the cell state diversity and heterogeneity in pathogen-infected plant cells. While snATAC-seq and spatial transcriptomics are relatively new and interesting but aside from a great resource for plant immunity community, the data and analysis do not yield significant new findings. The key conclusions are similar from the bulk RNA-seq and ATAC-seq data. The manuscript also falls short of providing both conceptual and technical advances that would make it a strong candidate for Nature.

Major concerns:

- The authors tried to integrate the snRNA-seq and snATAC-seq data and identified the close correlation of ACRs with some defense genes. However, the expression of ~50% defense gene is independent from chromatin states. Thus, the functional significance of chromatin and epigenetic regulation is unclear. How about the links between gene expression and chromatin accessibility in bulk RNA-seq and ATAC-seq data? If most links and non-links were also observed in bulk analysis, I am wondering what advantage and new information learned from single cell analysis when exploring the relationship of gene expression and chromatin accessibility.
- Another major concern is the data acquisition and analysis. The authors collected 4h, 6h, 9h, and 24h infection time points but only used mock (water infiltration) at 9h as a control. It has been established that some defense genes are regulated by the circadian/clock. Thus, it is important to have mock controls for all corresponding infection time points to separate the cells responding to clock vs infection.
- Cell type clustering is another concern. The classification of cell types is largely dependent on the PCA analysis of the snRNA-seq or snATAC-seq data. However, the dataset used for cell type clustering contain a mixed population of mock and infected cells, with a large proportion of mock cells (>15,000 mock cells out of 65,061 cells) (Fig.1b). This approach naturally introduced the heterogeneity and diversity, and artificially clustered the immune-responsive cells from the mock cells. The cell type clustering should be done separately with mock and infected cells. The authors can first use the mock cells to determine the cell types and then identify immune-responsive cell types in infected cell population using immune marker genes.
- Identification of the PRIMER cells (Fig. 6) are the most interesting part of the paper. However, lacking controls for the infiltration damage is a concern. The authors identified a rare cell population responding to infection. Are these cells clustered due to the mechanical infection damage? Why certain cells are sensitive to infection? Do these immune-responsive cells/cell types have different chromatin states? What is the function of these immune-responsive cells/cell types?
- In Line 155-175, 763, "linkage" or "linkage analysis" are not suitable words here. In genetics and genomics, "linkage" and "linkage analysis" are referred to the close locations of different DNA sequences.
- The pseudotime analysis and the related conclusion were largely based on the LinearGAM function. However, it appears that very few immune genes have the pattern of linear change after infection. Since the authors already have the data from different time points, why not use the real data instead of pseudotime analysis?
- In Line 307-316, very few AvrRpt2 strains were overlapped with tissue regions with high pseudotime scores based on Extend Dat Fig. 7f. More data and analysis are needed for making this conclusion.

Minor concerns:

- In Line 220, the reference for MERFISH was incorrect.
- In Line 232-235, the Fig. 4b and Fig. 4c were not correctly cited in the text.

Version 1:

Reviewer comments:

Referee #1

(Remarks to the Author)

The authors have addressed my previous concerns and comments. In particular, the combination of PRIMER and bystander cells is biologically interesting and will open the door for many investigations in the future. I do not have additional

comments or concerns.

Referee #2

(Remarks to the Author)

Dear Editor,

I have reviewed the revisions to the manuscript by Dr. Nobori and colleagues, and found their responses to mine and other reviewer's critiques both thoughtful and sufficient to overcome my concerns.

Referee #3

(Remarks to the Author)

This revised manuscript has significantly improved by providing additional experimental data and extensive analyses and has addressed most of my concerns.

Dear Reviewers,

We greatly appreciate the time the reviewers took to thoroughly evaluate and improve our study. We were pleased that the reviewers were generally enthusiastic about the identification of an immune cell state, PRIMER cell, that emerges at the nexus of immune active regions in infected plant leaves as well as the resource value of our data sets.

In the revised manuscript, we have provided further mechanistic insights into the role of PRIMER cells and their potential regulators, including a newly characterized transcription factor, GT-3A. We have also extended our analysis to another fundamental immune cell state that we have molecularly identified—bystander cell—that surrounds PRIMER cells and performed comprehensive comparisons between these immune cell states. The newly provided data and discussions signify the conceptual advance of our study. Detailed point-by-point responses are provided below.

The text is colored according to the following scheme:

Referees' comments

Our responses

Newly added or revised texts and figure legends

Referee #1 (Remarks to the Author):

This manuscript reports a large-scale analyses of Arabidopsis leaves infected with the bacterial pathogen *Pseudomonas syringae* using a variety of cutting-edge approaches (snMultiome, snRNA-seq + snATAC-seq, and spatial transcriptomics). The paper is well-written and computational analyses are well-done. This study will be an important resource for the plant-microbe community. However, the functional analyses and mechanistic insight is the weak point of the paper. In the past year, there have been several single-cell and spatial studies on the Arabidopsis-bacteria/fungal interaction (*Pseudomonas syringae* and *Colletotrichum*: three for single-cell and a recent combination of spatial metatranscriptomics for both plant and pathogens in *Nature Biotechnology*). This manuscript goes beyond those mentioned but is missing the level of biological insight likely required for this journal.

The concept of PRIMER cells is interesting, but functional analyses are lacking aside from overexpression of GT-3a. Many immune regulators have SA pathways impaired when overexpressed, what makes Gt-3a unique in this? How does the knockout line behave?

Thank you for acknowledging the importance of our study as a resource for the plant-microbe interactions community and for your interest in our finding of PRIMER cells. While negative regulation of SA pathways may be common among many immune regulators, what makes GT-3A unique is its highly restricted expression in PRIMER cells. Our GT-3A-ox data implies that this rare cell state-specific gene negatively regulates SA pathways.

To further address the reviewer's comments on functional analyses, we have generated a CRISPR KO mutant of GT-3A. Interestingly, the mutants also became more susceptible to infection of an ETI-triggering pathogen (Figure 6m), suggesting that the function of GT-3A,

potentially the suppression of SA pathways, in PRIMER cells is required for optimal defense against pathogens.

To clarify the significance of our finding of PRIMER cells, we have defined another cell state, bystander cell, which surrounds PRIMER cells (Figure 6e). Comparisons between PRIMER and bystander cells revealed distinct gene regulation in these fundamental immune cell states. Notably, CAMTA and WRKY motifs were enriched in PRIMER and bystander cells, respectively (Figure 6i). We also found cell state-specific transcriptome signatures that suggest the role of CAMTAs in PRIMER cells (Figure 6j). Additionally, we have performed snRNA-seq in *gt3a-KO* and showed that *GT-3A* in PRIMER cells may be important for the induction of genes involved in SA pathways in bystander cells (Extended Data Fig. 8f), illuminating a potential mode of communication between these cell states. We have provided a conceptual model based on our findings as a new figure (Figure 6n).

We speculate that while SA is a positive regulator of immunity against *Pseudomonas syringae*, this function may be specific to bystander cells, and the suppression of SA pathways may be important for immunity in PRIMER cells.

We believe that our findings and dataset open the door for the biology of plant immune cell states, which will significantly advance our understanding of the plant immune system as a multicellular system. We also believe that a deeper understanding of immune cell states requires significant technical advance to simultaneously monitor the molecular states of both plant and pathogen cells in their native context. These points have been discussed in our revised manuscript (Lines 326-335 and 353-362, shown below).

Figure 6m: Growth of *AvrRpt2* (infiltrated at $OD_{600} = 0.001$) in WT and *GT-3A* knockout line at 24 hpi. The asterisk indicates statistical significance (adjusted $P < 0.01$).

Figure 6i: Enriched motifs linked to genes strongly expressed in PRIMER (top) or bystander (bottom) cells.

Extended Data Fig. 8i: Motif activity score of CAMTA3

Figure 6j: Venn diagram of genes highly expressed in PRIMER or bystander cells and genes previously shown to be suppressed by CAMTA3.

Extended Data Fig. 8e: GO enrichment analysis of genes downregulated in a bystander cell cluster of gt3a-KO compared with WT.

Extended Data Fig. 8f: Violin plot showing expression ALD1 in a bystander cluster.

Figure 6n: A proposed model for the potential role and regulation of immune cell states.

These extensive new results have been incorporated into the main text.

Lines 325-335: “Therefore, we designate this cell state as the “PRImary ImmunE Responder (PRIMER)” cell from which immune responses might spread. Additionally, cells surrounding PRIMER cells are designated as bystander cells.

PRIMER and bystander cells showed distinct transcriptional and epigenetic signatures. We found that CAMTA motifs as well as a GT-3A motif were highly enriched in PRIMER cell-specific genes, whereas bystander cells were enriched with WRKY motifs (Figure 6i, Extended

Data Fig. 8i). Genes previously shown to be repressed by CAMTA3 were significantly overrepresented in bystander cells compared to PRIMER cells (Figure 6j; FDR = 5.0e-46 with hypergeometric test corrected by Benjamini-Hochberg method), supporting the transcriptional repressive role of CAMTA3 in PRIMER cells.

Lines 353-362: “We also tested a GT-3A knockout mutant (gt3a-KO) and showed that the mutant became more susceptible against AvrRpt2 (Fig. 6m), implying that the function of GT-3A in PRIMER cells is required for optimal defense against pathogens. snRNA-seq analysis of gt3a-KO mutant revealed genes that are potentially regulated by GT-3A, either directly or indirectly (Supplementary Table 7). Among such genes, expression of PUB36 in PRIMER cells was significantly impaired in gt3a-KO at both 9 and 24 hpi; PUB36 has GT-3A binding motif in the upstream region (Extended Data Fig. 8d). We also found that SA-related genes including ALD1 were lowly expressed in bystander cells in gt3a-KO compared with WT (Extended Data Fig. 8e,f), implying that induction of GT-3A in PRIMER cells may be important for proper induction of defense genes in surrounding cells.”

What is the mechanism behind GT-3a negative regulation and is this a common mechanism for other pathogens?

Mode of action of GT-3A

In a previous study, a trihelix transcription factor, BdTHX1, was shown to bind to an intronic region of a gene and suggested to transcriptionally repress the gene (Fan et al., 2018, PMID: 30224432). It is possible that a similar mechanism is at play for GT-3A, another a trihelix transcription factor. Interestingly, we found a GT-3A binding motif in the introns of key SA regulators (*CPB60g* and *SARD1*) (Supplementary Figure 3), which provides an interesting future opportunity to study the mechanisms of action of GT-3A in PRIMER cells. It has also been shown that trihelix TFs can form heterodimers as well as homodimers (Xie et al., 2009; Li et al., 2015); it would be informative to study proteins interacting with GT-3A in PRIMER cells. Another interesting finding is that infection by DC3000, an immune-suppressive pathogen, did not activate PRIMER cell genes (Extended Data Fig. 8g), suggesting that PRIMER cells are related to immunity rather than susceptibility (e.g., pathogens exploiting the response). Since a similar gene expression was observed in AvrRpt2-ETI and AvrRpm1-ETI, which originated from different pathogens, it is possible that the role of GT-3A is common between multiple pathogens. These points have been discussed in the Supplementary Information with an additional figure (Supplementary Information Lines 83-91).

Supplementary Information Lines 84-91: “In a previous study, a trihelix transcription factor BdTHX1 was shown to bind to an intronic region of a gene and was implied to transcriptionally suppress the gene. It is possible that a similar mechanism is at play for GT-3A. Intriguingly, we found a GT-3A binding motif in the introns of key SA regulators (CPB60G and SARD1) (Supplementary Figure 3), providing an interesting future opportunity to study the mechanisms of action of GT-3A in PRIMER cells. We found that PRIMER cell genes were not induced by DC3000 (Extended Data Fig. 8g), an immune-suppressive pathogen, suggesting that PRIMER cells are involved in immunity rather than susceptibility (e.g., pathogen exploiting the response).”

Supplementary Figure 3: The gene models of *CBP60g* and *SARD1* with ATAC-seq peaks and GT-3A binding motifs.

Extended Data Fig. 8g: Expression of *BON3* and *GT-3A* in cells in mock (top) or DC3000-infection (bottom) condition.

Regarding other pathogens:

To address the reviewer's question about other pathogens, we have tested the *GT-3A-ox* line for resistance against a fungal pathogen, *Colletotrichum higginsianum* (*Ch*), and showed that the mutant became susceptible to the fungal pathogen (Extended Data Fig. 8j), a result consistent with the bacterial pathogen data (Figure 6l). A previous study demonstrated that the SA pathway is essential for resistance against *Ch* (Liu et al., 2007, PMID: 17918632). Thus, this result aligns with the model suggesting that *GT-3A* negatively regulates the SA pathway when ectopically expressed. Our results imply that there may be an additional function of this TF that influences defense against fungal pathogens. We also note that a recent study has shown that *GT-3A* overexpression negatively regulates defense against nematodes in the root (Zhao et al., 2024). These results suggest that *GT-3A* functions in both roots and shoots against various types of pathogens. We have revised our manuscript to incorporate these information (Lines 391-394).

*Lines 391-394: "A recent study has shown that *GT-3A* negatively regulates plant defense against nematodes in the root, suggesting that *GT-3A* may play a role in defense against different types of pathogens that attack other tissues. We found that *GT-3A-ox* was more susceptible to the fungal pathogen *Colletotrichum higginsianum* (Extended Data Fig. 8j), supporting this hypothesis."*

Colletotrichum higginsianum
infection

Extended Data Fig. 8j: Lesion area created by *Colletotrichum higginsianum* infection in *Col-0* and two independent *GT-3A* overexpression lines and *pad3* mutant at 6 days post inoculation. Different letters indicate statistical significance (adjusted $P < 0.01$).

The concept of spatial restriction of immune activated cells directly surrounding cells undergoing ETI and their cell-cell communication is not an entirely new concept. For example: Nature Plants 2023, 9:1184-1190; Molecular Plant 2022, 6:1059-1075; Plant Cell Physiology 2018, 59:8-16).

We agree with the reviewer that the spatial zonation of immune responses has been recognized at a conceptual level with some supporting data, such as live imaging of selected genes or omics analysis of dissected tissue regions. However, our data provided an unparalleled depth of molecular and spatial information that supports and refines the conceptual model. The tissue dissection approach (such as Salguero-Linares et al., 2022, PMID: 35502144) is not able to capture the difference between PRIMER and bystander cells since these cell states are only one (or potentially a few) cells away, which goes beyond the resolution limit of previous approaches. Thus, our findings and datasets are highly unique products of the integration of single-cell multiomics and imaging-based spatial transcriptomics (MERFISH). This point has been discussed in Lines 369-373.

Lines 369-373: "Our molecularly defined spatiotemporal atlas of pathogen-infected leaves reveals various cell states with transcriptome and epigenome details, providing means to investigate individual cell states that have been hidden in conventional bulk/dissected tissue analyses and live imaging of a limited number of reporter lines."

How do PRIMER cells play a role in this phenomenon and is *GT-3a* a critical regulator in this process (or one of many synergistic regulators)?

We speculate that there are regulators other than *GT-3A* in PRIMER cells. As shown above, we have identified that other TF motifs, including CAMTA1/2/3 motifs, were significantly enriched in PRIMER cell-specific genes (Figure 6i). We also found cell state-specific transcriptome signatures suggesting the role of CAMTAs in PRIMER cells (Figure 6j). We have provided more data and discussion in our response to the reviewers' next question.

Dissecting the role of these TFs and downstream genes is an exciting future research area that our study has opened up.

How do negative regulators in PRIMER cells spread immune signals to the surrounding cells? As noted in *Nature Plants* 2023, 9:1184-1190, *Pseudomonas* growth is not restricted in the actual area undergoing ETI, but bacterial spread outside of this zone is restricted. The hypothesis that dying cells spread the signal has been noted in multiple papers, but without genetic evidence.

This is a very important question, and our data provide several hypotheses. There are many PRIMER cell-specific genes with unknown functions, some of which may serve as a mobile signal. Also, *ICS1* (a major SA biosynthesis gene) is induced in PRIMER cells (Extended Data Fig. 8h), implying that SA may be produced in PRIMER cells then translocated to neighboring cells via plasmodesmata or the apoplast. A previous study showed that SA accumulates in the apoplast upon pathogen infection (Carviel et al., 2014, PMID: 24594657). Understanding how the PRIMER cells function and communicate with neighboring cells will be a new research topic in the MPMI field, and we believe our study provides a wealth of useful information. We have discussed these points in the revised manuscript (shown below).

Supplementary Information Lines 94-102: “Various signaling molecules, such as ROS, Ca, and DAMPs, may be involved in the communication between PRIMER and bystander cells. We found a number of PRIMER cell-specific genes with unknown functions, some of which may serve as mobile signals. Also, ICS1 (a major SA biosynthesis gene) is induced in PRIMER cells (Extended Data Fig. 8h), suggesting that SA may be produced and translocated to neighboring cells via plasmodesmata or the apoplast. A previous study showed that SA accumulates in the apoplast and increases upon pathogen infection⁵. Understanding how the PRIMER cell functions and communicates with neighboring cells will be a new research topic in the field of molecular plant-microbe interaction, and we believe that our study provides a wealth of useful information.”

Extended Data Fig. 8h: Expression of ICS1 in cells infected by AvrRpt2. All time points were combined.

To understand more about the impact of GT-3A in neighboring cells, we have performed snRNA-seq in a *GT-3A* knockout mutant. We have identified genes, including *ALD1* whose expression was dampened in a bystander cell cluster in *gt3a-KO* compared to WT (Extended Data Fig. 8f), implying that *GT-3A* induction in PRIMER cells may be important for proper

induction of defense genes in the surrounding cells. We have discussed this point in Lines 359-362.

Lines 359-362: “We also found that SA-related genes including ALD1 were lowly expressed in bystander cells in gt3a-KO compared with WT (Extended Data Fig. 8e,f), implying that induction of GT-3A in PRIMER cells may be important for proper induction of defense genes in surrounding cells.”

Co-analysis of plant gene expression and bacterial distribution, effector translocation is key to understanding the mechanisms of different cell states, which will require substantial technical innovation in the future. We have discussed this point in the Supplementary Information (shown below).

Supplementary Information Lines 116-126: “Simultaneous spatial mapping of plant gene expression and bacterial colonization has great potential in elucidating interactions between plant cells and bacterial cells at the single-cell level (Extended Data Fig. 7g-i). In the future, time series MERFISH analysis of both immune-activating and -suppressive pathogen infection in replicates would provide a deeper understanding of how spatial relationships between plant and pathogen cells influence the responses of individual plant cells. However, it is challenging or impossible to fully capture such interactions on a 2D tissue section, and thus a 3D analysis of both plant genes and bacterial colonization is critical. Recently, we developed a technology called PHYTOmap, which can spatially map dozens of genes in 3D in whole-mount plant tissues. Such a method, combined with our spatiotemporal atlas, will open a new avenue toward a comprehensive characterization of cell populations identified in this study.”

Does GT-3A overexpression show uncontrolled or altered spatial patterns in plant immune activation compared to wild type? Similar to BON3/WRKY8/LSD1, do knockouts of these genes show impaired distribution of plant immune response?

This is an interesting question. We did not test the spatial pattern of gene expression in GT-3A-ox. Based on bulk RNA-seq, SA genes were repressed at the tissue level, and as snRNA-seq and MERFISH showed, SA genes are highly induced in bystander cells, so we speculate that these genes are misregulated due to uncontrolled overexpression of GT-3A. As discussed above, snRNA-seq of gt3a-KO showed that SA-related genes are misregulated in bystander cells in the absence of GT-3A in PRIMER cells (Extended Data Fig. 8e,f); we also found genes that are misregulated in PRIMER cells of gt3a-KO. To address the reviewer’s question, the following sentences have been added in the revised manuscript with new figures.

Lines 356-362: “snRNA-seq analysis of gt3a-KO mutant revealed genes that are potentially regulated by GT-3A, either directly or indirectly (Supplementary Table 7). Among such genes, expression of PUB36 in PRIMER cells was significantly impaired in gt3a-KO at both 9 and 24 hpi; PUB36 has GT-3A binding motif in the upstream region (Extended Data Fig. 8d). We also found that SA-related genes including ALD1 were lowly expressed in bystander cells in gt3a-KO compared with WT (Extended Data Fig. 8e,f), implying that induction of GT-3A in PRIMER cells may be important for proper induction of defense genes in surrounding cells.”

Extended Data Fig. 8d: Violin plot showing expression *PUB36* in the PRIMER cell cluster of WT and GT-3A knockout (*gt3a*-KO) plants. The gene model of *PUB36*, ATAC-seq peaks, and a GT-3A binding motif are shown below.

Extended Data Fig. 8e: GO enrichment analysis of genes downregulated in a bystander cell cluster of *gt3a*-KO compared with WT.

Extended Data Fig. 8f: Violin plot showing expression *ALD1* in a bystander cluster.

Two avirulent *Pseudomonas* (AvrRpt2 and AvrRpm1) strains were used in the manuscript. The timing and amplitude of immune responses triggered by these two effectors are different. Were you able to capture alterations in immune response intensity/distribution between the two?

Thank you for raising an important point. Previous bulk RNA-seq analysis revealed stronger and faster ETI triggered by AvrRpm1 than AvrRpt2 (Mine et al., 2018, PMID: 29794063). We observed more cells activating immune responses when infected by AvrRpm1. In Extended Data Fig. 1i, we have shown that a higher proportion of cells of AvrRpm1-infected leaves are found in immune active clusters (such as clusters 3, 11, and 12). Additionally, we have analyzed the distribution of gene expression level in each sample and in each major cluster. For many immune marker genes, we observed faster and stronger expression in the cells of AvrRpm1-infected leaves (Supplementary Figure 2). These results suggest that both the number of cells and the amplitude of immune responses contribute to the different immune responses observed at the bulk tissue level. We have discussed this point with a figure in the Supplementary Information.

Supplementary Figure 2: Violin plots showing expression of defense-related genes in cells in an immune-active cell cluster (cluster 3 of Fig. 1b).

Supplementary Information Lines 105-113: “Previous bulk RNA-seq analysis revealed stronger and faster ETI triggered by AvrRpm1 than by AvrRpt2. To determine whether AvrRpm1-ETI activates immune responses in a larger number of cells and whether the intensity of these responses is heightened within individual cells, we analyzed the distribution of gene expression levels across different samples, focusing on an immune-active major cluster (cluster 3 in Figure 1b). Our observations revealed that many immune marker genes exhibited faster and stronger expression in cells from AvrRpm1-infected leaves (Supplementary Figure 2). These findings suggest that both an increase in the number of responsive cells and the heightened amplitude of immune responses contribute to the overall differential immune response observed at the bulk tissue level.”

Referee #2 (Remarks to the Author):

Dr. Nobori and colleagues have put together an impressive dataset focused on understanding transcriptional dynamics associated with plant immunity and virulent disease caused by a well-studied model plant/pathogen system (*Pseudomonas syringae* vs. *Arabidopsis thaliana*). Overall, I would recommend this paper for publication. I found the paper to be of high interest to the not only the plant/pathogen field, but also more broadly to those studying multi-kingdom interactions, that can serve as a model for future studies. Combined single-cell multiome data is rare in the plant functional genomics literature, and the combination of this with spatially-resolved transcriptomics via MERFISH is highly novel. The main insight derived by the authors (the identification of a late-stage “PRIMER” cell state) was of particular interest. I was also impressed by the novel methodology in plant image segmentation, imputing transcription factor-target relationships using multiome data and existing TF binding motifs, and in implementing multiple experimental factors into a single-cell study. However, I do have some concerns that I believe should be addressed that will (I hope) improve the manuscript. These relate to analysis code, data presentation, experimental design, and other details that need more clarification.

Thank you for sharing your enthusiasm for our study and for providing useful suggestions to significantly improve our manuscript. We have now improved the data/code availability and provided more explanations for your questions. We have also performed additional experiments and analyses to strengthen the biological significance of our findings.

Data/Code:

1. After looking through the code repositories (on github), I found most of it sufficiently documented (with the R scripts better documented than the python code), and well-functionalized (a rarity, in my opinion).

Thank you for your positive comments. We have further improved the documentation and user-friendliness of our R and Python code (<https://github.com/tnobori/snMultiome> and https://github.com/amonell/Spatial_Plant_Pathogen_Atlas).

2. The major flaws in both repositories, however, are that they include hard links to data repositories that are not accessible (even via GEO, which hosts the finished R objects, and would require some pre-processing to partially recreate). Thus, I was not able to run the scripts to verify that they work (though I have no doubt that they do).

We have updated our GitHub repositories so that users can directly access source data hosted on our server. For instance, raw output files from the Cellranger-arc pipeline are available at http://neomorph.salk.edu/download/Nobori_etal_merfish/multiome/10x_outputs/, and our R script (1_qc_data_integration_figS1.R) directly loads these outputs to recreate combined datasets.

3. Data pre-processing scripts were also not provided (i.e. to go from sequence data to raw counts matrices), which could resolve another question I had about data processing (discussed below).

We have provided data preprocessing scripts, which can be accessed in our GitHub repository ([tnobori/snMultiome/scripts/snMultiome_preprocessing_cellranger.sh](https://github.com/tnobori/snMultiome/scripts/snMultiome_preprocessing_cellranger.sh)).

4. While both repositories do well to include libraries/packages that are needed for their analysis, these could be better provided in containerized form and hosted i.e. on dockerhub, or another container hosting service (or at the very least, provided as a Docker file/folder that can be later built). This is helpful for reproducing figures/analyses that can be very particular to the analysis environment.

Thank you for your suggestion to further improve the reproducibility of our analyses. We have created devcontainers that can be used to run our spatial processing pipelines (Python and R) in a containerized form with Visual Studio code. Below is a screenshot of the “setup” section of our updated GitHub repository (https://github.com/amonell/Spatial_Plant_Pathogen_Atlas). For snMultiome analysis pipelines (R), we have provided a script (`scripts/_config_multiome.R`) that installs necessary packages.

Setup

This repository contains a `devcontainer` to allow to run the scripts in a reproducible manner. Please see the documentation for further information on how to use devcontainers. We have three devcontainers for the spatial pipelines, and each notebook should be run with the devcontainer specified at the top. The Python scripts can be run with devcontainers creating the environments `python_plant_pathogen_atlas` and `python_scvi_environment`, and all R scripts can be run with a devcontainer creating the environment `r_plant_pathogen_atlas`.

Running from VS Code

1. Open this repository in VS Code.
2. Open the Command Palette:
 - Windows/Linux: `Ctrl+Shift+P`
 - macOS: `Cmd+Shift+P`
3. Type and select "Remote-Containers: Open Folder in Container..."
4. Select this project directory.
5. Choose the devcontainer you want to use.

We are running Ubuntu 22.04.3 LTS.

5. GEO repository GSE248054 was not accessible without a reviewer token, which I did not find

Data presentation

We are sorry that we did not make the data accessible. We have now made the repository public.

6. Throughout the manuscript, clusters are often referred to by number, and presented in cluster numerical order. This makes interpretation of the data with respect to cell types difficult. I would suggest the authors re-number (or re-label) each cluster with a cell type label to help evaluate each panel. This would be especially helpful for plots in Ext. Data 1h, Ext. Data 2a and d.

Thank you for the suggestion. We agree that explicitly linking cluster numbers to cell type annotations would make it easier for readers to interpret our results. To address this, we have added cell type labels to the relevant figures, namely Fig. 1c, Ext. Data 1h, Ext. Data 2a and d.

7. For Ext. Data Fig 2d, why are some clusters missing? Did they not have any enrichment with the GO terms?

The missing clusters did have GO terms enriched. However, showing GO terms enriched for each cluster makes the figure too large, so we restricted the number of GO terms to be displayed. We have provided a supplementary table (Supplementary Table 8) summarizing the GO terms enriched in each cluster.

Experimental Design

8. The authors attempted to mitigate influences of the circadian clock by staggering the time of infection, then isolating infected leaf tissue at the same time of day. This might not totally circumvent clock influence, as plants can have different levels of immune responses depending on the time of day they were infected (see Wang, et al., 10.1038/nature09766, Bhardwaj, et al., 10.1371/journal.pone.0026968, review: Cheng, et al., 10.1016/j.chom.2019.07.009). A more thorough discussion of the impacts of the clock on immunity might be warranted, though I'm not sure deconvolving the clock from the snMultiome data is possible without better controls.

Thank you for raising an important point about the timing of infection. An alternative experimental design, often used in bulk omics studies, would have been to simultaneously infect plants and harvest at different times with appropriate mock controls for each sampling time point. This was practically challenging for our single-cell analysis because we needed to process as many samples as possible at the same time to run on the same 10x Genomics chip, and we needed to use fresh tissues.

This study aimed to capture different cell states in pathogen-infected leaves, which would include circadian effects. Our analysis did not focus on real-time gene expression dynamics, except for the pseudotime analysis to model an immune trajectory. Therefore, we believe that the current experimental setup is sufficient to draw most of the main conclusions of our study. The pseudotime trajectory inferred from the time series data (with potential circadian effects) formed a reasonable spatial gradient (Fig. 5f), suggesting that circadian effects did not significantly influence our pseudotime analysis. As suggested by the reviewer, we have discussed the potential influence of the clock in our revised manuscript in the Supplementary Information.

Supplementary Information Lines 21-35: "In a study with a time-course experiment, any circadian effect is of major concern and needs to be addressed with careful experimental design and analyses. In this study, we harvested all samples at a similar time of day to mitigate any potential circadian effect. This was achieved by infiltrating pathogens into leaves at different times of the day. An alternative experimental design, as often used in bulk omics studies, would have been to infect plants simultaneously and harvest at different times with proper mock controls for each sampling time point. This was practically challenging for our

single-cell analysis as we had to process as many samples as possible at the same time to run on the same 10x Genomics chip; and we had to use fresh tissues. This study aimed to capture diverse cell states in pathogen-infected leaves, which would include circadian effects. Our analysis did not focus on real-time course gene expression dynamics except for the pseudotime analysis to model an immune trajectory. The pseudotime trajectory inferred from the time series data (with potential circadian impact) formed a reasonable spatial gradient (Fig. 5f), implying that circadian effects did not significantly influence our pseudotime analysis. Therefore, we believe that the current experimental setup is sufficient to draw most of the main conclusions of our study.”

9. The authors also only sampled mock-treated plants at a single time point (9hpi), making analysis at other time points complicated, as the mock infection may induce changes related to tissue damage, which overlap with immunity. I think this is mostly important for accurate pseudotime interpolation, since it's difficult to know exactly how a control-treated plant would look like at each of the time points sampled (can't assume it will be like the sampled control). A discussion on experimental considerations (and justifications thereof) would add to this paper's strength.

We chose the 9-hour time point for the Mock treatment because this is when *Pto* AvrRpt2, the strain central to our study, elicited the strongest immune response in the plant. We acknowledge the reviewer's concern that infiltration may induce a wound response, particularly at earlier time points. To address this, we examined our snMultiome data for any signs of a wounding response. However, we did not observe clear indications of wound responses by AvrRpt2 at 4 hpi compared to Mock at 9 hpi (Rebuttal Figure 1 shows expression of *VSP2*, a known wound-induced gene). The absence of a strong immune or wound response to AvrRpt2 at 4 hpi suggests that tissue damage was minimal at this time point. Additionally, we note that syringe infiltration was performed on one or two corners of the leaf, and the bacterial suspension spread throughout the entire leaf. The entire leaves were then collected for analysis. This approach likely minimized the proportion of wounded cells, if any, in our data set. We have clarified the infiltration method in the revised manuscript (shown below).

Rebuttal Figure 1: Expression of *VSP2* in AvrRpt2-infected leaves.

Lines 864-866: "Syringe infiltration was performed on one or two corners of each leaf, and bacterial suspensions were spread throughout the entire leaf."

Other details

10. In data pre-processing, the authors mention they used cellranger and cellranger-arc to analyze single-nuclei RNA and ATAC-seq separately. I don't have experience with multiome data personally, but I thought that cellranger-arc handled both modalities. Was there

something specific about the version of cellranger used for RNA that was better than the method for cellranger-arc? Some clarity here might help future multiome data generators.

Thank you for pointing this out. The reviewer is correct, and the snMultiome data were all analysed with Cellranger-arc. In the revised manuscript, we added snRNA-seq (10x Chromium HT) data that was analysed with Cellranger, so we did not change the description in the Method.

11. Was there any assessment of chloroplast contamination? The authors mentioned the genome was removed from the analysis pipelines, and I'm not sure whether it would impact analysis, but it could be useful to include some metrics for others attempting to evaluate their data.

We observed the mean of 23.4*% and 26.0% of chloroplast reads in the RNA and ATAC data, respectively. We have rerun cellranger-arc with a reference genome/gene annotation with the chloroplast. Removing the chloroplast from the reference did not affect the final distribution of RNA and ATAC counts (Supplementary Figure 1). We have discussed this point in the Method section of the revised manuscript (Lines 987-991).

Lines 987-991: "We note that the mean of 23.4% and 26.0% of reads were mapped on the chloroplast genome in snRNA-seq and snATAC-seq, respectively. Removing the chloroplast genome from the reference did not affect the overall RNA and ATAC count distribution (Supplementary Figure 1). Count data were analyzed with the R packages Seurat v5 and Signac"

Supplementary Figure 1: Violin plots showing average transcripts per nucleus, genes per nucleus, ATAC reads per nucleus, and accessible chromatin regions (ACRs) per

nucleus across each sample analyzed with or without chloroplast information in the reference.

12. What about the default peak calling in cellranger-arc led to the re-annotation of peaks using cluster-specific data? Again, this would be very useful for the community.

Thank you for the suggestion. We compared the default peak calling and our peak calling. Compared to the default peak calling pipeline of cellranger-arc (24,394 peaks), this cluster-specific peak calling approach was able to capture more peaks (35,560 peaks), which is consistent with what was reported in the original Signac paper (Stuart et al., 2001, PMID: 34725479). We have discussed these points in the Methods section of the revised manuscript (Lines 1018-1020).

Lines 1018-1020: "Compared to the default peak calling pipeline of cellranger-arc (24,394 peaks), this cluster-specific peak calling approach was able to capture more peaks (35,560 peaks), which is consistent with a previous report."

13. When imputing a pseudotime trajectory, the authors mentioned they selected a start cell based on location in UMAP space, however as UMAP includes some stochastic steps, this definition is subject to some change. The authors do include the cell name as a hard-coded line in their jupyter notebook, but it might be even safer to have this called out in the manuscript as well.

We have revised our approach by computing pseudotime trajectories starting from each Mesophyll cell in the mock samples. These individual trajectories are averaged within each cell to create a single, unified pseudotime trajectory. The new results are overall very similar to our original analyses. We have updated the figures (Fig. 5d-f, Fig. 6a) and Methods.

Lines 1081-1084: "Pseudotime trajectories were constructed with each mesophyll cell in mock samples being used as a starting cell. Then, these individual trajectories were averaged within each cell to create a single, unified pseudotime trajectory. "

14. For the bacterial neighborhood analysis, what was the rationale behind selecting a 5-cell neighborhood? In their UMAP overlay of pseudotime and DC3000 colony detection, some colonies are detected in close proximity to the 24h pseudotime "hotspot" (Ext. Data Fig 7f), which could undermine the claim that DC3000 location is not associated with immune activity. I also wonder whether this under-representation of immune hotspots DC3000-infected leaf is reproducible across multiple plants, or whether it was artifact of the time selected (see Fig 5f, where 9h infected plants had lots of immune regions, while the 24 time point seems to have fewer). In either case (for the AvrRpt2 and DC3000 infected tissues), colony size does not appear to correlate well with pseudotime (Ext. Fig. 7g). Was ICS1 smFISH done on this tissue as well? This might provide further evidence that DC3000 is suppressing immune activity.

We have revised this analysis accordingly by calculating the average pseudotime value in neighborhoods ranging from 1 to 1000 nearest spatial neighbors. We observed a decreasing and increasing trend in the average pseudotime score in AvrRpt2 24 h and DC3000 24 h, respectively (Extended Data Fig. 7i, shown below), suggesting that plant cells near pathogen cells are highly immune-active upon AvrRpt2 infection, while the opposite is true for DC3000

infection. We have also provided *ICS1* smFISH analysis, which showed the lower expression of *ICS1* in the DC3000-infected leaves (Extended Data Fig. 7f, shown below). We noted, in the original manuscript, limitations of the current spatial co-mapping of plant gene expression and bacterial distribution. We have further revised this section based on the reviewer's comment about the generality of our observation in the present study (Supplementary Information Lines 116-126). We have also updated the Methods (Lines 1246-1252).

Supplementary Information Lines 116-126: "Simultaneous spatial mapping of plant gene expression and bacterial colonization has great potential in elucidating interactions between plant cells and bacterial cells at the single-cell level (Extended Data Fig. 7g-i). In the future, time series MERFISH analysis of both immune-activating and -suppressive pathogen infection in replicates would provide a deeper understanding of how spatial relationships between plant and pathogen cells influence the responses of individual plant cells. However, it is challenging or impossible to fully capture such interactions on a 2D tissue section, and thus a 3D analysis of both plant genes and bacterial colonization is critical. Recently, we developed a technology called PHYTOmap, which can spatially map dozens of genes in 3D in whole-mount plant tissues. Such a method, combined with our spatiotemporal atlas, will open a new avenue toward a comprehensive characterization of cell populations identified in this study."

Extended Data Fig. 7i: Averaged pseudotime scores of cells surrounding each bacterial colony. The x-axis shows the number of nearest plant cells analyzed for each bacterial colony.

Extended Data Fig. 7f: smFISH of *ICS1* in leaves infected by *AvrRpt2* or *DC3000* at 24 hpi.

Lines 1246-1252: "Bacterial neighborhood analysis

To determine the level of immunity of the neighborhood around each bacteria colony, we identified the 1 to 1000 nearest cells in proximity to the colony. Then, the smoothed imputed pseudotime values of these neighboring cells were averaged to obtain a single mean pseudotime value for each neighbor size. Error curves were calculated using the standard error divided by the mean at each neighborhood size. These values serve as indicators of the overall immunity level of the area around each colony.”

15. The authors rely heavily on clustering (and sub-clustering) their single-cell multiome dataset for defining cell states, cell types, etc. They state that the “major clusters” might be too broadly defined and that they may miss additional heterogeneity. I wonder whether the authors considered clustering their data at a higher resolution, or using a tool like scSHC (Grabski, et al., 10.1038/s41592-023-01933-9) which applies a statistical test to determine whether cells should be included in a cluster or not, and also provides a method to de-novo cluster cells.

We agree that our major clusters contain additional heterogeneity. That is why we performed an additional round of clustering (sub-clustering) and identified various molecular signatures. PRIMER cells were identified from one of the sub-clusters (Figure 6e). Determining biologically relevant clustering resolution is an active area of research that likely requires sample-specific optimization and validation. As suggested by the reviewer, we have tested scSHC with a batch-corrected mode and the testClusters function. However, we did not observe any improvement in clustering in our analysis (Rebuttal Figure 2). Therefore, we kept the current clustering approach.

Rebuttal Figure 2: UMAP plot colored and labeled with cluster annotations inferred by scSHC.

16. L91-92 authors state that increased number of ATAC-seq ACRs in snMultiome dataset is likely due to the ability to capture cell type-specific loci. Is there a way to quantify this? That is, look for peaks that are unique to the ATAC-seq dataset, then test whether they are cell type-specific? This could be a neat result if true.

We compared our peaks with previously published bulk ATAC-seq data from immune-active Arabidopsis leaves (Ding et al., 2021, PMID: 34387350). We found that 98% of the peaks identified in bulk ATAC-seq were also detected in our snATAC-seq, whereas 46% of the peaks

detected in snATAC-seq were unique to the single-cell data. This result was discussed in the revised manuscript in Lines 83-85 (shown below). We also analyzed the proportion of cluster-specific peaks within shared or unique peaks. 18% and 2% of shared and unique peaks, respectively, were cluster-specific peaks, suggesting that in many cases cell type-specific peaks have been identified in bulk ATAC-seq without information on which cell type these peaks are enriched for, supporting the value of single-cell analyses.

Lines 83-85 “Our snATAC-seq data identified more ACRs than previously reported bulk ATAC-seq data from immune-activated leaves with large overlap, suggesting that snATAC-seq can capture both known and unknown ACRs (Extended Data Fig. 1e).”

Extended Data Fig. 1e: Bar plot showing the number of ATAC-seq peaks identified in previous bulk ATAC-seq data and the present snATAC-seq data. Shared and assay unique peaks are shown in blue and red, respectively.

17. L124-125 authors state that some clusters contain a mix of immune states, likely due to developmental signatures, but do not elaborate – what do they mean by developmental signature?

Immune active cells and non-active cells could be clustered together because they share substantial number of cell type marker gene expression. We have added more explanation in the revised manuscript to clarify this point (Lines 117-120).

Lines 117-120: “Although the major clusters captured immune-active cells in mesophyll and epidermis (Fig. 1c), clusters for other cell types, such as vasculature, contained both immune-active and non-active cells, likely due to strong developmental signatures (e.g., vasculature marker genes are strongly expressed regardless of immune activation).”

18. L261-263 and Ext. Data Fig. 7 – authors state that the imputed expression of ALD1 correlating with the MERFISH-derived expression of the same gene is evidence that the integration is accurate, however ALD1 is part of the dataset that was used for integration. This

seems circular to me. I think the better evidence is from ICS1, which was not in the MERFISH dataset. Related, more details might be useful on the smFISH in conjunction with MERFISH; this was not clear to me.

Thank you for this suggestion. We agree with the reviewer, and we now focus on *ISC1* smFISH data to validate our imputation approach (Lines 259-261). We have also provided additional explanations on smFISH in the Method (Lines 1132-1133).

Lines 259-261: "Imputed ICS1 (not included in the MERFISH panel) expression accurately predicted real spatial expression of ICS1 (based on smFISH) (Extended Data Fig. 7a,b), indicating the accuracy of data imputation."

Lines 1132-1133: "The smFISH result was provided as an image without quantitative information."

Referee #3 (Remarks to the Author):

The manuscript by Nobori et al profiled time-resolved single-cell RNA-seq and ATAC-seq as well as the spatial transcriptomics in Arabidopsis leaves infected by bacterial pathogen. This work is an extension of two recent single-cell transcriptomics studies of Arabidopsis leaf cells infected by *Pseudomonas syringae* (Zhu et al 2023 and Tang et al 2023). Not surprisingly, the authors found the cell state diversity and heterogeneity in pathogen-infected plant cells. While snATAC-seq and spatial transcriptomics are relatively new and interesting but aside from a great resource for plant immunity community, the data and analysis do not yield significant new findings. The key conclusions are similar from the bulk RNA-seq and ATAC-seq data. The manuscript also falls short of providing both conceptual and technical advances that would make it a strong candidate for Nature.

We are sorry that we did not adequately communicate the conceptual and technical advances of our study. We provided significant conceptual advances made only possible by advanced technologies and analyses done in our study. We also believe that the findings, data set, and methods are of great interest to the broader research community. The manuscript has now been extensively revised to strengthen these points. We first address the reviewer's main criticisms: (1) conceptual advance, (2) Technical advance, and (3) Why single-cell and spatial omics? We also emphasize (4) the value of this study as a resource.

Conceptual advances

We have identified many potentially novel immune cell states that have been missed in previous studies using bulk tissue analysis (Fig 1d). As noted by the reviewer (and other reviewers), one of the most interesting is the PRIMER cell. To clarify and strengthen the significance of this finding, we have highlighted another fundamental cell state, the bystander cell, which surrounds PRIMER cells. Our newly added data provide further functional and regulatory insights into these fundamental immune cell states. For instance, we found differential enrichment of CAMTA and WRKY TF binding sites in PRIMER and bystander cells, respectively (Figure 6i,j). We have also performed snRNA-seq in a newly generated CRISPR KO mutant of *GT-3A*, a PRIMER cell marker gene, and found potential roles for this TF in these two immune cell states (Extended Data Fig. 8d-f). Studying these cell states is critical to understanding cell autonomous (PRIMER cells) and non-cell autonomous (bystander cells)

responses. Some of the most important questions in the field of plant immunity (as highlighted in a recent review by Jones et al., 2024, PMID: 38670067) are "what happens in the cell where ETI has been induced (cell-autonomous responses)?" and "how does local ETI spread to neighboring cells (non-cell autonomous responses)?" Our study lays the groundwork for addressing these questions. To highlight this conceptual advance, we have provided a schematic diagram regarding the roles of PRIMER and bystander cells in plant immunity (Figure 6n).

Figure 6n: A proposed model for the potential role and regulation of immune cell states.

Technical advances

There have been efforts to understand plant immunity with spatial and temporal resolution by employing live imaging of selected genes or omics analysis of dissected tissue regions (such as Betsuyaku et al., 2018, PMID: 29177423; Salguero-Linares et al., 2022, PMID: 35502144; Zhou et al., 2020, PMID: 32032516). However, the novel findings in our study were only possible through the integration of snMultiome and MERFISH, which provided an unprecedented depth of molecular and spatial information compared to previously available datasets. Unbiased single-cell omics analysis identified distinct cell populations, such as PRIMER and bystander cells, that cannot be distinguished by a tissue dissection approach (Salguero-Linares et al., 2022), because these cell states are only one (or potentially a few) cells away, which is beyond the resolution limit of previous approaches. Multiplexed *in situ* gene expression analysis allowed efficient validation of observed cell populations in a spatial context, which is not feasible with traditional imaging with reporter lines, due to the large number of genes to be tested. Thus, our findings and datasets are a unique product of the integration of single-cell multiomics and imaging-based spatial transcriptomics (MERFISH), which we believe represents a major technical advance in the plant biology field. This point has been discussed in Lines 369-373 (shown below).

Lines 369-373: "Our molecularly defined spatiotemporal atlas of pathogen-infected leaves reveals various cell states with transcriptome and epigenome details, providing means to investigate individual cell states that have been hidden in conventional bulk/dissected tissue analyses and live imaging of a limited number of reporter lines."

Why single-cell and spatial omics?

To highlight the advances over traditional bulk omics, we further compared our snMultiome data with bulk RNA-seq and ATAC-seq datasets. **RNA:** snRNA-seq provides higher resolution for studying gene expression than bulk RNA-seq. We have analyzed previously published time-course bulk RNA-seq data and shown that bulk RNA-seq data fails to capture cell population information. We've also demonstrated a case in gene co-expression analysis where bulk RNA-seq data can potentially lead to misinterpretation of data. In this example, we showed that *CRK12* and *CRK41* are highly correlated in the time-course bulk RNA-seq data, which may imply that these genes function together in the same cell. However, in our snRNA-seq data, these genes were expressed largely in different cells. These examples highlight the value of single-cell transcriptome analyses compared to bulk transcriptome analyses. **ATAC:** We compared our ATAC-seq peaks with previously published bulk ATAC-seq data from immune-active Arabidopsis leaves (Ding et al., 2021, PMID: 34387350). We found that 98% of the peaks identified in bulk ATAC-seq were also detected in our snATAC-seq, whereas 46% of the peaks detected in snATAC-seq were unique to the single-cell data. These points have been clarified in the revised manuscript (shown below).

Lines 125-129: "The subclustering of our snRNA-seq data revealed complex immune responses within leaf tissue, which is not captured by bulk RNA-seq (Extended Data Fig. 3h). We also found cases where genes appear to be highly co-expressed at the bulk transcriptome level but uniquely expressed at the sub-cluster level (Extended Data Fig. 3i,j), further highlighting the value of single-cell analyses."

Extended Data Fig. 3h, Heatmap showing expression of immune-related genes shown in Figure 1d in a previous time-course bulk RNA-seq study, where *A. thaliana* leaves were infected by DC3000, *AvrRpt2*, or *AvrRpm1* or treated with mock control (water). These data do not capture the presence of various cell populations identified in Figure 1d.

Extended Data Fig. 3i, Scatter plots showing the correlation between CRK12 and CRK41 in the time course bulk RNA-seq (top) and single-nucleus RNA-seq (snRNA-seq; bottom). For snRNA-seq analysis, pseudobulk gene expression data of the subclusters (shown in c) were used.

Extended Data Fig. 3j: Expression of CRK12 and CRK41 upon infection with AvrRpm1 as shown in the UMAP. These apparently correlated genes in bulk RNA-seq showed a highly different expression pattern at the single-cell level.

Lines 83-85: “Our snATAC-seq data identified more ACRs than previously reported bulk ATAC-seq data of immune-activated leaves, with large overlaps, suggesting that snATAC-seq can capture both known and unknown ACRs (Extended Data Fig. 1e). “

Extended Data Fig. 1e: Bar plot showing the number of ATAC-seq peaks identified in previous bulk ATAC-seq data and the present snATAC-seq data. Shared and assay unique peaks are shown in blue and red, respectively.

We would also like to highlight the power of the integrative analysis of single-cell multiomics and spatial transcriptomics. The identification of GT-3A, a PRIMER cell-specific TF that plays a role in immunity, was made possible by the enrichment of this TF in the PRIMER cell population in independent snRNA-seq (Figure 6f) and MERFISH (Figure 6g) datasets, further supported by snATAC-seq data showing enrichment of GT-3A binding sites in the genome of PRIMER cells (Figure 6h). This serves as a proof-of-concept for the identification of novel cell state-specific function and gene regulatory mechanisms that can be applied to a wide range of questions in plant biology. We have revised our manuscript to clarify these points.

Lines 380-382: “our integrative snMultiome and MERFISH analysis identified a TF, GT-3A, as another PRIMER cell marker gene and demonstrated that it also negatively regulates plant immunity against pathogen attack (Fig. 6i,j).”

Value as a resource

In addition to providing a dataset of great importance, which this reviewer and other reviewers appreciate, this study pioneered several analyses of the unprecedented dataset in the field. These include novel cell segmentation strategies, multimodal integration of single-cell and spatial data, and spatial mapping of snMultiome data. Based on the suggestions of Reviewer 2, we have significantly improved the data and code availability so that a broad community can benefit from our new analysis pipelines. Please see our updated analysis pipeline on our GitHub repositories (<https://github.com/tnobori/snMultiome> and https://github.com/amonell/Spatial_Plant_Pathogen_Atlas).

Major concerns:

- The authors tried to integrate the snRNA-seq and snATAC-seq data and identified the close correlation of ACRs with some defense genes. However, the expression of ~50% defense gene is independent from chromatin states. Thus, the functional significance of chromatin and epigenetic regulation is unclear. How about the links between gene expression and chromatin accessibility in bulk RNA-seq and ATAC-seq data? If most links and non-links were also observed in bulk analysis, I am wondering what advantage and new information learned from single cell analysis when exploring the relationship of gene expression and chromatin accessibility.

There is clear functional significance in linked ACRs and genes. For example, WRKY motifs are significantly correlated with defense gene expression, which is consistent with the knowledge that WRKYs are critical defense TFs. Thus, our single-cell data can capture such known and also most likely unknown regulatory relationships. The lack of correlation between ACRs and gene expression is also interesting. For example, some ACRs are always open while associated genes are tightly regulated, suggesting other regulatory mechanisms, such as DNA methylation, which has been proposed in the context of plant immunity before (Halter et al., 2021, PMID: 33470193). Overall, our single-cell multiome data revealed various modes of defense gene regulation, providing useful information for future mechanistic studies. We have discussed this point further in the Supplementary Information (Lines 68-81).

Supplementary Information Lines 69-81: "We found some immune genes were "not linked" and associated with constitutively opened chromatin, while their expression was tightly regulated upon pathogen attack (Fig. 2f). The upstream regions of such non-linked immune genes were enriched with WRKY binding motifs (Fig. 2g,h), implying that tight control of WRKY TFs, which can act as transcriptional activators or repressors, may regulate downstream genes. Constitutively accessible chromatin may facilitate the quick binding of TFs to the DNA for rapid induction of defense genes. It is also possible that other epigenetic modifications regulate non-linked genes. It has been shown that DNA methylation could inhibit the binding of WRKY TFs to DNA by steric hindrance. It would be interesting to profile DNA methylation at the single-cell level to see if there is such gene regulation. Recently developed methods enable simultaneous single-cell profiling of transcriptome, chromatin accessibility, and DNA methylation from the same cells. Such "single-cell triple-omics" technologies will further advance our understanding of gene regulation in plants."

Some links could also be found in bulk omics. However, identifying links between ACRs and genes from bulk omics data would require large-scale RNA-seq and ATAC-seq with matching sample conditions, which, to our knowledge, are not currently available. Since our snATAC-seq identified more ACRs than bulk ATAC-seq (discussed above), it is likely that the single-cell approach has a higher power to detect links than bulk data.

We believe our study is unique in identifying various cell populations (such as PRIMER and bystander cells) with potential regulatory links specific to certain cell populations that bulk RNA and ATAC-seq analyses cannot reveal.

- Another major concern is the data acquisition and analysis. The authors collected 4h, 6h, 9h, and 24h infection time points but only used mock (water infiltration) at 9h as a control. It has been established that some defense genes are regulated by the circadian/clock. Thus, it is

important to have mock controls for all corresponding infection time points to separate the cells responding to clock vs infection.

Thank you for raising this important point. We chose 9 h for Mock because this is the time point where *Pto* AvrRpt2 (the strain of focus in our study with both multiome and MERFISH analyses) triggered the strongest response in the plant. We intended to ensure that the strong response was not due to an infiltration artifact. Nevertheless, your concern about clock regulation is also an important point. As indicated in the Method section, we harvested all samples at a similar time of day to mitigate the potential circadian effect. This was achieved by infiltrating pathogens to leaves at different times of the day. It is possible that there is a potential impact of the time of infection, but this cannot be controlled by including a mock in each time point. This study aimed to capture diverse cell states in pathogen-infected leaves, which would include circadian effects. Our analysis did not focus on real-time course gene expression dynamics. Thus, we believe the current experimental setup is sufficient to draw the main conclusions in our study. As the reviewer suggested, we discussed the potential impact of the clock more thoroughly in our revised manuscript in Supplementary Information.

Supplementary Information Lines 21-35: "In a study with a time-course experiment, any circadian effect is of major concern and needs to be addressed with careful experimental design and analyses. In this study, we harvested all samples at a similar time of day to mitigate any potential circadian effect. This was achieved by infiltrating pathogens into leaves at different times of the day. An alternative experimental design, as often used in bulk omics studies, would have been to infect plants simultaneously and harvest at different times with proper mock controls for each sampling time point. This was practically challenging for our single-cell analysis as we had to process as many samples as possible at the same time to run on the same 10x Genomics chip; and we had to use fresh tissues. This study aimed to capture diverse cell states in pathogen-infected leaves, which would include circadian effects. Our analysis did not focus on real-time course gene expression dynamics except for the pseudotime analysis to model an immune trajectory. The pseudotime trajectory inferred from the time series data (with potential circadian impact) formed a reasonable spatial gradient (Fig. 5f), implying that circadian effects did not significantly influence our pseudotime analysis. Therefore, we believe that the current experimental setup is sufficient to draw most of the main conclusions of our study."

- Cell type clustering is another concern. The classification of cell types is largely dependent on the PCA analysis of the snRNA-seq or snATAC-seq data. However, the dataset used for cell type clustering contain a mixed population of mock and infected cells, with a large proportion of mock cells (>15,000 mock cells out of 65,061 cells) (Fig.1b). This approach naturally introduced the heterogeneity and diversity, and artificially clustered the immune-responsive cells from the mock cells. The cell type clustering should be done separately with mock and infected cells. The authors can first use the mock cells to determine the cell types and then identify immune-responsive cell types in infected cell population using immune marker genes.

As shown in Extended Data Fig. 1h, most clusters showed mixed representation of cells from Mock and each infected sample. When leaves are infected with a low dose of pathogens, some cells do not respond to pathogen attack, thus their molecular signatures resemble Mock cells. Our integrated clustering maximized the identification of diverse cell populations, not

only known developmental cell types, but also immune cell states within cell types, which is a unique contribution of our study. We used *Harmony* to correct batch effects during data integration. Cell type annotation was possible in infected tissues as demonstrated by robust marker gene expression. Using clustering in Mock sample as a reference and integrating infected samples to them might lead to the loss of unique cell populations existing in infected samples (i.e., immune cell states). Thus, we believe our integrated clustering approach is appropriate for this purpose.

- Identification of the PRIMER cells (Fig. 6) are the most interesting part of the paper. However, lacking controls for the infiltration damage is a concern. The authors identified a rare cell population responding to infection. Are these cells clustered due to the mechanical infection damage? Why certain cells are sensitive to infection? Do these immune-responsive cells/cell types have different chromatin states? What is the function of these immune-responsive cells/cell types?

Thank you for recognizing the significance of the PRIMER cells in our study. We have addressed each of your questions separately:

Mechanical damage?

Our mock sample is infiltrated with water, serving as a control for syringe infiltration. We did not find PRIMER cells with marker gene expression in mock condition (Extended Data Fig. 8g), indicating that PRIMER cells emerge in response to pathogen infection. Interestingly, infection by DC3000, an immune-suppressive pathogen, did not activate PRIMER cell genes either (Extended Data Fig. 8g), suggesting that PRIMER cells are related to immunity rather than susceptibility (e.g., pathogen exploiting the response). We have discussed this point in the Supplementary Information.

Supplementary Information Lines 89-91: “We found that PRIMER cell genes were not induced by DC3000 (Extended Data Fig. 8g), an immune-suppressive pathogen, suggesting that PRIMER cells are involved in immunity rather than susceptibility (e.g., pathogen exploiting the response).”

Extended Data Fig. 8g: Expression of BON3 and GT-3A in cells in mock (top) or DC3000-infection (bottom) condition.

Why are certain cells sensitive to infection?

Thank you for this insightful point. The extent to which individual cells' responsiveness to pathogen infection varies remains unclear, and our study was not designed to address this question. While the heterogeneous cell responses we observed might primarily result from the heterogeneous pathogen distribution, it is also plausible that varying "immune potentials" influenced by the basal expression of immune receptors or other physiological states play a role. To address this, more detailed examinations of the spatial relationships between plant and bacterial cells using new technologies are critical. Additionally, developing a technique to trace the molecular state before infection from different immune states after infection will be essential. There are emerging technologies that can potentially be useful in addressing this (for example Choi et al., 2022, PMID: 35794474). We have discussed this important point in Supplementary Information.

Supplementary Information Lines 126-130: "An outstanding question is "Are different immune cell states defined by pre-existing cell states with different immune potentials?" In addition to carefully assessing the spatial relationships between plant and pathogen cells in 3D, it is crucial to develop a new method to trace the molecular state before infection from different immune states after infection."

Do these immune-responsive cells/cell types have different chromatin states?

Thank you for the question. Yes, we have shown that, in addition to GT-3A, CAMTA1/2/3 motifs are highly enriched in the ACRs associated with PRIMER cell-specific genes, whereas WRKY motifs are enriched in bystander cell-associated genes (Figure 6i, Extended Data Fig. 8i), suggesting cell state-specific chromatin states. We have further shown that CAMTA3-suppressed genes were significantly more enriched in bystander cells than in PRIMER cells, suggesting a role for CAMTA3 in bystander cells. The new results and text have been added in the revised manuscript (Lines 329-335).

Lines 329-335: "PRIMER and bystander cells showed distinct transcriptional and epigenetic signatures. We found that CAMTA motifs as well as a GT-3A motif were highly enriched in PRIMER cell-specific genes, whereas bystander cells were enriched with WRKY motifs (Figure 6i, Extended Data Fig. 8i). Genes previously shown to be repressed by CAMTA3 were significantly overrepresented in bystander cells compared to PRIMER cells (Figure 6j; FDR = 5.0e-46 with hypergeometric test corrected by Benjamini-Hochberg method), supporting the transcriptional repressive role of CAMTA3 in PRIMER cells."

Figure 6i: Enriched motifs linked to genes strongly expressed in PRIMER (top) or bystander (bottom) cells.

Extended Data Fig. 8i: Motif activity score of CAMTA3

Figure 6j: Venn diagram of genes highly expressed in PRIMER or bystander cells and genes previously shown to be suppressed by CAMTA3.

What is the function of these immune-responsive cells/cell types?

Thank you for raising an important question that will require extensive future research. To begin to address this question, we have generated a CRISPR knockout mutant of the PRIMER cell marker gene *GT-3A* and shown that the mutant became more susceptible to pathogen infection (Figure 6m), suggesting that proper gene regulation in PRIMER cells is important for successful defense. Additionally, we have performed snRNA-seq in the *GT-3A* knockout mutant and found that different genes are affected in PRIMER and bystander cells. For instance, the induction of a bystander cell marker gene, *ALD1*, was impaired in the *GT-3A* knockout mutant (Extended Data Fig. 8e), implying that PRIMER cells play a role in

transducing immune signals to surrounding cells. GO enrichment analysis showed that GT-3A in PRIMER cells contribute to expression of SA-related genes in bystander cells (Extended Data Fig. 8g). These new data have been incorporated into the revised manuscript (shown below).

Lines 353-362: “We also tested a GT-3A knockout mutant (*gt3a-KO*) and showed that the mutant became more susceptible against *AvrRpt2* (Fig. 6m), implying that the function of GT-3A in PRIMER cells is required for optimal defense against pathogens. snRNA-seq analysis of *gt3a-KO* mutant revealed genes that are potentially regulated by GT-3A, either directly or indirectly (Supplementary Table 7). Among such genes, expression of *PUB36* in PRIMER cells was significantly impaired in *gt3a-KO* at both 9 and 24 hpi; *PUB36* has GT-3A binding motif in the upstream region (Extended Data Fig. 8d). We also found that SA-related genes including *ALD1* were lowly expressed in bystander cells in *gt3a-KO* compared with WT (Extended Data Fig. 8e,f), implying that induction of GT-3A in PRIMER cells may be important for proper induction of defense genes in surrounding cells. “

Figure 6m: Growth of *AvrRpt2* (infiltrated at $OD_{600} = 0.001$) in WT and GT-3A knockout line at 24 hpi. The asterisk indicates statistical significance (adjusted $P < 0.01$).

Extended Data Fig. 8d: Violin plot showing expression *PUB36* in the PRIMER cell cluster of WT and GT-3A knockout (*gt3a-KO*) plants. The gene model of *PUB36*, ATAC-seq peaks, and a GT-3A binding motif are shown below.

Extended Data Fig. 8e: GO enrichment analysis of genes downregulated in a bystander cell cluster of *gt3a*-KO compared with WT.

Extended Data Fig. 8f: Violin plot showing expression of *ALD1* in a bystander cluster

- In Line 155-175, 763, “linkage” or “linkage analysis” are not suitable words here. In genetics and genomics, “linkage” and “linkage analysis” are referred to the close locations of different DNA sequences.

Thank you for pointing this out. We have replaced “linkages” with “links” or “peak/ACR-to-gene linkages”, which is used in a previous study (Signac paper; PMID: 34725479), to avoid confusion.

- The pseudotime analysis and the related conclusion were largely based on the LinearGAM function. However, it appears that very few immune genes have the pattern of linear change after infection. Since the authors already have the data from different time points, why not use the real data instead of pseudotime analysis?

An infected leaf is composed of cells with highly heterogeneous states. The time at which each cell is activated may vary depending on its spatial relationship to pathogen cells, and pathogens may also move during infection. This requires prediction of the state of each cell,

and pseudotime analysis is a commonly used approach to achieve this, although it may not be a perfect solution. Indeed, our real-time data were useful in validating our pseudotime analysis, e.g., cells with low pseudotime values were enriched at early time points. Thus, we believe that our analysis takes advantage of both pseudotime and real-time data to overcome heterogeneity within tissues. We have revised the text to make our idea clearer (shown below).

Lines 276-279: “We hypothesized that by applying pseudotime analysis to a single developmental cell type with various immune states, we could better model the temporal dynamics of the heterogeneous infection/immune responses within an infected leaf.”

- In Line 307-316, very few AvrRpt2 strains were overlapped with tissue regions with high pseudotime scores based on Extend Dat Fig. 7f. More data and analysis are needed for making this conclusion.

We agree that there are AvrRpt2 cells that do not overlap with immune active plant cells. However, it is clear that cells in AvrRpt2-infected plants co-localize with immune active regions more frequently than cells in DC3000-infected plants as shown in Extended Data Fig. 7h. In our initial analysis (Extended Data Fig. 7h), we took 5 nearest neighbour plant cells for each bacterial colony to calculate averaged immune activity. In the revised manuscript, we have strengthened this analysis by testing the number of neighbours ranging from 1 to 1000 (Extended Data Fig. 7i). We observed a decreasing and increasing trend in the average pseudotime score in AvrRpt2 24 h and DC3000 24 h, respectively (Extended Data Fig. 7i), suggesting that plant cells near pathogen cells are highly immune active upon AvrRpt2 infection, while the opposite is true for DC3000 infection.

We also agree with the reviewer that more data are needed to fully understand the spatial relationships between bacterial distribution and plant cell states. We believe that a significant technical advance is needed to achieve this. We have discussed this point in the revised manuscript (Supplementary Information Lines 116-126, shown below)

Extended Data Fig. 7i: Averaged pseudotime scores of cells surrounding each bacterial colony. The x-axis shows the number of nearest plant cells analyzed for each bacterial colony.

Supplementary Information Lines 116-126: “Simultaneous spatial mapping of plant gene expression and bacterial colonization has great potential in elucidating interactions between plant cells and bacterial cells at the single-cell level (Extended Data Fig. 7g-i). In the future, time series MERFISH analysis of both immune-activating and -suppressive pathogen infection in replicates would provide a deeper understanding of how spatial relationships between plant and pathogen cells influence the responses of individual plant cells. However, it is challenging

or impossible to fully capture such interactions on a 2D tissue section, and thus a 3D analysis of both plant genes and bacterial colonization is critical. Recently, we developed a technology called PHYTOmap, which can spatially map dozens of genes in 3D in whole-mount plant tissues. Such a method, combined with our spatiotemporal atlas, will open a new avenue toward a comprehensive characterization of cell populations identified in this study.”

Minor concerns:

- In Line 220, the reference for MERFISH was incorrect.
- In Line 232-235, the Fig. 4b and Fig. 4c were not correctly cited in the text.

Thank you for spotting these errors. We have updated the text accordingly.

Dear Reviewers,

We greatly appreciate the time and effort the reviewers dedicated to evaluating our revised manuscript. We are pleased that our revisions have sufficiently addressed your concerns and we are grateful for your valuable feedback, which has significantly improved the quality of our work. Thank you for your thoughtful consideration.

Referees' comments:

Referee #1 (Remarks to the Author):

The authors have addressed my previous concerns and comments. In particular, the combination of PRIMER and bystander cells is biologically interesting and will open the door for many investigations in the future. I do not have additional comments or concerns.

We are grateful for the reviewer's constructive feedback, especially the suggestions on enhancing the mechanistic insights, which have significantly strengthened our manuscript.

Referee #2 (Remarks to the Author):

Dear Editor,

I have reviewed the revisions to the manuscript by Dr. Nobori and colleagues, and found their responses to mine and other reviewer's critiques both thoughtful and sufficient to overcome my concerns.

We appreciate the reviewer's constructive comments, including feedback on data and code accessibility, which have greatly helped improve our manuscript.

Referee #2 (Remarks on code availability):

The code looks satisfactory, however one notable absence is the reliance on the "sample_sheet.csv" document that is omitted by the authors. Instructions for how to generate this sheet using NCBI accessions would enable someone to run their code with minimal setup. Another omission is code to generate the Arabidopsis reference used to map their data. I did not attempt to install or run the code as this would have taken a great deal of time to sort out (though I believe it is possible to do so).

Thank you for the additional input to improve data accessibility. We have made it easier to download raw fastq files for data analysis by hosting them on our server. We have also provided instructions on how to generate the sample sheet. The Arabidopsis reference was created by manually modifying a TAIR reference, so we cannot provide the code. Instead, we have included a more detailed explanation of the modifications in the Methods section. The reference can also be directly downloaded from our server, which is integrated into the code.

LINE 750: An Arabidopsis thaliana TAIR10 genome was downloaded from <https://plants.ensembl.org/>, and the chloroplast genome was manually removed. The Arabidopsis thaliana TAIR10 gene annotation file was manually modified by removing chloroplast genes and replacing semicolons in the "gene_name" column with hyphens to prevent errors during *cellranger-arc* processing.

The following instructions have been added to github (<https://github.com/tnobori/snMultiome/tree/main>).

snMultiome / scripts / snMultiome_preprocessing_cellranger.sh 
```
tnobori Update snMultiome_preprocessing_cellranger.sh

Code Blame Executable File · 44 lines (35 loc) · 1.68 KB

1  #!/bin/bash
2
3
4  # Reference Files:
5  #
6  # Genome and annotation files:
7  #   - Download the genome and annotation files used in this study from the following URL:
8  #   http://neomorph.salk.edu/download/Nobori_etal_merfish/multiome/arabidopsis_genome/
9  #
10 # Cell Ranger reference file:
11 #   - Download the Cell Ranger reference file from the following URL:
12 #   http://neomorph.salk.edu/download/Nobori_etal_merfish/multiome/cellranger_reference/
13
14
15 # Instructions for Creating a Sample Sheet
16 #
17 # 1. Download FASTQ files:
18 #   - Use a command like `wget` in the terminal to download the FASTQ files from the following URL:
19 #   http://neomorph.salk.edu/download/Nobori_etal_merfish/multiome/fastq_files/
20 #
21 # 2. Create a sample sheet:
22 #   - Follow the guideline provided by 10x Genomics for specifying input FASTQ files:
23 #   https://www.10xgenomics.com/support/software/cell-ranger-arc/latest/analysis/inputs/specifying-input-fastq-count
24 #
25 # 3. The sample sheet should be a CSV file with the following format:
26 #
27 #   fastq,sample,library_type
28 #   path_to_RNA_fastq,sample_name,Gene Expression
29 #   path_to_ATAC_fastq,sample_name,Chromatin Accessibility
30 #
31 # Replace `path_to_RNA_fastq` and `path_to_ATAC_fastq` with the actual paths to your RNA and ATAC FASTQ files, respectively.
32 # Replace `sample_name` with the name of your sample.
33
```

Referee #3 (Remarks to the Author):

This revised manuscript has significantly improved by providing additional experimental data and extensive analyses and has addressed most of my concerns.

We are grateful for the reviewer's feedback, especially the criticism regarding the significance of our research, which has significantly contributed to improving our manuscript.